# Gap-Dependent Bounds for Federated $Q$-Learning

Haochen Zhang [1] [*]  Zhong Zheng [1] [*]  Lingzhou Xue [1]

## Abstract

We present the first gap-dependent analysis of regret and communication cost for online federated $Q$-Learning in tabular episodic finite-horizon Markov decision processes (MDPs). Existing federated reinforcement learning (FRL) methods focus on worst-case scenarios, leading to $\sqrt{T}$-type regret bounds and communication cost bounds with a $\log T$ term scaling with the number of agents $M$, states $S$, and actions $A$, where $T$ is the average total number of steps per agent. In contrast, our novel framework leverages the benign structures of MDPs, such as a strictly positive suboptimality gap, to achieve a $\log T$-type regret bound and a refined communication cost bound that disentangles exploration and exploitation. Our gap-dependent regret bound reveals a distinct multi-agent speedup pattern, and our gap-dependent communication cost bound removes the dependence on $MSA$ from the $\log T$ term. Notably, our gap-dependent communication cost bound also yields a better global switching cost when $M = 1$, removing $SA$ from the $\log T$ term.

## 1. Introduction

Federated reinforcement learning (FRL) is a distributed learning framework that combines the principles of reinforcement learning (RL) (Sutton & Barto, 2018) and federated learning (FL) (McMahan et al., 2017). Focusing on sequential decision-making, FRL aims to learn an optimal policy through parallel explorations by multiple agents under the coordination of a central server. Often modeled as a Markov decision process (MDP), multiple agents independently interact with an initially unknown environment and collaboratively train their decision-making models with limited information exchange between the agents. This

approach accelerates the learning process with low communication costs. In this paper, we focus on the online FRL tailored for episodic tabular MDPs with inhomogeneous transition kernels. Specifically, we assume the presence of a central server and $M$ local agents in the system. Each agent interacts independently with an episodic MDP consisting of $S$ states, $A$ actions, and $H$ steps per episode.

Multiple recent works studied the online FRL for tabular MDPs. Zheng et al. (2024) proposed model-free algorithms FedQ-Hoeffding and FedQ-Bernstein that show the regret bounds $\tilde{O}(\sqrt{MH^4SAT})$ and $\tilde{O}(\sqrt{MH^3SAT})$ respectively under $O(MH^3SA\log T)$ rounds of communications. Here, $T$ is the average total number of steps for each agent, and $\tilde{O}$ hides logarithmic factors. Zheng et al. (2025a) proposed FedQ-Advantage that improved the regret to $\tilde{O}(\sqrt{MH^2SAT})$ under a reduced communication rounds of $O(f_M H^2 SA(\log H)\log T)$ where $f_M \in \{1, M\}$ reflects the optional forced synchronization scheme. Chen et al. (2022) and Labbi et al. (2024) proposed model-based algorithms that extend the single-agent algorithm UCBVI (Azar et al., 2017). Byzan-UCBVI (Chen et al., 2022) reaches regret $\tilde{O}(\sqrt{MH^3S^2AT})$ under $O(MHSA\log T)$ rounds of communications. Fed-UCBVI (Labbi et al., 2024) reaches the regret $\tilde{O}(\sqrt{MH^2SAT})$ under $O(HSA\log T + MHSA\log\log T)$ rounds of communications. Here, model-based methods require estimating the transition kernel so that their memory requirements scale quadratically with the number of states $S$. Model-free methods, which are also called $Q$-Learning methods (Watkins, 1989), directly learn the action-value function, and their memory requirements only scale linearly with $S$. The regret $\tilde{O}(\sqrt{MH^2SAT})$ reached by both FedQ-Advantage and Fed-UCBVI is almost optimal compared to the regret lower bound $\tilde{O}(\sqrt{MH^2SAT})$ (Jin et al., 2018; Domingues et al., 2021). In summary, all the works above provided worst-case guarantees for all possible MDPs and proved $\sqrt{T}$-type regret bounds and communication cost bounds that linearly depend on $MSA\log T$ or $SA\log T$. The results of these works are also summarized in Table 1.

In practice, RL algorithms often perform better than their worst-case guarantees, as they can be significantly improved under MDPs with benign structures (Zanette & Brunskill, 2019). This motivates the problem-dependent analysis exploiting benign MDPs (Wagenmaker et al., 2022a; Zhou

---
[*]Equal contribution [1]Department of Statistics, The Pennsylvania State University, University Park, PA 16802, USA. Correspondence to: Lingzhou Xue <lzxue@psu.edu>.

*Proceedings of the 42$^{nd}$ International Conference on Machine Learning*, Vancouver, Canada. PMLR 267, 2025. Copyright 2025 by the author(s).

*Table 1.* Comparison of online FRL algorithms

| Algorithm | Gap-dependent | Regret | Number of rounds |
|---|---|---|---|
| Byzan-UCBVI (Chen et al., 2022) | $\times$ | $\tilde{O}(\sqrt{MH^3S^2AT})$ | $O(MHSA\log T)$ |
| FedQ-Hoeffding (Zheng et al., 2024) | $\times$ | $\tilde{O}(\sqrt{MH^4SAT})$ | $O(MH^3SA\log T)$ |
| FedQ-Bernstein (Zheng et al., 2024) | $\times$ | $\tilde{O}(\sqrt{MH^3SAT})$ | $O(MH^3SA\log T)$ |
| FedQ-Advantage (Zheng et al., 2025a) | $\times$ | $\tilde{O}(\sqrt{MH^2SAT})$ | $O(f_MH^2SA(\log H)\log T)$ |
| Fed-UCBVI (Labbi et al., 2024) | $\times$ | $\tilde{O}(\sqrt{MH^2SAT})$ | $O^*(HSA\log T)$ |
| Our work | $\checkmark$ | $O^*\left(\frac{H^6SA\log(MSAT)}{\Delta_{\min}}\right)$ | $O^*(H^2\log T)$ |

In this table, $\tilde{O}$ hides logarithmic factors and $O^*$ hides logarithmic lower-order terms, such as $\log\log T$ and $\sqrt{\log T}$, as well as constants. Parameter $f_M \in \{1, M\}$ indicates the optional forced synchronization scheme.

et al., 2023; Zhang et al., 2024b). One of the benign structures is based on the dependency on the positive suboptimality gap: for every state, the best actions outperform others by a margin. It is important because nearly all non-degenerate environments with finite action sets satisfy some sub-optimality gap conditions (Yang et al., 2021). For single-agent algorithms, Simchowitz & Jamieson (2019); Dann et al. (2021) analyzed gap-dependent regret for model-based methods, and Yang et al. (2021); Xu et al. (2021); Zheng et al. (2025b) analyzed model-free methods. Here, Yang et al. (2021) focused on UCB-Hoeffding proposed by Jin et al. (2018), while Xu et al. (2021) proposed an algorithm that did not use upper confidence bounds (UCB). Zheng et al. (2025b) analyzed UCB-Advantage (Zhang et al., 2020) and Q-EarlySettled-Advantage (Li et al., 2021), which used variance reduction techniques. All of these works reached regrets that logarithmically depend on $T$, which is much better than the worst-case $\sqrt{T}$-type regrets. However, no literature works on the gap-dependent regret for online FRL. This motivates the following open question:

*Is it possible to establish gap-dependent regret bounds for online FRL algorithms that are logarithmic in $T$?*

Meanwhile, recent works have proposed FRL algorithms for tabular episodic MDPs in various settings, such as the offline setting (Woo et al., 2024) and scenarios where a simulator is available (Woo et al., 2023; Salgia & Chi, 2024). Different from the online methods, state-of-the-art algorithms for these settings do not update the implemented behavior policies (exploration) and reach $MSA$-free logarithmic bounds on communication rounds, whereas the worst-case communication cost bounds for online FRL methods require the dependence on $M$, $S$, and $A$ in the $\log T$ term (e.g., $O(MH^3SA\log T)$ in Zheng et al. (2024)). While increased communication for exploration is reasonable, existing online FRL methods cannot quantify the communication cost paid for exploring non-optimal actions or exploiting optimal policies under the worst-case MDPs since the suboptimality

gaps can be arbitrarily close to 0 (see Section 5.1 for more explanations). This leads to the dependence on $M$, $S$, and $A$ for the $\log T$ term, which motivates the following open question:

*Is it possible to establish gap-dependent communication cost upper bounds for online FRL algorithms that disentangle exploration and exploitation and remove the dependence on $MSA$ from the $\log T$ term?*

A closely related evaluation criterion for online RL is the global switching cost, which is defined as the times for policy switching. It is important in applications with restrictions on policy switching, such as compiler optimization (Ashouri et al., 2018), hardware placements (Mirhoseini et al., 2017), database optimization (Krishnan et al., 2018), and material discovery (Nguyen et al., 2019). Next, we review related literature on single-agent model-free RL algorithms. Under the worst-case MDPs, Bai et al. (2019) modified the algorithms in Jin et al. (2018), achieving a switching cost of $O(H^3SA\log T)$, and UCB-Advantage (Zhang et al., 2020) reached an improved switching cost of $O(H^2SA\log T)$, with both algorithms depending on $SA\log T$. In gap-dependent analysis, Zheng et al. (2025b) proved that UCB-Advantage enjoyed a switching cost that linearly depends on $S\log T$. Whether single-agent model-free RL algorithms can avoid the dependence on $SA$ for the $\log T$ term remains an open question.

In addition, multiple technical challenges exist when trying to establish gap-dependent bounds and improve the existing worst-case ones. First, gap-dependent regret analysis often relies on controlling the error in the value function estimations. However, the techniques for model-free methods (Yang et al., 2021; Xu et al., 2021; Zheng et al., 2025b) can only adapt to instant policy updates in single-agent methods, while FRL often uses delayed policy updates for a low communication cost. Second, proving low communication costs for FRL algorithms often requires actively estimating the number of visits to each state-action-step triple (see, e.g.,

Woo et al. (2023)). However, this is challenging for online algorithms because the implemented policy is actively updated, and a universal stationary visiting probability is unavailable. Existing online FRL methods reached logarithmic communication costs by controlling the visit and synchronization with the event-triggered synchronization conditions. These conditions guaranteed a sufficient increase in the number of visits to one state-action-step triple between synchronizations. However, this analysis is insufficient for the estimation of visiting numbers and results in the dependence on $SA$ for the $\log T$ term.

**Summary of Our Contributions.** We give an affirmative answer to these important open questions by proving the first gap-dependent bounds on both regret and communication cost for online FRL in the literature. We focus on FedQ-Hoeffding (Zheng et al., 2024), an online FRL algorithm designed for tabular episodic finite-horizon MDPs. Our contributions are summarized as follows.

**Gap-Dependent Regret (Theorem 3.1).** Denote $\Delta_{\min}$ as the minimum nonzero suboptimality gap for all the state-action-step triples. We prove that FedQ-Hoeffding guarantees a gap-dependent expected regret of

$$O\left(\frac{H^6 SA \log(MSAT)}{\Delta_{\min}} + C_f\right) \quad (1)$$

where $C_f = M\sqrt{H^7}SA\sqrt{\log(MSAT)} + MH^5SA$ provides the gap-free part. This bound is logarithmic in $T$ and better than the worst-case $\sqrt{T}$-type regret discussed above when $T$ is large enough. When $M = 1$, (1) reduces to the single-agent gap-dependent regret upper bound established in Yang et al. (2021) for UCB-Hoeffding (Jin et al., 2018), which is the single-agent counterpart of FedQ-Hoeffding. When $T$ is large enough and $\Delta_{\min}$ is small enough, (1) shows a better multi-agent speedup in terms of the average regret per episode, compared to the $\sqrt{T}$-type worst-case regrets shown in Zheng et al. (2024). We will present the theoretical details in Section 3.2 and Section 4. Our numerical experiments in Appendix B.1 also demonstrate the $\log T$-pattern of the regret for any given MDP.

**Gap-Dependent Communication Cost (Theorem 3.3).** We prove that under some general uniqueness of optimal policies, for any $p \in (0, 1)$, with probability at least $1 - p$, both the number of communication rounds and the number of different implemented policies required by FedQ-Hoeffding are upper bounded by

$$O\left(MH^3 SA \log(MH^2 \iota_0) + H^3 SA \log\left(\frac{H^5 SA}{\Delta_{\min}^2}\right)\right.$$
$$\left. + H^3 S \log\left(\frac{MH^9 SA \iota_0}{\Delta_{\min}^2 C_{st}}\right) + H^2 \log\left(\frac{T}{HSA}\right)\right). \quad (2)$$

Here, $C_{st} \in (0, 1]$ represents the minimum of the nonzero visiting probabilities to all state-step pairs under optimal policies, and $\iota_0 = \log(MSAT/p)$. Since the communication cost of each round is $O(MHS)$, the total communication cost is (2) multiplied by $MHS$.

Compared to the existing worst-case communication rounds that depend on $MSA \log T$ (Zheng et al., 2024; 2025a; Qiao et al., 2022) or $SA \log T$ (Zheng et al., 2025a; Labbi et al., 2024), the first three terms in (2) only logarithmically depend on $1/\Delta_{\min}$ and $\log T$, and the last term removes the dependence on $MSA$ from the $\log T$ term. This improvement is significant since $M$ represents the number of collaborating agents, and $SA$ represents the complexity of the state-action space that is often the bottleneck of RL methods (Jin et al., 2018). Compared to the $SA$-free communication rounds for FRL methods that do not update policies, (2) quantifies the cost of multiple components in online FRL: the first two terms represent the cost for exploration, and the last two terms show the cost of implementing the optimal policy (exploitation). Further technical details are provided in Section 3.3 and Section 5. Our numerical experiments, presented in Appendix B.2, demonstrate that the $\log T$ term in the communication cost is independent of $M$, $S$, and $A$.

When $M = 1$, FedQ-Hoeffding becomes a single-agent algorithm with low global switching cost shown in (2) (Corollary 3.4). It removes the dependence on $SA$ from the $\log T$ term compared to existing model-free methods (Bai et al., 2019; Zhang et al., 2020; Zheng et al., 2025b).

**Technical Novelty and Contributions.** We develop a new theoretical framework for the gap-dependent analysis of online FRL with delayed policy updates. It provides two features simultaneously: controlling the error in the estimated value functions (Lemma 4.1) and estimating the number of visits (Lemma 5.2). The first feature helps prove the gap-dependent regret (1), and the second is key to proving the bound (2) for communication rounds. Here, to overcome the difficulty of estimating visiting numbers, we develop a new technical tool: concentrations on visiting numbers under varying policies. We establish concentration inequalities for visits with the stationary visiting probability of the optimal policies via error recursion on episode steps. This step relies on the logarithmic number of visits with suboptimal actions instead of the algorithm settling on the same policy. It provides better estimations of visiting numbers.

We also establish the following techniques with the tool and nonzero minimum suboptimality gap: (a) Lemma 5.1: Exploring visiting discrepancies between optimal actions and suboptimal actions. This validates the concentrations above. (b) Lemma 5.3: Showing agent-wise simultaneous sufficient increase of visits. This helps remove the linear dependency on $M$ in the last three terms of (2). (c) Lemma 5.4: Showing state-wise simultaneous sufficient increase of visits for states with unique optimal actions. This helps remove the linear dependence on $SA$ from the last term in (2).

To the best of our knowledge, these techniques are new to the literature for online model-free FRL methods. They will be of independent interest in the gap-dependent analysis of other online RL and FRL methods in controlling or estimating the number of visits.

## 2. Background and Problem Formulation

### 2.1. Preliminaries

We begin by introducing the mathematical framework of Markov decision processes. In this paper, we assume that $0/0 = 0$. For any $C \in \mathbb{N}$, we use $[C]$ to denote the set $\{1, 2, \ldots C\}$. We use $\mathbb{I}[x]$ to denote the indicator function, which equals 1 when the event $x$ is true and 0 otherwise.

**Tabular Episodic Markov Decision Process (MDP).** A tabular episodic MDP is denoted as $\mathcal{M} := (\mathcal{S}, \mathcal{A}, H, \mathbb{P}, r)$, where $\mathcal{S}$ is the set of states with $|\mathcal{S}| = S, \mathcal{A}$ is the set of actions with $|\mathcal{A}| = A$, $H$ is the number of steps in each episode, $\mathbb{P} := \{\mathbb{P}_h\}_{h=1}^{H}$ is the transition kernel so that $\mathbb{P}_h(\cdot \mid s, a)$ characterizes the distribution over the next state given the state action pair $(s, a)$ at step $h$, and $r := \{r_h\}_{h=1}^{H}$ is the collection of reward functions. We assume that $r_h(s, a) \in [0, 1]$ is a deterministic function of $(s, a)$, while the results can be easily extended to the case when $r_h$ is random.

In each episode, an initial state $s_1$ is selected arbitrarily by an adversary. Then, at each step $h \in [H]$, an agent observes a state $s_h \in \mathcal{S}$, picks an action $a_h \in \mathcal{A}$, receives the reward $r_h = r_h(s_h, a_h)$ and then transits to the next state $s_{h+1}$. The episode ends when an absorbing state $s_{H+1}$ is reached.

**Policies and Value functions.** A policy $\pi$ is a collection of $H$ functions $\left\{\pi_h : \mathcal{S} \to \Delta^{\mathcal{A}}\right\}_{h \in [H]}$, where $\Delta^{\mathcal{A}}$ is the set of probability distributions over $\mathcal{A}$. A policy is deterministic if for any $s \in \mathcal{S}$, $\pi_h(s)$ concentrates all the probability mass on an action $a \in \mathcal{A}$. In this case, we denote $\pi_h(s) = a$. Let $V_h^{\pi} : \mathcal{S} \to \mathbb{R}$ and $Q_h^{\pi} : \mathcal{S} \times \mathcal{A} \to \mathbb{R}$ denote the state value function and the state-action value function at step $h$ under policy $\pi$. Mathematically, for any $(s, a, h) \in \mathcal{S} \times \mathcal{A} \times [H]$,

$$V_h^{\pi}(s) := \sum_{t=h}^{H} \mathbb{E}_{(s_t, a_t) \sim (\mathbb{P}, \pi)} \left[r_t(s_t, a_t) \mid s_h = s\right]$$

and

$$Q_h^{\pi}(s, a) := r_h(s, a) +$$
$$\sum_{t=h+1}^{H} \mathbb{E}_{(s_t, a_t) \sim (\mathbb{P}, \pi)} \left[r_t(s_t, a_t) \mid (s_h, a_h) = (s, a)\right].$$

Since the state and action spaces and the horizon are all finite, there exists an optimal policy $\pi^{\star}$ that achieves the optimal value $V_h^{\star}(s) = \sup_{\pi} V_h^{\pi}(s) = V_h^{\pi^*}(s)$ for all $(s, h) \in \mathcal{S} \times [H]$ (Azar et al., 2017). The Bellman equation

and the Bellman optimality equation can be expressed as

$$
\begin{cases}
V_h^{\pi}(s) = \mathbb{E}_{a' \sim \pi_h(s)}[Q_h^{\pi}(s, a')] \\
Q_h^{\pi}(s, a) := r_h(s, a) + \mathbb{E}_{s' \sim \mathbb{P}_h(\cdot|s,a)} V_{h+1}^{\pi}(s') \\
V_{H+1}^{\pi}(s) = 0, \forall (s, a, h) \in \mathcal{S} \times \mathcal{A} \times [H],
\end{cases}
$$
$$
\begin{cases}
V_h^{\star}(s) = \max_{a' \in \mathcal{A}} Q_h^{\star}(s, a') \\
Q_h^{\star}(s, a) := r_h(s, a) + \mathbb{E}_{s' \sim \mathbb{P}_h(\cdot|s,a)} V_{h+1}^{*}(s') \\
V_{H+1}^{\star}(s) = 0, \forall (s, a, h) \in \mathcal{S} \times \mathcal{A} \times [H].
\end{cases}
\tag{3}
$$

**Suboptimality Gap.** For any given MDP, we can provide the following formal definition of the suboptimality gap.

**Definition 2.1.** For any $(s, a, h) \in \mathcal{S} \times \mathcal{A} \times [H]$, the suboptimality gap is defined as

$$\Delta_h(s, a) := V_h^{\star}(s) - Q_h^{\star}(s, a).$$

(3) implies that for any $(s, a, h)$, $\Delta_h(s, a) \geq 0$. Then, it is natural to define the minimum gap, which is the minimum non-zero suboptimality gap.

**Definition 2.2.** We define the **minimum gap** as

$$\Delta_{\min} := \inf \left\{\Delta_h(s, a) \mid \Delta_h(s, a) > 0, \ \forall (s, a, h)\right\}.$$

We remark that if

$$\{\Delta_h(s, a) \mid \Delta_h(s, a) > 0, (s, a, h) \in \mathcal{S} \times \mathcal{A} \times [H]\} = \emptyset,$$

then all policies are optimal, leading to a degenerate MDP. Therefore, we assume that the set is nonempty and $\Delta_{\min} > 0$ in the rest of this paper. Definitions 2.1 and 2.2 and the non-degeneration are standard in the literature of gap-dependent analysis (Simchowitz & Jamieson, 2019; Yang et al., 2021; Xu et al., 2020).

**Global Switching Cost.** We provide the following definition for any algorithm with $U > 1$ episodes, which is also used in Bai et al. (2019) and Qiao et al. (2022).

**Definition 2.3.** The global switching cost for any learning algorithm with $U$ episodes is defined as

$$N_{\text{switch}} := \sum_{u=1}^{U-1} \mathbb{I}[\pi^{u+1} \neq \pi^u].$$

Here, $\pi^u$ is the policy implemented in the $u$-th episode.

### 2.2. The Federated RL Framework

We consider an FRL setting with a central server and $M$ agents, each interacting with an independent copy of $\mathcal{M}$. The agents communicate with the server periodically: after receiving local information, the central server aggregates it and broadcasts certain information to the agents to coordinate their exploration.

For agent $m$, let $U_m$ be the number of generated episodes, $\pi^{m,u}$ be the policy in the $u$-th episode of agent $m$, and $x_1^{m,u}$

be the corresponding initial state. The regret of $M$ agents over $\hat{T} = H \sum_{m=1}^{M} U_m$ total steps is

$$\text{Regret}(T) = \sum_{m \in [M]} \sum_{u=1}^{U_m} \left( V_1^{\star}(s_1^{m,u}) - V_1^{\pi^{m,u}}(s_1^{m,u}) \right).$$

Here, $T := \hat{T}/M$ is the average total steps for $M$ agents.

We also define the communication cost of an algorithm as the number of scalars (integers or real numbers) communicated between the server and agents.

## 3. Performance Guarantees

### 3.1. FedQ-Hoeffding Algorithm

In this subsection, we briefly review FedQ-Hoeffding. Details are provided in Algorithm 1 and Algorithm 2 in Appendix C.1. FedQ-Hoeffding proceeds in rounds, indexed by $k \in [K]$. Round $k$ consists of $n^{m,k}$ episodes for agent $m$, where the specific value of $n^{m,k}$ will be determined later.

**Notations.** For the $j$-th ($j \in [n^{m,k}]$) episode for agent $m$ in the $k$-th round, we use $\{(s_h^{k,j,m}, a_h^{k,j,m}, r_h^{k,j,m})\}_{h=1}^{H}$ to denote the corresponding trajectory. Denote $n_h^{m,k}(s,a)$ as the number of times that $(s,a,h)$ has been visited by agent $m$ in round $k$, $n_h^k(s,a) := \sum_{m=1}^{M} n_h^{m,k}(s,a)$ as the total number of visits in round $k$ for all agents, and $N_h^k(s,a)$ as the total number of visits to $(s,a,h)$ among all agents before the start of round $k$. We also use $\{V_h^k : \mathcal{S} \to \mathbb{R}\}_{h=1}^{H}$ and $\{Q_h^k : \mathcal{S} \times \mathcal{A} \to \mathbb{R}\}_{h=1}^{H}$ to denote the global estimates of the state value function and state-action value function at the beginning of round $k$. Before the first round, both estimates are initialized as $H$.

**Coordinated Exploration.** At the beginning of round $k$, the server decides a deterministic policy $\pi^k = \{\pi_h^k\}_{h=1}^{H}$, and then broadcasts it along with $\{N_h^k(s, \pi_h^k(s))\}_{s,h}$ and $\{V_h^k(s)\}_{s,h}$ to agents. Here, $\pi^1$ can be chosen arbitrarily. Then, the agents execute $\pi^k$ and start collecting trajectories. During the exploration in round $k$, every agent $m$ will monitor its number of visits to each $(s,a,h)$. For any agent $m$, at the end of each episode, if any $(s,a,h)$ has been visited by

$$c_h^k(s,a) = \max \left\{ 1, \left\lfloor \frac{N_h^k(s,a)}{MH(H+1)} \right\rfloor \right\} \quad (4)$$

times by agent $m$, the agent will send a signal to the server, which will then abort all agents' exploration. Here, we say that $(s,a,h)$ **satisfies the trigger condition in round** $k$. During the exploration, for all $(s,a,h)$, agent $m$ adaptively calculates $n_h^{m,k}(s,a)$ and the local estimate for the next-step return $v_{h+1}^{m,k}(s,a)$ by

$$\sum_{j=1}^{n^{m,k}} V_{h+1}^k\left(s_{h+1}^{k,j,m}\right) \mathbb{I}\left[\left(s_h^{k,j,m}, a_h^{k,j,m}\right) = (s,a)\right].$$

At the end of round $k$, each agent sends

$$\left\{ r_h\left(s, \pi_h^k(s)\right), n_h^{m,k}\left(s, \pi_h^k(s)\right), v_{h+1}^{m,k}\left(s, \pi_h^k(s)\right) \right\}_{s,h}$$

to the central server for aggregation.

**Updates of Estimated Value Functions.** The central server calculates $n_h^k(s,a)$, $N_h^{k+1}(s,a)$ for all triples. While letting $Q_h^{k+1}(s,a) = Q_h^k(s,a)$ for triples such that $n_h^k(s,a) = 0$, it updates the estimated value functions for each triple with positive $n_h^k(s,a)$ as follows.

**Case 1:** $N_h^k(s,a) < 2MH(H+1) =: i_0$. This case implies that each client can visit each $(s,a)$ pair at step $h$ at most once. Let $Q = Q_h^k(s,a)$. Then the server iteratively update $Q$ using the following assignment:

$$Q \xleftarrow{+} \eta_t \left( r_h + V_{h+1}^{k,t} + b_t - Q \right), \quad t = N_h^k + 1, \ldots, N_h^{k+1}$$

and then assign $Q_h^{k+1}(s,a)$ with $Q$. Here, $r_h, N_h^k, N_h^{k+1}$ are abbreviations for their respective values at $(s,a)$, $\eta_t \in (0,1]$ is the learning rate, $b_t > 0$ is a bonus, and $V_{h+1}^{k,t}$ represents the $(t - N_h^k)$-th nonzero value in $\{v_{h+1}^{m,k}(s,a)\}_{m=1}^{M}$.

**Case 2:** $N_h^k(s,a) \geq i_0$. In this case, the central server calculates the global estimate of the expected return $v_{h+1}^k(s,a) = \sum_{m=1}^{M} v_{h+1}^{m,k}(s,a)/n_h^k(s,a)$ and updates the $Q$-estimate as

$$Q_h^{k+1} = \left(1 - \eta_{s,a}^{h,k}\right) Q_h^k + \eta_{s,a}^{h,k}\left(r_h + v_{h+1}^k\right) + \beta_{s,a,h}^k.$$

Here, $r_h, Q_h^k, Q_h^{k+1}, v_{h+1}^k$ are abbreviations for their respective values at $(s,a)$, $\eta_{s,a}^{h,k} \in (0,1]$ is the learning rate and $\beta_{s,a,h}^k > 0$ represents the bonus.

After updating the estimated $Q$-function, the central server updates the estimated $V$-function and the policy as

$$V_h^{k+1}(s) = \min \left\{ H, \max_{a' \in \mathcal{A}} Q_h^{k+1}(s,a') \right\}$$

and

$$\pi_h^{k+1}(s) = \arg\max_{a' \in \mathcal{A}} Q_h^{k+1}(s,a').$$

Such update implies that FedQ-Hoeffding is an optimism-based method. It then proceeds to round $k+1$.

In FedQ-Hoeffding, agents only send local estimates instead of original trajectories to the central server. This guarantees a low communication cost for each round, which is $O(MHS)$. In addition, the event-triggered termination condition with the threshold (4) limits the number of new visits in each round, with which Zheng et al. (2024) proved the linear regret speedup under worst-case MDPs. Moreover, it guarantees that the number of visits to the triple that satisfies the trigger condition sufficiently increases after this round. This is the key to proving the worst-case logarithmic communication cost in Zheng et al. (2024).

## 3.2. Gap-Dependent Regret

Next, we provide a new gap-dependent regret upper bound for FedQ-Hoeffding algorithm.

**Theorem 3.1.** *Let $\iota_1 = \log(MSAT)$. For FedQ-Hoeffding (Algorithms 1 and 2), $\mathbb{E}\left(\text{Regret}(T)\right)$ can be bounded by*

$$O\left(\frac{H^6 SA\iota_1}{\Delta_{\min}} + M\sqrt{H^7}SA\sqrt{\iota_1} + MH^5 SA\right). \quad (5)$$

The proof is provided in Appendix F. Theorem 3.1 shows that the regret is logarithmic in $T$ for MDPs with positive minimum gap $\Delta_{\min}$. When $T$ is sufficiently large, it is better than the $\sqrt{T}$-type worst-case regrets in the literature.

When $M = 1$, the bound reduces to

$$O\left(\frac{H^6 SA \log(SAT)}{\Delta_{\min}}\right),$$

which matches the result in Yang et al. (2021) for the single-agent counterpart, UCB-Hoeffding algorithm. Therefore, when $T$ is sufficiently large, for the average regret per episode defined as $\text{Regret}(T)/(MT)$, the ratio between FedQ-Hoeffding and UCB-Hoeffding is $\tilde{O}(1/M)$, which serves as our error reduction rate. As a comparison, it is better than the rates under worst-case MDPs for online FRL methods in the literature, which are $\tilde{O}(1/\sqrt{M})$ because of their linear dependency on $\sqrt{MT}$. We will also demonstrate this $\tilde{O}(1/M)$ pattern in the numerical experiments in Appendix B.1.

## 3.3. Gap-Dependent Communication Cost

We first introduce two additional assumptions:

(I) **Full synchronization.** Similar to Zheng et al. (2024), we assume that there is no latency during the communications, and the agents and server are fully synchronized (McMahan et al., 2017). This means $n^{m,k} = n^k$ for each agent $m$.
(II) **Random initialization.** We assume that the initial states $\{s_1^{k,j,m}\}_{k,j,m}$ are randomly generated following some distribution on $\mathcal{S}$, and the generation is not affected by any result in the learning process.

Next, we introduce a new concept: G-MDPs.

**Definition 3.2.** A G-MDP satisfies two conditions:

(a) The stationary visiting probabilities under optimal policies are unique: if both $\pi^{*,1}$ and $\pi^{*,2}$ are optimal policies, then we have $\mathbb{P}\left(s_h = s|\pi^{*,1}\right) = \mathbb{P}\left(s_h = s|\pi^{*,2}\right) =: \mathbb{P}_{s,h}^*$.

(b) Let $\mathcal{A}_h^*(s) = \{a \mid a = \arg\max_{a'} Q_h^*(s, a')\}$. For any $(s, h) \in \mathcal{S} \times [H]$, if $\mathbb{P}_{s,h}^* > 0$, then $|\mathcal{A}_h^*(s)| = 1$, which means that the optimal action is unique.

G-MDPs represent MDPs with generally unique optimal policies. (a) and (b) above characterize the general uniqueness, and an MDP with a unique optimal policy is a G-MDP.

Compared to requiring a unique optimal policy, G-MDPs allow the optimal actions to vary outside the support under optimal policies, i.e., the state-step pairs with $\mathbb{P}_{s,h}^* = 0$.

For a G-MDP, we define $C_{st} = \min\{\mathbb{P}_{s,h}^* \mid s \in \mathcal{S}, h \in [H], \mathbb{P}_{s,h}^* > 0\}$. Thus, $0 < C_{st} \leq 1$ reflects the minimum visiting probability on the support of optimal policies. Next, we provide gap-dependent upper bound for the number communication rounds and communication costs.

**Theorem 3.3.** *For any $p \in (0, 1)$, define $\iota_0 = \log(\frac{MSAT}{p})$. Then under the full synchronization and random initialization assumptions, with probability at least $1 - p$, FedQ-Hoeffding (Algorithm 1 and Algorithm 2) satisfies the following relationship for any given G-MDP:*

$$K \leq O\left(MH^3 SA \log(MH^2\iota_0) + H^3 SA \log\left(\frac{H^5 SA}{\Delta_{\min}^2}\right)\right.$$
$$\left. + H^3 S \log\left(\frac{MH^9 SA\iota_0}{\Delta_{\min}^2 C_{st}}\right) + H^2 \log\left(\frac{T}{HSA}\right)\right). \quad (6)$$

We can get the upper bound of total communication cost by multiplying the upper bound in (6) and $O(MHS)$, the communication cost of each round in FedQ-Hoeffding. We will highlight the key technical tools for proving Theorem 3.3 in Section 5.1, provide a sketch of proof in Section 5.2, and give a complete proof in Appendix G.

Compared to existing worst-case costs that depend on $SA$ (Zheng et al., 2025a; Labbi et al., 2024) or $MSA$ (Zheng et al., 2024; 2025a; Qiao et al., 2022) for $\log T$, (6) is better when $T$ is sufficiently large since the first three terms only logarithmically depend on $1/\Delta_{\min}$ and $\log T$, and the last term that is logarithmic in $T$ removes the dependency on $MSA$. Moreover, (6) highlights the cost for different procedures in FedQ-Hoeffding: the first two terms represent the cost for exploration, and the last two terms show the cost when exploiting the optimal policies. We will provide more theoretical explanations in Section 5. Our numerical experiments in Appendix B.2 also demonstrate that the $\log T$ term in the communication cost is independent of $M$, $S$, and $A$.

Since FedQ-Hoeffding implements a fixed policy in each round, when $M = 1$, the algorithm reduces to a single-agent algorithm with a low global switching cost. The result is formally shown in Corollary 3.4.

**Corollary 3.4.** *For any $p \in (0, 1)$, define $\iota_2 = \log(\frac{SAT}{p})$. Then under the random initialization assumption, for any given G-MDP, with probability at least $1 - p$, the global switching cost for FedQ-Hoeffding algorithm (Algorithm 1 and Algorithm 2 with $M = 1$) can be bounded by*

$$O\left(H^3 SA \log\left(\frac{H^5 SA\iota_2}{\Delta_{\min}^2}\right) + H^3 S \log\left(\frac{1}{C_{st}}\right)\right.$$
$$\left. + H^2 \log\left(\frac{T}{HSA}\right)\right).$$

Given that the switching costs of existing single-agent model-free algorithms depend on $SA$ (Bai et al., 2019; Zhang et al., 2020) or $S$ (Zheng et al., 2025b) for $\log T$[1], our $\log T$-dependency is better by removing the factor $SA$.

At the end of this section, we briefly discuss FedQ-Bernstein, another online FRL algorithm in Zheng et al. (2024). Compared to FedQ-Hoeffding, FedQ-Bernstein uses different bonuses ($b_t$ and $\beta_{s,a,h}^k$) that incorporate variance estimators. Although FedQ-Bernstein achieves a $\sqrt{H}$ factor improvement in worst-case regret while maintaining identical worst-case communication costs (Zheng et al., 2024), our analysis in Appendix F and Appendix G shows both algorithms share the same gap-dependent bounds ((5), (6)). Whether FedQ-Bernstein can achieve tighter gap-dependent regret bounds remains an open question.

## 4. Bounding the Regret with (5)

In this section, we bound the gap-dependent regret by controlling the error in value function estimations. Define $\text{clip}[x \mid y] := x \cdot \mathbb{I}[x \geq y]$. Let $\iota = \log(\frac{2SAHT_1}{\delta})$ where $\delta \in (0,1)$ and $T_1 \leq 2\hat{T} + MHSA$ is an known upper bound of the total steps $\hat{T}$ as defined in (e) of Lemma E.1. We provide Lemma 4.1 to control the total error in the value function estimations $(Q_h^k - Q_h^*)(s,a)$.

**Lemma 4.1.** *For FedQ-Hoeffding (Algorithm 1 and Algorithm 2), for any $\delta \in (0,1)$, with probability at least $1 - \delta$, the following two conclusions hold for any $\epsilon \in (0, H]$:*

$$\sum_{h=1}^{H} \sum_{k,j,m} \mathbb{I}[(Q_h^k - Q_h^*)(s_h^{k,j,m}, a_h^{k,j,m}) \geq \epsilon] \leq C_\epsilon. \quad (7)$$

$$\sum_{h=1}^{H} \sum_{k,j,m} \text{clip}[(Q_h^k - Q_h^*)(s_h^{k,j,m}, a_h^{k,j,m}) \mid \epsilon] \leq \epsilon C_\epsilon. \quad (8)$$

*Here*

$$C_\epsilon = c_0 \left( \frac{H^6 SA\iota}{\epsilon^2} + \frac{MH^5 SA + M\sqrt{H^7}SA\sqrt{\iota}}{\epsilon} \right),$$

*where $c_0 > 0$ is a sufficiently large constant.*

The proof of Lemma 4.1 is in Appendix F.2. Both bounds depend on $\log T$ when $\epsilon$ is fixed. Compared to the methods for single-agents algorithms (see, e.g., Yang et al. (2021)), Lemma 4.1 also accommodates the delayed policy updates, and its dependency on $M$ reflects the cost of collaborating multiple agents. We will let $\epsilon = \Delta_{\min}$ later.

---

[1]In the literature, these bounds are for local switching cost that counts the state-step pairs where the policy switches. The local switching cost is greater than or equal to the global switching cost, but these works didn't find tighter bounds for the global switching cost. We refer readers to Bai et al. (2019) for more information.

Next, Lemma 4.2 characterizes the relationship between the expected regret and the total error $(Q_h^k - Q_h^*)(s,a)$.

**Lemma 4.2.** *For FedQ-Hoeffding (Algorithm 1 and Algorithm 2), the expected regret $\mathbb{E}(\text{Regret}(T))$ is bounded by*

$$\mathbb{E}\left[ \sum_{h=1}^{H} \sum_{k,j,m} \text{clip}[(Q_h^k - Q_h^*)(s_h^{k,j,m}, a_h^{k,j,m}) \mid \Delta_{\min}] \right].$$

The proof of Lemma 4.2 is provided in Appendix F.3. By combining Equation (8) in Lemma 4.1 and Lemma 4.2 and using the definition of expectation, we can bound the expected regret and finish the proof of Theorem 3.1. Further details can be found in Appendix F.4.

## 5. Bounding the Communication Cost with (6)

### 5.1. Bounding the Number of Visits

In this subsection, we introduce the new technical tool for estimating visiting numbers. We first provide Lemma 5.1 that quantifies the frequency and the probability of implementing non-optimal actions.

**Lemma 5.1.** *For any $\delta \in (0,1)$ and any given deterministic optimal policy $\pi^*$, with probability at least $1 - 3\delta$, we have*

$$\sum_{h=1}^{H} \sum_{k,j,m} \mathbb{I}\left[ a_h^{k,j,m} \notin \mathcal{A}_h^*(s_h^{k,j,m}) \right] \leq C_{\min} \quad (9)$$

$$\sum_{k,j,m} \mathbb{P}\left( a_h^{k,j,m} \neq \pi_h^*(s_h^{k,j,m}) \mid \pi^k \right) \leq 4C_{\min}, \forall h \in [H]. \quad (10)$$

*Here $C_{\min}$ equals $C_\epsilon$ in Lemma 4.1 with $\epsilon = \Delta_{\min}$.*

For each $a_h^{k,j,m} \notin \mathcal{A}_h^*(s_h^{k,j,m})$, the optimism of FedQ-Hoeffding ensures that

$$\left(Q_h^k - Q_h^*\right)\left(s_h^{k,j,m}, a_h^{k,j,m}\right) \geq \Delta_{\min}$$

with high probability. Therefore, by taking $\epsilon = \Delta_{\min}$ in (7), we can bound

$$\mathbb{I}\left[ a_h^{k,j,m} \notin \mathcal{A}_h^*(s_h^{k,j,m}) \right]$$

in (9) and its conditional expectation in (10). See Appendix G.2 for details of the proof.

Since $C_{\min}$ scales logarithmically with $T$, (9) shows that the frequency of non-optimal action selections becomes negligible compared to $T$ asymptotically. This means that most states in the learning process are generated under optimal actions and reveals the visiting discrepancy between optimal and non-optimal actions in the gap-dependent analysis.

Such discrepancy helps us quantify the communication cost paid for exploring non-optimal actions. The threshold of

the synchronization condition (4) implies that the number of visits to the triple $(s, a, h)$ that satisfies the trigger condition increases by at least $1/(2MH(H+1))$ times. Consequently, the logarithmic upper bound for non-optimal visits, as provided in (9), implies a $\log\log(T)$-type communication cost for exploration, which is reflected in the first two terms of (6). These two terms depend on $SA$ because FedQ-Hoeffding only ensures a sufficient increase in the number of visits for one triple in a round. We remove the dependency on $M$ from the second term by proving agent-wise simultaneous sufficient increase of visits (Lemma 5.3), leveraging the stationary visiting probability under their common policy in a round.

Next, we bound the number of visits to the optimal visits. For any $k' \in [K]$, let $R_{k'} = \sum_{k=1}^{k'} \sum_{j,m} 1$ be the number of episodes in the first $k'$ rounds. Lemma 5.2 quantifies the difference between the number of visits to any $(s, a, h)$ with $a \in \mathcal{A}_h^*(s)$ in the first $k'$ rounds and the expected number of visits $R_{k'} \mathbb{P}_{s,h}^*$ under the optimal policy.

**Lemma 5.2.** *For any $\delta \in (0, 1)$, with probability at least $1 - 5\delta$, the following conclusion holds simultaneously for any $(s, h, k') \in \mathcal{S} \times [H] \times [K]$:*

$$\left| \sum_{k=1}^{k'} \sum_{j,m} \mathbb{I}\left[ s_h^{k,j,m} = s, a_h^{k,j,m} \in \mathcal{A}_h^*(s) \right] - R_{k'} \mathbb{P}_{s,h}^* \right|$$
$$\leq 5\sqrt{R_{k'} \mathbb{P}_{s,h}^* \iota} + 32HC_{\min}.$$

Lemma 5.2 establishes that the average number of visits to $(s, a, h)$ with $a \in \mathcal{A}_h^*(s)$ per episode will converge to the stationary visiting probability $\mathbb{P}_{s,h}^*$ under the optimal policies. Furthermore, it implies that for any $(s, a, h, k)$ such that $\mathbb{P}_{s,h}^* > 0$ and $a = \pi_h^*(s)$,

$$N_h^{k+1}(s,a) \in \left[ R_k \mathbb{P}_{s,h}^* - 5\sqrt{R_k \mathbb{P}_{s,h}^* \iota} - 32HC_{\min}, \right.$$
$$\left. R_k \mathbb{P}_{s,h}^* + 5\sqrt{R_k \mathbb{P}_{s,h}^* \iota} + 32HC_{\min} \right].$$

Therefore, when $N_h^k(s,a)$ is sufficiently large (ensuring that both $R_{k-1} \mathbb{P}_{s,h}^*$ and $R_k \mathbb{P}_{s,h}^*$ are sufficiently large), the ratio $N_h^{k+1}(s,a)/N_h^k(s,a)$ approximates $R_k/R_{k-1}$. Since $R_k/R_{k-1}$ is independent of $(s, a, h)$, the number of visits to each optimal $(s, a, h)$ ($\mathbb{P}_{s,h}^* > 0$ and $a$ is the optimal action) increases at similar speed. This explains why the communication cost for exploiting the unique optimal action after sufficient visits (the last term of (6)) does not depend on the factor $SA$. The dependence on $M$ is also removed due to the agent-wise simultaneous sufficient increase. Additionally, we remark that the third term of (6) accounts for cost with insufficient visit counts.

Finally, we provide the intuition for the proof of Lemma 5.2. Standard concentration inequalities typically relate the number of visits of $(s, h)$ to the policy-dependent probability

$\mathbb{P}(s_h = s \mid \pi^k)$. However, the varying policies employed by FedQ-Hoeffding across different rounds prevent direct alignment between the executed policy $\pi^k$ and the optimal policy $\pi^*$. To overcome this challenge, our proof establishes a relationship between $\mathbb{P}(s_h = s \mid \pi^k)$ and the optimal stationary visiting probabilities $\mathbb{P}_{s,h}^*$ through error recursion over the step $h$. This analysis exploits the discrepancy in visit counts between optimal and non-optimal actions, which is a distinctive feature enabled by the gap-dependent structure. Especially, we prove that for any $h' \in [H]$,

$$\sum_s \left| \mathbb{P}\left( s_{h'}^{k,j,m} = s \mid \pi^k \right) - \mathbb{P}_{s,h'}^* \right|$$
$$\leq 2 \sum_{h=1}^{h'-1} \mathbb{P}\left( a_h^{k,j,m} \neq \pi_h^*(s_h^{k,j,m}) \mid \pi^k \right),$$

which is further bounded by (10) in Lemma 5.1 and helps complete the proof of Lemma 5.2. See Appendix G.3 for more details of the proof.

### 5.2. Proof Sketch of Theorem 3.3

With the tools introduced in Section 5.1, we outline the key steps in proving the gap-dependent bound (6) for the number of communication rounds.

Let $\iota' = \log\left( \frac{2MSAHT_1}{\delta} \right)$, $i_1 = 200MH(H+1)\iota'$, $i_2 = 6500H^3C_{\min}/C_{st}$ and $\tilde{C} = 1/(H(H+1))$. In this subsection, for any $(s, h) \in \mathcal{S} \times [H]$ such that $\mathbb{P}_{s,h}^* > 0$, we use $\pi_h^*(s)$ to denote its unique optimal action.

Lemma 5.3 shows agent-wise simultaneous sufficient increase of visits for the triple $(s, a, h)$ that satisfies the trigger condition in round $k$ when $N_h^k(s,a) > i_1$.

**Lemma 5.3.** *For any $\delta \in (0, 1)$, with probability at least $1 - \delta$,*

$$N_h^{k+1}(s,a) \geq \left( 1 + \tilde{C}/3 \right) N_h^k(s,a)$$

*holds simultaneously for any $(s, a, h, k) \in \mathcal{S} \times \mathcal{A} \times [H] \times [K]$ such that $N_h^k(s,a) > i_1$ and the triple $(s, a, h)$ satisfies the trigger condition (4) in round $k$.*

The proof of Lemma 5.3 can be found in Appendix G.4.

Lemma 5.4 shows the state-wise simultaneous sufficient increase of visits for states with unique optimal actions, which is proved based on Lemma 5.2.

**Lemma 5.4.** *For any $\delta \in (0, 1)$, with probability at least $1 - 5\delta$, the following events hold simultaneously for any $k \in [K]$: If there exists $(s_0, a_0, h_0) \in \mathcal{S} \times \mathcal{A} \times [H]$, such that it satisfies the trigger condition (4) in round $k$ and $N_{h_0}^k(s_0, a_0) > i_1 + i_2$, then $a_0 \in \mathcal{A}_{h_0}^*(s_0)$.*

*Furthermore, if the state-action-step triple $(s_0, a_0, h_0)$ also satisfies that $\mathbb{P}_{s_0, h_0}^* > 0$, then for any $(s', h') \in \mathcal{S} \times [H]$*

*such that $\mathbb{P}^*_{s',h'} > 0$, we have*

$$N_{h'}^{k+1}\left(s', \pi_{h'}^*(s')\right) \geq \left(1 + \tilde{C}/6\right) N_{h'}^k\left(s', \pi_{h'}^*(s')\right)$$

The complete proof of Lemma 5.4 is in Appendix G.5.

We now analyze the number of rounds in which the trigger condition is satisfied, categorized according to the four cases corresponding to the terms in (6). A detailed discussion can be found in Appendix G.6.

**Type-I Trigger**: It occurs when a triple $(s, a, h)$ satisfies the trigger condition in round $k$ with $N_h^k(s, a) \leq i_1$.

For each time the trigger condition is met by a triple $(s, a, h)$, the number of visits to it increases by at least $\tilde{C}/2M$ times. Therefore, the maximum number of Type-I triggers for any triple $(s, a, h)$ is

$$O\left(\frac{\log(i_1)}{\log(1 + \tilde{C}/(2M))}\right) = O\left(MH^2 \log(i_1)\right).$$

Thus, the number of rounds with Type-I triggers is no more than $O\left(MH^3SA \log(i_1)\right)$.

**Type-II Trigger**: It occurs when a triple $(s, a, h)$ satisfies the trigger condition in round $k$ with $i_1 < N_h^k(s, a) \leq i_1 + i_2$ and either $a \notin \mathcal{A}_h^*(s)$ or $a \in \mathcal{A}_h^*(s)$ and $\mathbb{P}^*_{s,h} = 0$.

By Lemma 5.3, which establishes the agent-wise simultaneous sufficient increase, the number of visits to the triple $(s, a, h)$ increases by at least $\tilde{C}/3$ times each time the trigger condition is satisfied.

Furthermore, as shown in (9) of Lemma 5.1 and Lemma 5.2 with $\mathbb{P}^*_{s,h} = 0$, for state-action-step triple $(s, a, h)$ where $a \notin \mathcal{A}_h^*(s)$ or $a \in \mathcal{A}_h^*(s)$ and $\mathbb{P}^*_{s,h} = 0$, the total number of visits is bounded by $32HC_{\min}$ with high probability. Consequently, the maximum number of Type-II triggers for any such triple is

$$O\left(\frac{\log(32HC_{\min}/i_1)}{\log(1 + \tilde{C}/3)}\right) \leq O\left(H^2 \log\left(\frac{H^5SA}{\Delta_{\min}^2}\right)\right).$$

Then the upper bound for the number of rounds with Type-II triggers is

$$O\left(H^3SA \log\left(\frac{H^5SA}{\Delta_{\min}^2}\right)\right).$$

**Type-III Trigger**: It occurs when a triple $(s, a, h)$ satisfies the trigger condition in round $k$ with $i_1 < N_h^k(s, a) \leq i_1 + i_2$, $a \in \mathcal{A}_h^*(s)$ and $\mathbb{P}^*_{s,h} > 0$.

For any triple $(s, a, h)$ that satisfies Type-III triggers, condition (b) of Definition 3.2 ensures that $a$ is the unique optimal action $\pi_h^*(s)$. Therefore, at most $HS$ different triples can satisfy Type-III trigger conditions.

When such a trigger occurs, we have $N_h^k(s, a) > i_1$, and Lemma 5.3 implies that the number of visits to the triple $(s, a, h)$ increases by at least $\tilde{C}/3$ times. Therefore, the maximum number of Type-III triggers for any such triple is

$$O\left(\frac{\log(i_2/i_1 + 1)}{\log(1 + \tilde{C}/3)}\right) \leq O\left(H^2 \log(i_2)\right).$$

Then the number of rounds with Type-III triggers is no more than $O(H^3S \log(i_2))$.

**Type-IV Trigger**: It occurs when a triple $(s, a, h)$ satisfies the trigger condition in round $k$ with $N_h^k(s, a) > i_1 + i_2$.

In this case, whenever the trigger condition is satisfied by $(s, a, h)$ in round $k$, we have $N_h^k(s, a) > i_2 > 32HC_{\min}$ and $a \in \mathcal{A}_h^*(s)$ by Lemma 5.4. Furthermore, since Lemma 5.2 establish an upper bound of $32HC_{\min}$ on the number of visits to triples $(s', a', h')$ where $\mathbb{P}^*_{s',h'} = 0$, we can conclude that with high probability, $\mathbb{P}^*_{s,h} > 0$ and $a = \pi_h^\star(s)$ holds.

By Lemma 5.4, for any state-step pair $(s', h) \in \mathcal{S} \times [H]$ such that $\mathbb{P}^*_{s',h'} > 0$, the number of visits to $(s', \pi_{h'}^*(s'), h')$ simultaneously increases by at least $\tilde{C}/6$ times. Therefore, the maximum number of rounds with Type-IV triggers is

$$O\left(\frac{\log(\hat{T}/(i_1 + i_2))}{\log(1 + \tilde{C}/6)}\right) \leq O\left(H^2 \log\left(\frac{T}{HSA}\right)\right).$$

By aggregating the bounds on the number of communication rounds across all four cases, we derive the gap-dependent upper bound presented in (6).

# 6. Conclusion

In this paper, we establish the first gap-dependent bounds on regret and communication cost for online federated $Q$-Learning in tabular episodic finite-horizon MDPs, addressing two important open questions in the literature. While existing FRL methods focus on worst-case MDPs, we show that when MDPs exhibit benign structures, such as a strictly positive suboptimality gap, the worst-case bounds can be significantly improved. Specifically, we prove that both FedQ-Hoeffding and FedQ-Bernstein can achieve logarithmic regret. Additionally, we derive a gap-dependent communication cost upper bound that disentangles exploration and exploitation, with the $\log T$ term in the bound being independent of $M$, $S$, and $A$. This makes our work the first result in the online FRL literature to achieve such a low communication cost. When $M = 1$, our gap-dependent communication cost upper bound also yields a tighter global switching cost upper bound, removing the dependence on $SA$ from the $\log T$ term.

## Acknowledgment

The work of H. Zhang, Z. Zheng, and L. Xue was supported by the U.S. National Science Foundation under the grants DMS-1953189 and CCF-2007823 and by the U.S. National Institutes of Health under the grant 1R01GM152812.

## Impact Statement

This work significantly advances federated reinforcement learning (FRL) by improving regret and communication efficiency. Federated reinforcement learning has privacy-preserving properties by design, as it enables agents to learn collaboratively without sharing raw data. This feature is instrumental in various areas, such as healthcare, finance, and education, where sensitive information must be protected.

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

**Organization of the appendix.** In the appendix, Appendix A reviews related works. Appendix B presents the results of our numerical experiments, demonstrating a $\log T$-type regret and showing that the $\log T$ term of the communication cost is independent of $M$, $S$, and $A$. Appendix C provides algorithmic details for both the FedQ-Hoeffding and FedQ-Bernstein algorithms. Appendix D and Appendix E include some useful lemmas. Appendix F contains the proof of the gap-dependent regret bound (Theorem 3.1). Appendix G presents the proof of the gap-dependent communication cost bound (Theorem 3.3).

# A. Related Work

**online RL for Single Agent RL with Worst-Case Regret.** There are mainly two types of algorithms for reinforcement learning: model-based and model-free learning. Model-based algorithms learn a model from past experience and make decisions based on this model, while model-free algorithms only maintain a group of value functions and take the induced optimal actions. Due to these differences, model-free algorithms are usually more space-efficient and time-efficient compared to model-based algorithms. However, model-based algorithms may achieve better learning performance by leveraging the learned model.

Next, we discuss the literature on model-based and model-free algorithms for finite-horizon tabular MDPs with worst-case regret. Auer et al. (2008), Agrawal & Jia (2017), Azar et al. (2017), Kakade et al. (2018), Agarwal et al. (2020), Dann et al. (2019), Zanette & Brunskill (2019), Zhang et al. (2021), Zhou et al. (2023) and Zhang et al. (2024b) worked on model-based algorithms. Notably, Zhang et al. (2024b) provided an algorithm that achieves a regret of $\tilde{O}(\min\{\sqrt{SAH^2T}, T\})$, which matches the information lower bound. Jin et al. (2018), Yang et al. (2021), Zhang et al. (2020), Li et al. (2021) and Ménard et al. (2021) work on model-free algorithms. The latter three works achieved the minimax regret of $\tilde{O}(\sqrt{SAH^2T})$.

**Suboptimality Gap.** When there is a strictly positive suboptimality gap, it is possible to achieve logarithmic regret bounds. In RL, earlier work obtained asymptotic logarithmic regret bounds (Auer & Ortner, 2007; Tewari & Bartlett, 2008). Recently, non-asymptotic logarithmic regret bounds were obtained (Jaksch et al., 2010; Ok et al., 2018; Simchowitz & Jamieson, 2019; He et al., 2021). Specifically, Jaksch et al. (2010) developed a model-based algorithm, and their bound depends on the policy gap instead of the action gap studied in this paper. Ok et al. (2018) derived problem-specific logarithmic type lower bounds for both structured and unstructured MDPs. Simchowitz & Jamieson (2019) extended the model-based algorithm in Zanette & Brunskill (2019) and obtained logarithmic regret bounds. Logarithmic regret bounds are also derived in linear function approximation settings He et al. (2021). Additionally, Nguyen-Tang et al. (2023) provides a gap-dependent regret bounds for offline RL with linear funciton approximation.

Specifically, for model free algorithm, Yang et al. (2021) showed that the optimistic $Q$-learning algorithm by Jin et al. (2018) enjoyed a logarithmic regret $O(\frac{H^6SAT}{\Delta_{\min}})$, which was subsequently refined by Xu et al. (2021). In their work, Xu et al. (2021) introduced the Adaptive Multi-step Bootstrap (AMB) algorithm. Zheng et al. (2025b) further improved the logarithmic regret bound by leveraging the analysis of the UCB-Advantage algorithm (Zhang et al., 2020) and Q-EarlySettled-Advantage algorithm (Li et al., 2021).

There are also some other works focusing on gap-dependent sample complexity bounds (Jonsson et al., 2020; Marjani & Proutiere, 2020; Al Marjani et al., 2021; Tirinzoni et al., 2022; Wagenmaker et al., 2022b; Wagenmaker & Jamieson, 2022; Wang et al., 2022; Tirinzoni et al., 2023).

**RL with Low Switching Cost and Batched RL.** Research in RL with low-switching cost aims to minimize the number of policy switches while maintaining comparable regret bounds to fully adaptive counterparts, and it can be applied to federated RL. In batched RL (Perchet et al., 2016; Gao et al., 2019), the agent sets the number of batches and the length of each batch upfront, implementing an unchanged policy in a batch and aiming for fewer batches and lower regret. Bai et al. (2019) first introduced the problem of RL with low-switching cost and proposed a $Q$-learning algorithm with lazy updates, achieving $\tilde{O}(SAH^3\log T)$ switching cost. This work was advanced by Zhang et al. (2020), which improved the regret upper bound and the switching cost. Additionally, Wang et al. (2021) studied RL under the adaptivity constraint. Recently, Qiao et al. (2022) proposed a model-based algorithm with $\tilde{O}(\log\log T)$ switching cost. Zhang et al. (2022b) proposed a batched RL algorithm that is well-suited for the federated setting.

**Multi-Agent RL (MARL) with Event-Triggered Communications.** We review a few recent works for online MARL with linear function approximations. Dubey & Pentland (2021) introduced Coop-LSVI for cooperative MARL. Min et al. (2023) proposed an asynchronous version of LSVI-UCB that originates from Jin et al. (2020), matching the same regret bound with improved communication complexity compared to Dubey & Pentland (2021). Hsu et al. (2024) developed two algorithms that incorporate randomized exploration, achieving the same regret and communication complexity as Min et al. (2023).

Dubey & Pentland (2021), Min et al. (2023) and Hsu et al. (2024) employed event-triggered communication conditions based on determinants of certain quantities. Different from our federated algorithm, during the synchronization in Dubey & Pentland (2021) and Min et al. (2023), local agents share original rewards or trajectories with the server. On the other hand, Hsu et al. (2024) reduces communication cost by sharing compressed statistics in the non-tabular setting with linear function approximation.

**Federated and Distributed RL**. Existing literature on federated and distributed RL algorithms highlights various aspects. For value-based algorithms, Guo & Brunskill (2015), Zheng et al. (2024), and Woo et al. (2023) focused on linear speed up. Agarwal et al. (2021) proposed a parallel RL algorithm with low communication cost. Woo et al. (2023) and Woo et al. (2024) discussed the improved covering power of heterogeneity. Wu et al. (2021) and Chen et al. (2023) worked on robustness. Particularly, Chen et al. (2023) proposed algorithms in both offline and online settings, obtaining near-optimal sample complexities and achieving superior robustness guarantees. In addition, several works have investigated value-based algorithms such as $Q$-learning in different settings, including Beikmohammadi et al. (2024), Jin et al. (2022), Khodadadian et al. (2022), Fan et al. (2023), Woo et al. (2023), Woo et al. (2024); Anwar & Raychowdhury (2021) Zhao et al. (2023), He et al. (2022), Yang et al. (2024) and Zhang et al. (2024a). The convergence of decentralized temporal difference algorithms has been analyzed by Doan et al. (2019), Doan et al. (2021), Chen et al. (2021b), Sun et al. (2020), Wai (2020), Wang et al. (2020), Zeng et al. (2021), and Liu & Olshevsky (2023).

Some other works focus on policy gradient-based algorithms. Communication-efficient policy gradient algorithms have been studied by Chen et al. (2021a) and Fan et al. (2021). Lan et al. (2023) further reduces the communication complexity and also demonstrates a linear speedup in the synchronous setting. Optimal sample complexity for global convergence in federated RL, even in the presence of adversaries, is studied in Ganesh et al. (2024). Lan et al. (2024) proposes an algorithm to address the challenge of lagged policies in asynchronous settings.

The convergence of distributed actor-critic algorithms has been analyzed by Shen et al. (2023) and Chen et al. (2022). Federated actor-learner architectures have been explored by Assran et al. (2019), Espeholt et al. (2018) and Mnih et al. (2016). Distributed inverse reinforcement learning has been examined by Banerjee et al. (2021), Gong et al. (2023), and Liu & Zhu (2022; 2023; 2024; 2025). Personalized federated learning has been discussed in (Hanzely & Richtárik, 2020; Li et al., 2020; Smith et al., 2017; Yu et al., 2024)

## B. Numerical Experiments

In this section, we conduct experiments[2]. All the experiments are conducted in a synthetic environment to demonstrate the $\log T$-type regret and reduced communication cost bound with the coefficient of the main term $O(\log T)$ being independent of $M, S, A$ in FedQ-Hoeffding algorithm (Zheng et al., 2024). We follow Zheng et al. (2024) and generate a synthetic environment to evaluate the proposed algorithms on a tabular episodic MDP. After setting $H, S, A$, the reward $r_h(s, a)$ for each $(s, a, h)$ is generated independently and uniformly at random from $[0, 1]$. $\mathbb{P}_h(\cdot \mid s, a)$ is generated on the $S$-dimensional simplex independently and uniformly at random for $(s, a, h)$. We also set the constant $c$ in the bonus term $b_t$ to be 2 and $\iota = 1$. We will first demonstrate the $\log T$-type regret of FedQ-Hoeffding algorithm.

### B.1. Logarithmic Regret and Speedup

In this section, we show that the regret for any given MDP follows a $\log T$ pattern. We consider two different values for the triple $(H, S, A)$: $(2, 2, 2)$ and $(5, 3, 2)$. For FedQ-Hoeffding algorithm, we set the agent number $M = 10$ and generate $T/H = 10^7$ episodes for each agent, resulting in a total of $10^8$ episodes. Additionally, to show the linear speedup effect, we conduct experiments with its single-agent version, the UCB-Hoeffding algorithm (Jin et al., 2018), where all the conditions except $M = 1$ remain the same. To show error bars, we also collect 10 sample paths for each algorithm under the same MDP environment.

The regret results are shown in Figure 1 and Figure 2. Both figures display performance metrics through two visualization panels: the left showing raw regret $\text{Regret}(T)$ versus the normalized horizon $T/H$, and the right plotting adjusted regret $\text{Regret}(T)/\log(T/H+1)$ versus $T/H$. All solid lines represent median values across 10 trials, with shaded areas indicating the 10th-90th percentile ranges. Specifically: the yellow lines show the regret results of FedQ-Hoeffding, the red lines represent the regret results UCB-Hoeffding, and the blue line displays the FedQ-Hoeffding regret scaled by $1/\sqrt{M}$ to

---

[2]All the experiments are run on a server with Intel Xeon E5-2650v4 (2.2GHz) and 100 cores. Each replication is limited to a single core and 50GB RAM. The code for the numerical experiments is included in the supplementary materials along with the submission.

demonstrate its regret error reduction speedup pattern.

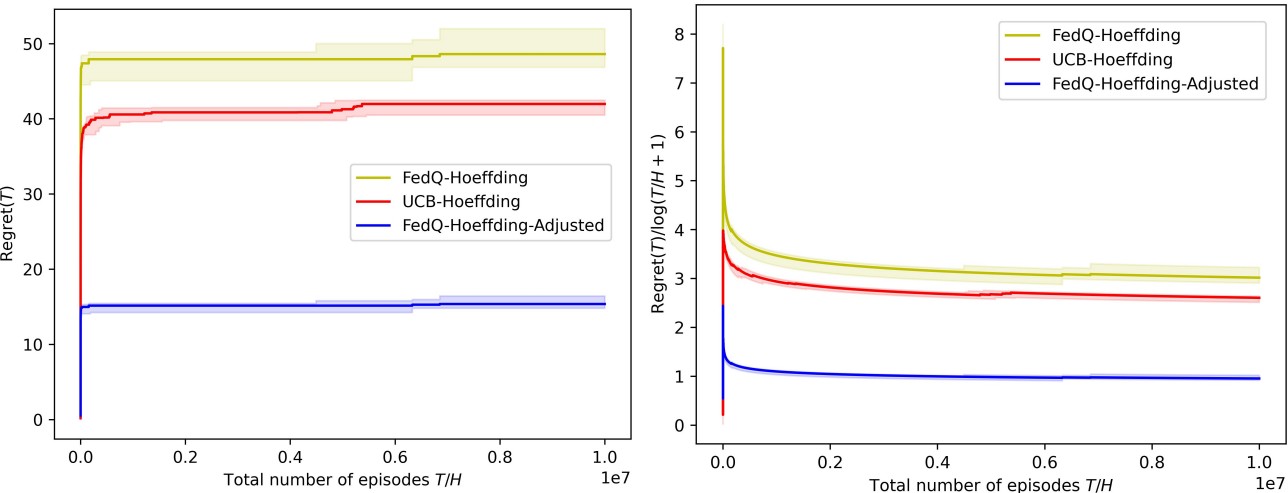

*Figure 1.* Regret results for $H = 2$, $S = 2$, and $A = 2$. The left panel directly shows the plot of $\text{Regret}(T)$ versus $T/H$, while the right panel illustrates the relationship between $\text{Regret}(T)/\log(T/H + 1)$ and $T/H$. In both plots, the yellow line represents the regret results of the FedQ-Hoeffding algorithm, while the red line represents the results of the UCB-Hoeffding algorithm. The blue line in each plot denotes the adjusted regret of the FedQ-Hoeffding algorithm, which is obtained by dividing the regret results of the yellow line by $\sqrt{M}$.

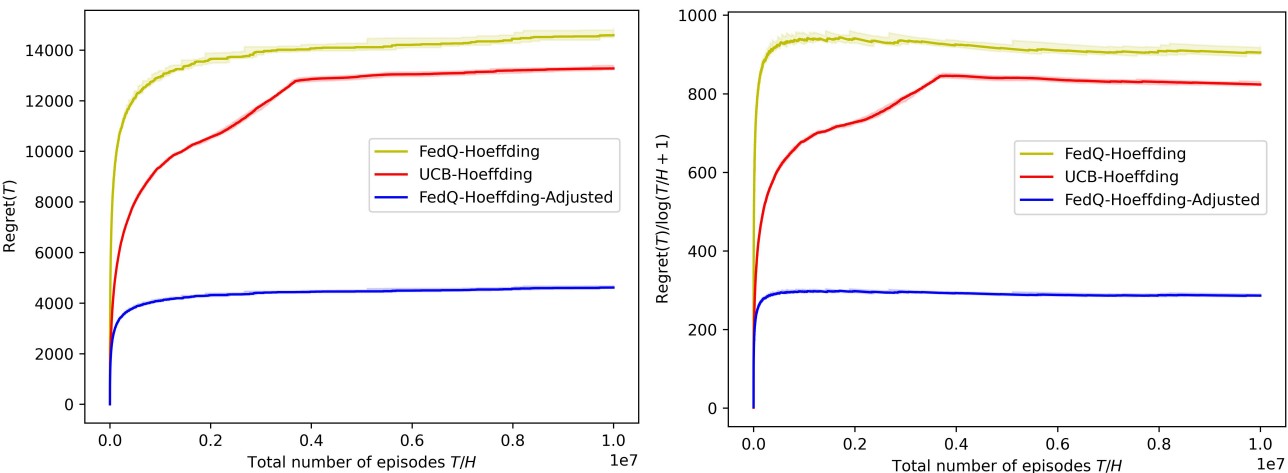

*Figure 2.* Regret results for $H = 5$, $S = 3$, and $A = 2$. The left panel directly shows the plot of $\text{Regret}(T)$ versus $T/H$, while the right panel illustrates the relationship between $\text{Regret}(T)/\log(T/H + 1)$ and $T/H$. In both plots, the yellow line represents the regret results of the FedQ-Hoeffding algorithm, while the red line represents the results of the UCB-Hoeffding algorithm. The blue line in each plot denotes the adjusted regret of the FedQ-Hoeffding algorithm, which is obtained by dividing the regret results of the yellow line by $\sqrt{M}$.

From the two groups of plots, we observe that the two yellow lines in the plots on the right side of Figure 1 and Figure 2 tend to approach horizontal lines as $T/H$ becomes sufficiently large. Since the y-axis represents $\text{Regret}(T)/\log(T/H + 1)$ in these two plots, we can conclude that the regret of the FedQ-Hoeffding algorithm follows a $\log T$-type pattern for any given MDP, rather than the $\sqrt{MT}$ pattern shown in the Theorem 4.1 of Zheng et al. (2024). This is consistent with the logarithmic regret result presented in Theorem 3.1. Furthermore, as $T/H$ becomes sufficiently large, we observe that the adjusted regret of FedQ-Hoeffding (represented by the blue lines) for both groups of $(H, S, A)$ is significantly lower than the corresponding regret of the single-agent version, UCB-Hoeffding (represented by the red lines). This further supports the conclusion that the regret of FedQ-Hoeffding does not follow a $\sqrt{MT}$ pattern, or else the blue lines and the red lines would be close to each other. Finally, as $T/H$ grows larger, we notice that the yellow lines and the red lines become close, confirming that the regret of FedQ-Hoeffding approaches that of UCB-Hoeffding as $T$ becomes sufficiently large. This also supports the error reduction rate $\tilde{O}(1/M)$ for the gap-dependent regret.

## B.2. Dependency of Communication Cost on $M$, $S$, and $A$

In this section, we will demonstrate that the coefficient of the $\log T$ term in the communication cost is independent of $M$, $S$ and $A$. To eliminate the influence of terms with lower orders of $\log T$, such as $\log(\log T)$ and $\sqrt{\log T}$ in Theorem 3.3, we will focus exclusively on the communication cost for sufficiently large values of $T$.

### B.2.1. DEPENDENCY ON $M$

To explore the dependency of communication cost on $M$, we set $(H, S, A) = (2, 2, 2)$ and let $M$ take values in $\{2, 4, 6, 8\}$. We generate $10^7$ episodes for each agent and only consider the communication cost after $5 \times 10^5$ episodes. The Figure 3 shows the communication cost results for each $M$ after $5 \times 10^5$ episodes.

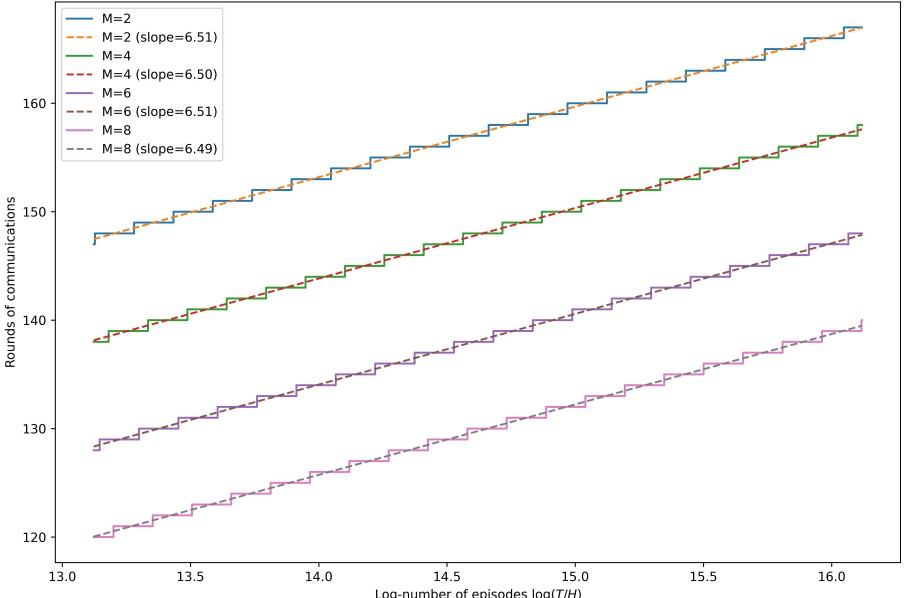

*Figure 3.* Number of communication rounds vs Log-number of Episodes for different $M$ Values with $H = 2$, $S = 2$ and $A = 2$. Each solid line represents the number of communication rounds for each value of $M \in \{2, 4, 6, 8\}$ after $5 \times 10^5$ episodes, while the dashed line represents the fitted line for each $M$.

In Figure 3, each solid line represents the communication cost for each value of $M \in \{2, 4, 6, 8\}$ after $5 \times 10^5$ episodes, while the dashed line represents the corresponding fitted line. Since the x-axis represents the log-number of episodes, $\log(T/H)$, the slope of the fitted line is very close to the coefficient of the $\log T$-term in the communication cost when $\log T$ is sufficiently large. We observe that the slopes of these fitted lines are very similar, which indicates that for any given MDP, the coefficient of the $\log T$-term in the communication cost is independent of $M$. If the coefficient were linearly dependent on $M$, as shown in Zheng et al. (2024), then for $M = 8$, the slope of the fitted line should be nearly four times the value of the slope of the fitted line for $M = 2$.

### B.2.2. DEPENDENCY ON $S$

To explore the dependency of communication cost on $S$, we set $(H, A, M) = (2, 2, 2)$ and let $S$ take values in $\{2, 4, 6, 8\}$. We generate $10^7$ episodes for each agent and only consider the communication cost after $5 \times 10^5$ episodes. The Figure 4 shows the communication cost results for each $S$ after $5 \times 10^5$ episodes.

In Figure 4, each solid line represents the communication cost for each value of $S \in \{2, 4, 6, 8\}$ after $5 \times 10^5$ episodes, while the dashed line represents the corresponding fitted line. Since the x-axis represents the log-number of episodes, $\log(T/H)$, the slope of the fitted line is very close to the coefficient of the $\log T$-term in the communication cost when $\log T$ is sufficiently large. We observe that the slopes of these fitted lines are very similar, which indicates that for any given MDP, the coefficient of the $\log T$-term in the communication cost is independent of $S$.

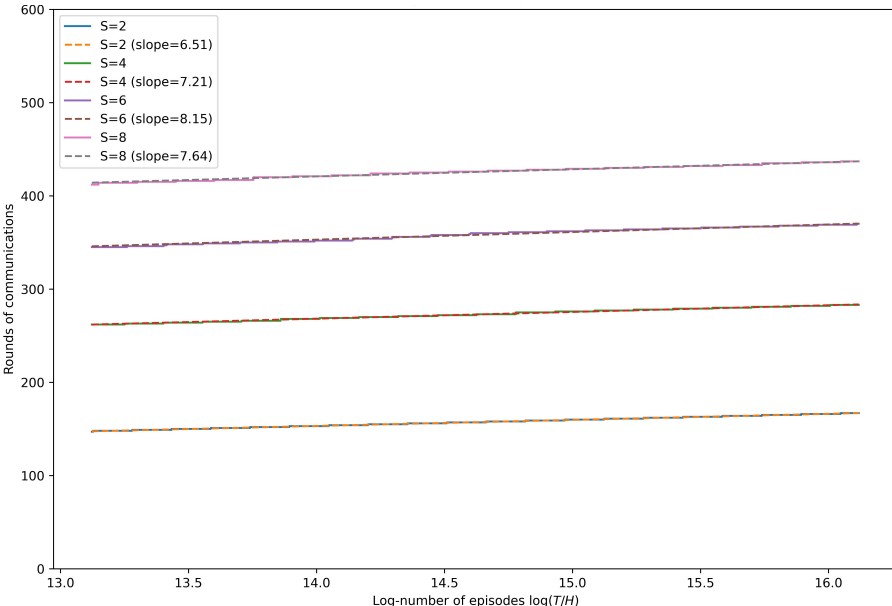

*Figure 4.* Number of communication rounds vs Log-number of Episodes for different $S$ Values with $H = 2$, $A = 2$ and $M = 2$. Each solid line represents the number of communication rounds for each value of $S \in \{2, 4, 6, 8\}$ after $5 \times 10^5$ episodes, while the dashed line represents the fitted line for each $S$.

### B.2.3. DEPENDENCY ON $A$

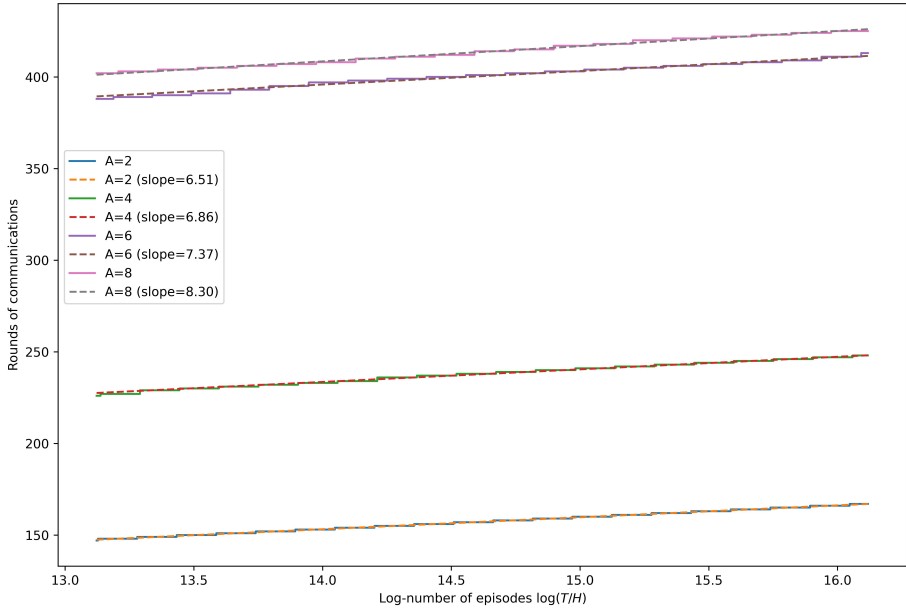

*Figure 5.* Number of communication rounds vs Log-number of Episodes for different $A$ Values with $H = 2$, $S = 2$ and $M = 2$. Each solid line represents the number of communication rounds for each value of $A \in \{2, 4, 6, 8\}$ after $5 \times 10^5$ episodes, while the dashed line represents the fitted line for each $A$.

To explore the dependency of communication cost on $A$, we set $(H, S, M) = (2, 2, 2)$ and let $A$ take values in $\{2, 4, 6, 8\}$. We generate $10^7$ episodes for each agent and only consider the communication cost after $5 \times 10^5$ episodes. The Figure 5 shows the communication cost results for each $A$ after $5 \times 10^5$ episodes.

In Figure 5, each solid line represents the communication cost for each value of $A \in \{2, 4, 6, 8\}$ after $5 \times 10^5$ episodes, while the dashed line represents the corresponding fitted line. Since the x-axis represents the log-number of episodes, $\log(T/H)$, the slope of the fitted line is very close to the coefficient of the $\log T$-term in the communication cost when $\log T$ is sufficently large. We observe that the slopes of these fitted lines are very similar, which indicates that for any given MDP, the coefficient of the $\log T$-term in the communication cost is independent of $A$.

## C. Algorithm Review

### C.1. FedQ-Hoeffding Algorithm

In this section, we present more details for Section 3.1. Denote $\eta_t = \frac{H+1}{H+t}$, $\eta_0^0 = 1$, $\eta_0^t = 0$ for $t \geq 1$, and $\eta_i^t = \eta_i \prod_{i'=i+1}^{t}(1 - \eta_{i'}), \forall 1 \leq i \leq t$. We also denote $\eta^c(t_1, t_2) = \prod_{t=t_1}^{t_2}(1 - \eta_t)$ for any positive integers $t_1 < t_2$. After receiving the information from each agent $m$, for each triple $(s, a, h)$ visited by the agents, the server sets $\eta_{s,a}^{h,k} = 1 - \eta^c\big(N_h^k(s, a) + 1, N_h^{k+1}(s, a)\big)$ and $\beta_{s,a,h}^k = \sum_{t=t^{k-1}+1}^{t^k} \eta_t^{t^k} b_t$, where the confidence bound is given by $b_t = c\sqrt{\frac{H^3 \iota}{t}}$ for some sufficiently large constant $c > 0$. Then the server updates the $Q$-estimate according to the following two cases.

**Case 1:** $N_h^k(s, a) < 2MH(H+1) =: i_0$. This case implies that each client can visit each $(s, a)$ pair at step $h$ at most once. Then, we denote $1 \leq m_{N_h^k} < m_{N_h^k+1} \ldots < m_{N_h^{k+1}} \leq M$ as the agent indices with $n_h^{m,k}(s, a) > 0$. The server then updates the global estimate of action values sequentially as follows:

$$Q_h^{k+1}(s, a) = (1 - \eta_t)Q_h^k(s, a) + \eta_t\big(r_h(x, a) + v_{h+1}^{m_t,k}(s, a) + b_t\big), t = N_h^k(s, a) + 1, \ldots N_h^{k+1}(s, a). \qquad (11)$$

**Case 2:** $N_h^k(s, a) \geq i_0$. In this case, the central server calculates $v_{h+1}^k(s, a) = \sum_{m=1}^{M} v_{h+1}^{m,k}(s, a)/n_h^k(s, a)$ and updates

$$Q_h^{k+1}(s, a) = (1 - \eta_{s,a}^{h,k})Q_h^k(s, a) + \eta_{s,a}^{h,k}\big(r_h(s, a) + v_{h+1}^k(s, a)\big) + \beta_{s,a,h}^k. \qquad (12)$$

After finishing updating the estimated $Q$ function, the server updates the estimated value function and the policy as follows:

$$V_h^{k+1}(s) = \min\Big\{H, \max_{a' \in \mathcal{A}} Q_h^{k+1}(s, a')\Big\}, \pi_h^{k+1}(s) = \arg\max_{a' \in \mathcal{A}} Q_h^{k+1}(s, a'), \forall(s, h) \in \mathcal{S} \times [H]. \qquad (13)$$

The details of the FedQ-Hoeffding algorithm are presented below.

---

**Algorithm 1** FedQ-Hoeffding (Central Server)

---

1: **Input:** $T_0 \in \mathbb{N}_+$.
2: **Initialization:** $k = 1, N_h^1(s, a) = 0, Q_h^1(s, a) = V_h^1(s) = H, \forall(s, a, h) \in \mathcal{S} \times \mathcal{A} \times [H]$ and $\pi^1 = \{\pi_h^1 : \mathcal{S} \to \mathcal{A}\}_{h \in [H]}$ is an arbitrary deterministic policy.
3: **while** $\sum_{h=1}^{H} \sum_{s,a} N_h^k(s, a) < T_0$ **do**
4:    Broadcast $\pi^k, \{N_h^k(s, \pi_h^k(s))\}_{s,h}$ and $\{V_h^k(s)\}_{s,h}$ to all clients.
5:    Wait until receiving an abortion signal and send the signal to all agents.
6:    Receive $\{r_h(s, \pi_h^k(s))\}_{s,h}, \{n_h^{m,k}(s, \pi_h^k(s))\}_{s,h,m}$ and $\{v_{h+1}^{m,k}(s, \pi_h^k(s))\}_{s,h,m}$ from clients.
7:    Calculate $N_h^{k+1}(s, a), n_h^k(s, a), v_{h+1}^k(s, a), \forall(s, h) \in \mathcal{S} \times [H]$ with $a = \pi_h^k(s)$.
8:    **for** $(s, a, h) \in \mathcal{S} \times \mathcal{A} \times [H]$ **do**
9:      **if** $a \neq \pi_h^k(s)$ **or** $n_h^k(s, a) = 0$ **then**
10:        $Q_h^{k+1}(s, a) \leftarrow Q_h^k(s, a)$.
11:      **else if** $N_h^k(s, a) < i_0$ **then**
12:        Update $Q_h^{k+1}(s, a)$ according to Equation (11).
13:      **else**
14:        Update $Q_h^{k+1}(s, a)$ according to Equation (12).
15:      **end if**
16:    **end for**
17:    Update $V_h^{k+1}$ and $\pi^{k+1}$ by Equation (13).
18:    $k \leftarrow k + 1$.
19: **end while**

---

---

**Algorithm 2** FedQ-Hoeffding (Agent $m$ in round $k$)

---

1: Initialize $n_h^m(s,a) = v_{h+1}^m(s,a) = r_h(s,a) = 0, \forall (s,a,h) \in \mathcal{S} \times \mathcal{A} \times [H]$.
2: Receive $\pi^k$, $\{N_h^k(s, \pi_h^k(s))\}_{s,h}$ and $\{V_h^k(s)\}_{s,h}$ from the central server.
3: **while** no abortion signal from the central server **do**
4:    **while** $n_h^m(s_h, a_h) < \max\left\{1, \lfloor \frac{1}{MH(H+1)} N_h^k(s_h, a_h) \rfloor\right\}, \forall (s,a,h) \in \mathcal{S} \times \mathcal{A} \times [H]$ **do**
5:       Collect a new trajectory $\{(s_h, a_h, r_h)\}_{h=1}^H$ with $a_h = \pi_h^k(s_h)$.
6:       $n_h^m(s_h, a_h) \leftarrow n_h^m(s_h, a_h) + 1$, $v_{h+1}^m(s_h, a_h) \leftarrow v_{h+1}^m(s_h, a_h) + V_{h+1}^k(s_{h+1})$, and $r_h(s_h, a_h) \leftarrow r_h, \forall h \in [H]$.
7:    **end while**
8:    Send an abortion signal to the central server.
9: **end while**
10: $n_h^{m,k}(s,a) \leftarrow n_h^m(s,a)$, $v_{h+1}^{m,k}(s,a) \leftarrow v_{h+1}^m(s,a), \forall (s,h) \in \mathcal{S} \times [H]$ with $a = \pi_h^k(s)$.
11: Send $\{r_h(s, \pi_h^k(s))\}_{s,h}$, $\{n_h^{m,k}(s, \pi_h^k(s))\}_{s,h}$ and $\{v_{h+1}^{m,k}(s, \pi_h^k(s))\}_{s,h}$ to the central server.

---

### C.2. FedQ-Bernstein Algorithm

The Bernstein-type algorithm differs from the Hoeffding-type algorithm Algorithms 1 and 2, in that it selects the upper confidence bound based on a variance estimator, akin to the approach used in the Bernstein-type algorithm in Jin et al. (2018). In this subsection, we first review the algorithm design in Zheng et al. (2024).

To facilitate understanding, we introduce additional notations exclusive to Bernstein-type algorithms, supplementing the already provided notations for the Hoeffding-type algorithm.

$$\mu_h^{m,k}(s,a) = \frac{1}{n_h^{m,k}(s,a)} \sum_{j=1}^{n^{m,k}} \left[V_{h+1}^k\left(s_{h+1}^{k,j,m}\right)\right]^2 \mathbb{I}[(s_h^{k,j,m}, a_h^{k,j,m}) = (s,a)].$$

$$\mu_h^k(s,a) = \frac{1}{N_h^{k+1}(s,a) - N_h^k(s,a)} \sum_{m=1}^M \mu_h^{m,k}(s,a) n_h^{m,k}(s,a).$$

Here, $\mu_h^{m,k}(s,a)$ is the sample mean of $[V_{h+1}^k(s_{h+1}^{k,j,m})]^2$ for all the visits of $(s,a,h)$ for the $m$-th agent during the $k$-th round and $\mu_h^k(s,a)$ corresponds to the mean for all the visits during the $k$-th round. We define $W_k(s,a,h)$ to denote the sample variance of all the visits before the $k$-th round, i.e.

$$W_k(s,a,h) = \frac{1}{N_h^k(s,a)} \sum_{i=1}^{N_h^k(s,a)} \left(V_{h+1}^{k^i}(s_{h+1}^{k^i,j^i,m^i}) - \frac{1}{N_h^k(s,a)} \sum_{i'=1}^{N_h^k(s,a)} V_{h+1}^{k^i}(s_{h+1}^{k^i,j^i,m^i})\right)^2.$$

Here, $(k^i, j^i, m^i)$ is the (round, episode, agent) index for the $i$-th visit to $(s,a,h)$ defined in Appendix E. Using the expressions of $\mu_h^k$ and $v_{h+1}^{m,k}$, we further find that

$$W_k(s,a,h) = \frac{1}{N_h^k(s,a)} \sum_{k'=1}^{k-1} \mu_h^{k'}(s,a) n_h^{k'}(s,a) - \left[\frac{1}{N_h^k(s,a)} \sum_{k'=1}^{k-1} v_{h+1}^{k'}(s,a) n_h^{k'}(s,a)\right]^2.$$

Therefore, the quantity $W_k(s,a,h)$ can be calculated efficiently in the following way. Define

$$W_{1,k}(s,a,h) = \sum_{k'=1}^{k-1} \mu_h^{k'}(s,a) n_h^{k'}(s,a), \quad W_{2,k}(s,a,h) = \sum_{k'=1}^{k-1} v_{h+1}^{k'}(s,a) n_h^{k'}(s,a), \tag{14}$$

then we have

$$W_{1,k+1}(s,a,h) = W_{1,k}(s,a,h) + \mu_h^k(s,a) n_h^k(s,a), \quad W_{2,k+1}(s,a,h) = W_{2,k}(s,a,h) + v_{h+1}^k(s,a) n_h^k(s,a) \tag{15}$$

and

$$W_{k+1}(s,a,h) = \frac{W_{1,k+1}(s,a,h)}{N_h^{k+1}(s,a)} - \left[\frac{W_{2,k+1}(s,a,h)}{N_h^{k+1}(s,a)}\right]^2. \tag{16}$$

This indicates that the central server, by actively maintaining and updating the quantities $W_{1,k}$ and $W_{2,k}$ and systematically collecting $n_h^{m,k}$s, $\mu_h^{m,k}$s and $v_{h+1}^{m,k}$s, is able to compute $W_{k+1}$.

Next, we define

$$\beta_t^{\mathrm{B}}(s,a,h) = c' \left( \min \left\{ \sqrt{\frac{H\iota}{t}(W_{k^t+1}(s,a,h) + H)} + \iota \frac{\sqrt{H^7 SA} + \sqrt{MSAH^6}}{t}, \sqrt{\frac{H^3\iota}{t}} \right\} \right), \tag{17}$$

in which $c' > 0$ is a positive constant. With this, the upper confidence bound $b_t(s,a,h)$ for a single visit is determined by $\beta_t^{\mathrm{B}}(s,a,h) = 2 \sum_{i=1}^t \eta_i^t b_t(s,a,h)$, which can be calculated as follows:

$$b_1(s,a,h) := \frac{\beta_1^{\mathrm{B}}(s,a,h)}{2}, \; b_t(s,a,h) := \frac{\beta_t^{\mathrm{B}}(s,a,h) - (1 - \eta_t)\beta_{t-1}^{\mathrm{B}}(s,a,h)}{2\eta_t}.$$

When there is no ambiguity, we use the simplified notation $\tilde{b}_t = b_t(s,a,h)$. In the FedQ-Bernstein algorithm, let $\tilde{\beta} = \beta_{t^k}^{\mathrm{B}}(s,a,h) - \eta^c(t^{k-1} + 1, t^k)\beta_{t^{k-1}}^{\mathrm{B}}(s,a,h)$. Then similar to the FedQ-Hoeffding, we can updates the global estimate of the value functions according to the following two cases.

- **Case 1:** $N_h^k(s,a) < i_0$. This case implies that each client can visit each $(s,a)$ pair at step $h$ at most once. Then, we denote $1 \le m_1 < m_2 \ldots < m_{t^k - t^{k-1}} \le M$ as the agent indices with $n_h^{m,k}(s,a) > 0$. The server then updates the global estimate of action values as follows:

$$Q_h^{k+1}(s,a) = (1 - \eta_t)Q_h^k(s,a) + \eta_t \left( r_h(x,a) + v_{h+1}^{m_t,k}(s,a) + \tilde{b}_t \right), t = t^{k-1} + 1, \ldots t^k. \tag{18}$$

- **Case 2:** $N_h^k(s,a) \ge i_0$. In this case, the central server calculates $v_{h+1}^k(s,a) = \sum_{m=1}^M v_{h+1}^{m,k}(s,a)/n_h^k(s,a)$ and updates the $Q$-estimate as

$$Q_h^{k+1}(s,a) = (1 - \eta_{s,a}^{h,k})Q_h^k(s,a) + \eta_{s,a}^{h,k} \left( r_h(s,a) + v_{h+1}^k(s,a) \right) + \tilde{\beta}/2. \tag{19}$$

Then we can present the FedQ-Bernstein Algorithm in Zheng et al. (2024).

---

**Algorithm 3** FedQ-Bernstein (Central Server)

---

1: **Input:** $T_0 \in \mathbb{N}_+$.
2: **Initialization:** $k = 1$, $N_h^1(s,a) = W_{1,k}(s,a,h) = W_{2,k}(s,a,h) = 0, Q_h^1(s,a) = V_h^1(s) = H, \forall(s,a,h) \in \mathcal{S} \times \mathcal{A} \times [H]$ and $\pi^1 = \{\pi_h^1 : \mathcal{S} \to \mathcal{A}\}_{h \in [H]}$ is an arbitrary deterministic policy.
3: **while** $\sum_{h=1}^H \sum_{s,a} N_h^k(s,a) < T_0$ **do**
4:     Broadcast $\pi^k$, $\{N_h^k(s,\pi_h^k(s))\}_{s,h}$ and $\{V_h^k(s)\}_{s,h}$ to all clients.
5:     Wait until receiving an abortion signal and send the signal to all agents.
6:     Receive $\{r_h(s,\pi_h^k(s))\}_{s,h}$, $\{n_h^{m,k}(s,\pi_h^k(s))\}_{s,h,m}$, $\{v_{h+1}^{m,k}(s,\pi_h^k(s))\}_{s,h,m}$ and $\{\mu_h^{m,k}(s,\pi_h^k(s))\}_{s,h,m}$ from clients.
7:     Calculate $N_h^{k+1}(s,a), n_h^k(s,a), v_{h+1}^k(s,a), \mu_h^k(s,a), \forall(s,h) \in \mathcal{S} \times [H]$ with $a = \pi_h^k(s)$.
8:     Calculate $W_k(s,a,h), W_{k+1}(s,a,h), W_{1,k+1}(s,a,h), W_{2,k+1}(s,a,h), \forall(s,h) \in \mathcal{S} \times [H]$ with $a = \pi_h^k(s)$ based on Equation (14), Equation (15) and Equation (16).
9:     **for** $(s,a,h) \in \mathcal{S} \times \mathcal{A} \times [H]$ **do**
10:       **if** $a \ne \pi_h^k(s)$ **or** $n_h^k(s,a) = 0$ **then**
11:         $Q_h^{k+1}(s,a) \leftarrow Q_h^k(s,a)$.
12:       **else if** $N_h^k(s,a) < i_0$ **then**
13:         Update $Q_h^{k+1}(s,a)$ according to Equation (18).
14:       **else**
15:         Update $Q_h^{k+1}(s,a)$ according to Equation (19).
16:       **end if**
17:     **end for**
18:     Update $V_h^{k+1}$ and $\pi^{k+1}$ by Equation (13).
19:     $k \leftarrow k + 1$.
20: **end while**

---

---

**Algorithm 4** FedQ-Bernstein (Agent $m$ in round $k$)

---

1: $n_h^m(s,a) = v_{h+1}^m(s,a) = r_h(s,a) = \mu_h^m(s,a) = 0, \forall (s,a,h) \in \mathcal{S} \times \mathcal{A} \times [H]$.
2: Receive $\pi^k$, $\{N_h^k(s, \pi_h^k(s))\}_{s,h}$ and $\{V_h^k(s)\}_{s,h}$ from the central server.
3: **while** no abortion signal from the central server **do**
4:   **while** $n_h^m(s_h, a_h) < \max\left\{1, \lfloor \frac{1}{MH(H+1)} N_h^k(s_h, a_h) \rfloor\right\}, \forall (s,a,h) \in \mathcal{S} \times \mathcal{A} \times [H]$ **do**
5:     Collect a new trajectory $\{(s_h, a_h, r_h)\}_{h=1}^H$ with $a_h = \pi_h^k(s_h)$.
6:     $n_h^m(s_h, a_h) \leftarrow n_h^m(s_h, a_h) + 1$, $v_{h+1}^m(s_h, a_h) \leftarrow v_{h+1}^m(s_h, a_h) + V_{h+1}^k(s_{h+1})$, $\mu_h^m(s_h, a_h) \leftarrow \mu_h^m(s_h, a_h) + \left[V_{h+1}^k(s_{h+1})\right]^2$, and $r_h(s_h, a_h) \leftarrow r_h, \forall h \in [H]$.
7:   **end while**
8:   Send an abortion signal to the central server.
9: **end while**
10: $n_h^{m,k}(s,a) \leftarrow n_h^m(s,a), v_{h+1}^{m,k}(s,a) \leftarrow v_{h+1}^m(s,a)$ and $\mu_h^{m,k}(s,a) \leftarrow \mu_h^m(s,a)/n_h^m(s,a), \forall (s,h) \in \mathcal{S} \times [H]$ with $a = \pi_h^k(s)$.
11: Send $\{r_h(s, \pi_h^k(s))\}_{s,h}$, $\{n_h^{m,k}(s, \pi_h^k(s))\}_{s,h}$, $\{\mu_h^{m,k}(s, \pi_h^k(s))\}_{s,h}$ and $\{v_{h+1}^{m,k}(s, \pi_h^k(s))\}_{s,h}$ to the central server.

---

## D. Technical Lemmas

**Lemma D.1.** (Freedman's inequality, Theorem EC.1 of Li et al. (2024)) *Consider a filtration $\mathcal{F}_0 \subset \mathcal{F}_1 \subset \mathcal{F}_2 \subset \cdots$, and let $\mathbb{E}_k$ stand for the expectation conditioned on $\mathcal{F}_k$. Suppose that*

$$Y_n = \sum_{k=1}^n X_k \in \mathbb{R},$$

*where $\{X_k\}$ is a real-valued scalar sequence obeying*

$$|X_k| \le R \quad and \quad \mathbb{E}_{k-1}[X_k] = 0 \quad for\ all\ k \ge 1$$

*for some quantity $R < \infty$. We also define*

$$W_n := \sum_{k=1}^n \mathbb{E}_{k-1}[X_k^2].$$

*In addition, suppose that $W_n \le \sigma^2$ holds deterministically for some given quantity $\sigma^2 < \infty$. Then for any positive integer $m \ge 1$, with probability at least $1 - \delta$, one has*

$$|Y_n| \le \sqrt{8 \max\left\{W_n, \frac{\sigma^2}{2^m}\right\} \log \frac{2m}{\delta}} + \frac{4}{3} R \log \frac{2m}{\delta}.$$

**Lemma D.2.** (Lemma 10 in Zhang et al. (2022a)) *Let $X_1, X_2, \ldots$ be a sequence of random variables taking value in $[0, l]$. Define $\mathcal{F}_k = \sigma(X_1, X_2, \ldots, X_{k-1})$ and $Y_k = \mathbb{E}[X_k | \mathcal{F}_k]$ for $k \ge 1$. For any $\delta > 0$, we have that*

$$\mathbb{P}\left[\exists n, \sum_{k=1}^n X_k \ge 3 \sum_{k=1}^n Y_k + l \log(1/\delta)\right] \le \delta$$

*and*

$$\mathbb{P}\left[\exists n, \sum_{k=1}^n Y_k \ge 3 \sum_{k=1}^n X_k + l \log(1/\delta)\right] \le \delta.$$

## E. Key Lemmas

In this section, we introduce some useful lemmas which will be used in the proofs. Before starting, we define $k^i(s,a,h)$, $j^i(s,a,h)$, and $m^i(s,a,h)$ as the **round**, **episode**, and **agent** indices, respectively, for the $i$-th visit to the state-action-step triple $(s,a,h)$ in chronological order. Under the full synchronization assumption, these indices can be constructed as:

$$k^i(s,a,h) = \sup\left\{k \in \mathbb{N}_+ : N_h^k(s,a) < i\right\},$$

$$j^i(s,a,h) = \sup\left\{ j \in \mathbb{N}_+ : \sum_{j'=1}^{j-1}\sum_{m=1}^{M} \mathbb{I}\left[(s,a) = (s_h^{k^i,j',m}, a_h^{k^i,j',m})\right] < i - N_h^{k^i}(s,a) \right\},$$

$$m^i(s,a,h) = \sup\left\{ m \in \mathbb{N}_+ : \sum_{m'=1}^{m-1} \mathbb{I}\left[(s,a) = (s_h^{k^i,j^i,m'}, a_h^{k^i,j^i,m'})\right] \right.$$
$$\left. < i - N_h^{k^i}(s,a) - \sum_{j'=1}^{j^i-1}\sum_{m=1}^{M} \mathbb{I}\left[(s,a) = (s_h^{k^i,j',m}, a_h^{m,k^i,j',m})\right] \right\}.$$

When there is no ambiguity, we use $k^i$, $m^i$ and $j^i$ for short. Next, we begin to introduce the lemmas. First, Lemma E.1 establishes some relationships between some quantities used in Algorithm 1 and Algorithm 2.

**Lemma E.1.** (Paraphrased from Lemma B.1 in Zheng et al. (2024)). *The following relationships hold for both algorithms.*

*(a)* $T_0 \leq \hat{T}$.

*(b)* $N_h^K(s,a) \leq \sum_{s,a} N_h^K(s,a) \leq T_0/H$.

*(c) For any $(s,a,h,k) \in \mathcal{S} \times \mathcal{A} \times [H] \times [K]$, we have*

$$n_h^{m,k}(s,a) \leq \max\left\{1, \left\lfloor \frac{N_h^k(s,a)}{MH(H+1)} \right\rfloor\right\}, \forall m \in [M],$$

*If $N_h^k(s,a) < i_0$,*

$$n_h^{m,k}(s,a) \leq 1, \; n_h^k(s,a) \leq M.$$

*If $N_h^k(s,a) \geq i_0$,*

$$n_h^{m,k}(s,a) \leq \frac{N_h^k(s,a)}{MH(H+1)}, n_h^k(s,a) \leq \frac{N_h^k(s,a)}{H(H+1)}.$$

*(d) For any $(s,a,h) \in \mathcal{S} \times \mathcal{A} \times [H]$,*

$$N_h^{K+1}(s,a) \leq \sum_{s,a} N_h^{K+1}(s,a) \leq \left(1 + \frac{1}{H(H+1)}\right)\frac{T_0}{H} + MSA.$$

*(e) Let*

$$T_1 = \left(1 + \frac{1}{H(H+1)}\right)T_0 + MHSA,$$

*then we have $\hat{T} \leq T_1 \leq 2\hat{T} + MHSA$.*

*(f)* $K \leq \frac{T_1}{H}$.

*Proof of Lemma E.1.* (a), (b), (c) can be directly proved given and Algorithm 1 and Algorithm 2.

(d) By property (b) and (c), it holds that

$$\sum_{s,a} N_h^{K+1}(s,a) \leq \sum_{s,a} N_h^K(s,a) + \sum_{s,a} n_h^K(s,a) \leq \frac{T_0}{H} + \sum_{s,a}\left(M + \frac{N_h^k(s,a)}{H(H+1)}\right) \leq \left(1 + \frac{1}{H(H+1)}\right)\frac{T_0}{H} + MSA.$$

(e) With conclusion (d), we have $\hat{T} = \sum_{s,a,h} N_h^{K+1}(s,a) \leq T_1$. The second inequality is because of (a).

(f) It is because $K \leq \hat{T}/H \leq T_1/H$. $\qquad\square$

Next, we define new weights $\tilde{\eta}_i^t$. For any $(s, a, h, k) \in \mathcal{S} \times \mathcal{A} \times [H] \times [K]$, we let $t = N_h^k(s, a)$ and $i \in [t] \bigcup \{0\}$. Let $t' = N_h^{k^i}(s, a)$ and $t'' = N_h^{k^i+1}(s, a)$, we denote

$$\tilde{\eta}_i^t(s, a, h) = \eta_i^t \mathbb{I}[t' < i_0] + \frac{1 - \eta^c(t' + 1, t'')}{t'' - t'} \eta^c(t'' + 1, t) \mathbb{I}[t' \geq i_0],$$

and we will use the simplified notation $\tilde{\eta}_i^t$ when there is no ambiguity. In Lemma E.2, we will present some properties of the new weights and their relationship with the original weights $\eta_i^t$.

**Lemma E.2.** *The following properties holds:*

*(a) For all $t \in \mathbb{N}_+$, $\sum_{i=t}^{\infty} \eta_t^i = 1 + 1/H$.*

*(b) For any $k, k' \in \mathbb{N}_+$ such that $t = N_h^{k'}(s, a)$ and $k < k'$, we have*

$$\sum_{i=N_h^k+1}^{N_h^{k+1}} \tilde{\eta}_{i'}^t(s, a, h) = \sum_{i=N_h^k+1}^{N_h^{k+1}} \eta_i^t,$$

*which further indicates that*

$$\sum_{i=1}^{t} \tilde{\eta}_i^t = \mathbb{I}[t > 0].$$

*(c) For any $t \in \mathbb{N}_+$ and any $i \in [t]$, we have that*

$$\tilde{\eta}_i^t / \eta_i^t \leq \exp(1/H).$$

*(d) For any $t \in \mathbb{N}_+$ and any $(s, a, h) \in \mathcal{S} \times \mathcal{A} \times [H]$, if $t < i$, $k^i(s, a, h) = k$ and $N_h^k(s, a) \geq i_0$, we have that $\eta_t^{N_h^k}/\eta_t^i \leq \exp(1/H)$.*

*(e) $1/t^\alpha \leq \sum_{i=1}^{t} \eta_i^t / i^\alpha \leq 2/t^\alpha$.*

*Proof.* Here (a), (b) and (c) are from Lemma B.2 and B.3 in Zheng et al. (2024) and (e) is from Lemma 1 of Li et al. (2021), so here we only prove the property (d). Note that

$$\frac{\eta_t^{N_h^k}}{\eta_t^i} = \prod_{q=N_h^k+1}^{i} (1 - \eta_q)^{-1} \overset{\text{(I)}}{\leq} \left(1 - \eta_{N_h^k+1}\right)^{-(i-N_h^k)} \overset{\text{(II)}}{\leq} \left(1 - \eta_{N_h^k+1}\right)^{-\frac{N_h^k}{H(H+1)}} = \left(1 + \frac{H+1}{N_h^k}\right)^{\frac{N_h^k}{H(H+1)}} \leq \exp(1/H).$$

Here (I) is because $\eta_q$ is monotonically decreasing. (II) is because $i - N_h^k(s, a) \leq n_h^k(s, a) \leq \frac{N_h^k(s, a)}{H(H+1)}$ for $N_h^k(s, a) \geq i_0$ by (c) of Lemma E.1. $\qquad \square$

**Lemma E.3.** *For any non-negative weight sequence $\{\omega_h^{k,j,m}\}_{h,k,j,m}$ and $\alpha \in [0, 1)$, it holds for any $h \in [H]$ that:*

$$\sum_{k,j,m,N_h^k > 0} \frac{\omega_h^{k,j,m}}{N_h^k(s_h^{k,j,m}, a_h^{k,j,m})^\alpha} \leq \sum_{k,j,m} \omega_h^{k,j,m} \frac{\mathbb{I}\left[0 < N_h^k(s_h^{k,j,m}, a_h^{k,j,m}) < M\right]}{N_h^k(s_h^{k,j,m}, a_h^{k,j,m})^\alpha} + \frac{2^\alpha}{1 - \alpha}(SA\|\omega\|_{\infty,h})^\alpha \|\omega\|_{1,h}^{1-\alpha}$$

$$\leq 2MSA\|\omega\|_{\infty,h} + \frac{2^\alpha}{1 - \alpha}(SA\|\omega\|_{\infty,h})^\alpha \|\omega\|_{1,h}^{1-\alpha}.$$

*Here, $\|\omega\|_{\infty,h} = \max_{k,j,m}\{\omega_h^{k,j,m}\}$ and $\|\omega\|_{1,h} = \sum_{k,j,m} \omega_h^{k,j,m}$.*

*Proof.* We can decompose the summation into two terms

$$\sum_{k,j,m,N_h^k>0} \frac{\omega_h^{k,j,m}}{N_h^k(s_h^{k,j,m}, a_h^{k,j,m})^\alpha}$$

$$= \sum_{k,j,m} \frac{\omega_h^{k,j,m}}{N_h^k(s_h^{k,j,m}, a_h^{k,j,m})^\alpha} \left( \mathbb{I}\left[ 0 < N_h^k(s_h^{k,j,m}, a_h^{k,j,m}) < M \right] + \mathbb{I}\left[ N_h^k(s_h^{k,j,m}, a_h^{k,j,m}) \geq M \right] \right)$$

$$= \sum_{s,a} \sum_{k,j,m} \frac{\omega_h^{k,j,m}}{(N_h^k(s,a))^\alpha} \mathbb{I}[(s_h^{k,j,m}, a_h^{k,j,m}) = (s,a)] \left( \mathbb{I}\left[ 0 < N_h^k(s,a) < M \right] + \mathbb{I}\left[ N_h^k(s,a) \geq M \right] \right).$$

Let $k_0(s,a) = \max\{k \mid 1 \leq k \leq K, N_h^k(s,a) < M\}$. Then for the first term, it holds that

$$\sum_{k,j,m} \omega_h^{k,j,m} \frac{\mathbb{I}\left[ 0 < N_h^k(s_h^{k,j,m}, a_h^{k,j,m}) < M \right]}{N_h^k(s_h^{k,j,m}, a_h^{k,j,m})^\alpha}$$

$$= \sum_{s,a} \sum_{k,j,m} \frac{\omega_h^{k,j,m}}{(N_h^k(s,a))^\alpha} \mathbb{I}[(s_h^{k,j,m}, a_h^{k,j,m}) = (s,a)] \mathbb{I}\left[ 0 < N_h^k(s,a) < M \right]$$

$$\leq \|\omega\|_{\infty,h} \sum_{s,a} \sum_{k,j,m} \mathbb{I}[(s_h^{k,j,m}, a_h^{k,j,m}) = (s,a)] \mathbb{I}\left[ 0 < N_h^k(s,a) < M \right]$$

$$= \|\omega\|_{\infty,h} \sum_{s,a} \sum_{k=1}^{k_0} \sum_{j,m} \mathbb{I}\left[ 0 < N_h^k(s,a) < M \right]$$

$$= \|\omega\|_{\infty,h} \sum_{s,a} N_h^{k_0+1}(s,a) \leq 2MSA\|\omega\|_{\infty,h}. \tag{20}$$

The last inequality is because $N_h^{k_0+1}(s,a) = N_h^{k_0}(s,a) + n_h^{k_0}(s,a) \leq 2M$ by $N_h^{k_0}(s,a) < M$. For the second term, let

$$c_h(s,a) = \sum_{k,j,m} \omega_h^{k,j,m} \mathbb{I}[(s_h^{k,j,m}, a_h^{k,j,m}) = (s,a)] \mathbb{I}\left[ N_h^k(s,a) \geq M \right] = \sum_{k=k_0+1}^K \sum_{j,m} \omega_h^{k,j,m} \mathbb{I}[(s_h^{k,j,m}, a_h^{k,j,m}) = (s,a)].$$

Then we have $\sum_{s,a} c_h(s,a) \leq \sum_{k,j,m} \omega_h^{k,j,m} = \|\omega\|_{1,h}$. Given the term

$$\sum_{k,j,m} \frac{\omega_h^{k,j,m}}{(N_h^k(s,a))^\alpha} \mathbb{I}[(s_h^{k,j,m}, a_h^{k,j,m}) = (s,a)] \mathbb{I}\left[ N_h^k(s,a) \geq M \right],$$

when the weights $\omega_h^{k,j,m}$ concentrates on smallest round indices with largest values of $\frac{1}{(N_h^k(s,a))^\alpha}$, we can obtain the largest value. Let $k_0(s,a) < k_1 < k_2 < ... < k_t \leq K$ be all round indices that satisfy $n_h^{k_i}(s,a) > 0$ and let $k_{t+1} = K + 1$. Then we have:

$$c_h(s,a) \leq \|\omega\|_{\infty,h} \sum_{k=k_0+1}^K \sum_{j,m} \mathbb{I}[(s_h^{k,j,m}, a_h^{k,j,m}) = (s,a)] = \|\omega\|_{\infty,h} \sum_{i=1}^t n_h^{k_i}(s,a).$$

Let

$$q = \max\left\{ q \mid 0 \leq q \leq t, \|\omega\|_{\infty,h} \sum_{i=1}^q n_h^{k_i}(s,a) \leq c_h(s,a) \right\},$$

and

$$d = c_h(s,a) - \|\omega\|_{\infty,h} \sum_{i=1}^q n_h^{k_i}(s,a).$$

Then for $q \leq t$, we have the following inequality:

$$\sum_{k,j,m} \frac{\omega_h^{k,j,m}}{(N_h^k(s,a))^\alpha} \mathbb{I}[(s_h^{k,j,m}, a_h^{k,j,m}) = (s,a)] \mathbb{I}\left[ N_h^k(s,a) \geq M \right] \leq \sum_{i=1}^q \|\omega\|_{\infty,h} \frac{n_h^{k_i}(s,a)}{(N_h^{k_i}(s,a))^\alpha} + \frac{d}{(N_h^{k_{q+1}}(s,a))^\alpha}. \tag{21}$$

Note that for any $0 < y < x$ and $\alpha \in [0, 1)$, we have:

$$\frac{x - y}{x^\alpha} \leq \frac{1}{1 - \alpha}(x^{1-\alpha} - y^{1-\alpha}). \tag{22}$$

Then, for any $i \in [t]$, let $x = N_h^{k_i}(s, a)$ and $y = N_h^{k_i+1}(s, a)$, it holds that:

$$\frac{n_h^{k_i}(s, a)}{(N_h^{k_i}(s, a))^\alpha} \leq 2^\alpha \frac{n_h^{k_i}(s, a)}{(N_h^{k_i+1}(s, a))^\alpha} \leq 2^\alpha \left( \frac{(N_h^{k_i+1}(s, a))^{1-\alpha} - (N_h^{k_i}(s, a))^{1-\alpha}}{1 - \alpha} \right). \tag{23}$$

Here the first inequality is because $N_h^{k_i+1}(s, a) = N_h^{k_i}(s, a) + n_h^{k_i}(s, a) \leq 2N_h^{k_i}(s, a)$ by (c) of Lemma E.1. Summing up Equation (23) from 1 to $q$, we know

$$
\begin{aligned}
\sum_{i=1}^q \frac{n_h^{k_i}(s, a)}{(N_h^{k_i}(s, a))^\alpha} &\leq 2^\alpha \sum_{i=1}^q \frac{(N_h^{k_i+1}(s, a))^{1-\alpha} - (N_h^{k_i}(s, a))^{1-\alpha}}{1 - \alpha} \\
&\leq 2^\alpha \sum_{i=1}^q \frac{(N_h^{k_{i+1}}(s, a))^{1-\alpha} - (N_h^{k_i}(s, a))^{1-\alpha}}{1 - \alpha} \\
&= 2^\alpha \left( \frac{(N_h^{k_{q+1}}(s, a))^{1-\alpha}}{1 - \alpha} - \frac{(N_h^{k_1}(s, a))^{1-\alpha}}{1 - \alpha} \right) \\
&\leq 2^\alpha \frac{\left( \sum_{i=1}^q n_h^{k_i}(s, a) \right)^{1-\alpha}}{1 - \alpha}. 
\end{aligned} \tag{24}
$$

The second inequality is because $k_i + 1 \leq k_{i+1}$ and thus $N_h^{k_i+1}(s, a) \leq N_h^{k_{i+1}}(s, a)$. The last inequality is because for any $x > 1$ and $0 \leq \alpha < 1$, we have the following inequality

$$x^{1-\alpha} \leq (x - 1)^{1-\alpha} + 1,$$

and we can let $x = N_h^{k_{q+1}}(s, a)/N_h^{k_1}(s, a)$. Applying Equation (24) to Equation (21), for $q < t$, we have

$$
\begin{aligned}
&\sum_{k,j,m} \frac{\omega_h^{k,j,m}}{(N_h^k(s, a))^\alpha} \mathbb{I}[(s_h^{k,j,m}, a_h^{k,j,m}) = (s, a)] \mathbb{I}\left[ N_h^k(s, a) \geq M \right] \\
&\leq 2^\alpha \|\omega\|_{\infty,h} \frac{\left( \sum_{i=1}^q n_h^{k_i}(s, a) \right)^{1-\alpha}}{1 - \alpha} + \frac{d}{(N_h^{k_{q+1}}(s, a))^\alpha} \\
&\leq 2^\alpha \left( \|\omega\|_{\infty,h} \frac{\left( \sum_{i=1}^q n_h^{k_i}(s, a) \right)^{1-\alpha}}{1 - \alpha} + \frac{d}{(N_h^{k_{q+1}+1}(s, a))^\alpha} \right) \\
&= (2\|\omega\|_{\infty,h})^\alpha \left( \frac{\left( \|\omega\|_{\infty,h} \sum_{i=1}^q n_h^{k_i}(s, a) \right)^{1-\alpha}}{1 - \alpha} + \frac{d}{(N_h^{k_{q+1}+1}(s, a) \|\omega\|_{\infty,h})^\alpha} \right) \\
&\leq (2\|\omega\|_{\infty,h})^\alpha \left( \frac{\left( \|\omega\|_{\infty,h} \sum_{i=1}^q n_h^{k_i}(s, a) \right)^{1-\alpha}}{1 - \alpha} + \frac{d}{(c_h(s, a))^\alpha} \right) \\
&\leq (2\|\omega\|_{\infty,h})^\alpha \frac{(c_h(s, a))^{1-\alpha}}{1 - \alpha}. 
\end{aligned} \tag{25}
$$

Here the second inequality is because $N_h^{k_{q+1}+1}(s, a) \leq 2N_h^{k_{q+1}}(s, a)$ for $q < t$. the second last inequality is because $c_h(s, a) \leq N_h^{k_{q+1}+1}(s, a) \|\omega\|_{\infty,h}$ by the definition of $q$. The last inequality is by Equation (22) with $x = c_h(s, a)$ and $y = \|\omega\|_{\infty,h} \sum_{i=1}^q n_h^{k_i}(s, a)$.

We can also prove the Equation (25) for $q = t$ with $d = 0$. In this case, by applying Equation (24) to Equation (21), it holds that

$$\sum_{k,j,m} \frac{\omega_h^{k,j,m}}{(N_h^k(s,a))^\alpha} \mathbb{I}[(s_h^{k,j,m}, a_h^{k,j,m}) = (s,a)] \mathbb{I}\left[N_h^k(s,a) \geq M\right] \leq 2^\alpha \|\omega\|_{\infty,h} \frac{\left(\sum_{i=1}^q n_h^{k_i}(s,a)\right)^{1-\alpha}}{1-\alpha}$$

$$= (2\|\omega\|_{\infty,h})^\alpha \frac{(c_h(s,a))^{1-\alpha}}{1-\alpha}.$$

Therefore, with Equation (25), we can conclude that

$$\sum_{s,a} \sum_{k,j,m} \frac{\omega_h^{k,j,m}}{(N_h^k(s,a))^\alpha} \mathbb{I}[(s_h^{k,j,m}, a_h^{k,j,m}) = (s,a)] \mathbb{I}\left[N_h^k(s,a) \geq M\right] \leq \frac{2^\alpha \|\omega\|_{\infty,h}^\alpha}{1-\alpha} \sum_{s,a} (c_h(s,a))^{1-\alpha}$$

$$\leq \frac{2^\alpha}{1-\alpha} (SA)^\alpha \|\omega\|_{\infty,h}^\alpha \|\omega\|_{1,h}^{1-\alpha}. \qquad (26)$$

The last inequality is by Hölder's inequality, as $\sum_{s,a} c_h(s,a)^{1-\alpha} \leq (SA)^\alpha \|\omega\|_{1,h}^{1-\alpha}$. Combining the results of Equation (20) and Equation (26), we prove the following conclusion:

$$\sum_{k,j,m,N_h^k>0} \frac{\omega_h^{k,j,m}}{N_h^k(s_h^{k,j,m}, a_h^{k,j,m})^\alpha} \leq 2MSA\|\omega\|_{\infty,h} + \frac{2^\alpha}{1-\alpha}(SA)^\alpha \|\omega\|_{\infty,h}^\alpha \|\omega\|_{1,h}^{1-\alpha}.$$

$\square$

# F. Proofs of Theorem 3.1

### F.1. Auxillary Lemmas

In this section, we provide the proof of the gap-dependent regret bound (Theorem 3.1) for both FedQ-Hoeffding and Fed-Bernstein algorithms together. We first provide several lemmas describing the key properties of $Q$-estimates $Q_h^k(s,a)$.

**Lemma F.1.** (Lemma C.1 of Zheng et al. (2024)). *Using $\forall(s,a,h,k)$ as the simplified notation for $\forall(s,a,h,k) \in \mathcal{S} \times \mathcal{A} \times [H] \times [K]$. Then given any $\delta \in (0,1)$, with probability at least $1 - \delta$, for FedQ-Hoeffding algorithm (Algorithm 1 and Algorithm 2), the following event holds:*

$$\mathcal{G}_1 = \left\{ 0 \leq (Q_h^k - Q_h^\star)(s,a) \leq \eta_0^{N_h^k} H + \sum_{i=1}^{N_h^k} \tilde{\eta}_i^{N_h^k} (V_{h+1}^{k^i} - V_{h+1}^*)(s_{h+1}^{k^i,j^i,m^i}) + \beta_{N_h^k}^{\mathrm{H}}(s,a,h), \ \forall(s,a,h,k) \right\}.$$

*Here, for some sufficiently large constant $c > 0$,*

$$\beta_{N_h^k}^{\mathrm{H}}(s,a,h) = \sum_{i=1}^{N_h^k} \eta_i^{N_h^k} c\sqrt{\frac{H^3\iota}{i}}.$$

**Lemma F.2.** (Lemma E.1 of Zheng et al. (2024)). *Given any $\delta \in (0,1)$, with probability at least $1 - \delta$, for FedQ-Bernstein algorithm (Algorithm 3 and Algorithm 4), the following event holds:*

$$\mathcal{G}_2 = \left\{ 0 \leq (Q_h^k - Q_h^\star)(s,a) \leq \eta_0^{N_h^k} H + \sum_{i=1}^{N_h^k} \tilde{\eta}_i^{N_h^k} (V_{h+1}^{k^i} - V_{h+1}^*)(s_{h+1}^{k^i,j^i,m^i}) + \beta_{N_h^k}^{\mathrm{B}}(s,a,h), \ \forall(s,a,h,k) \right\}.$$

*Here, $\beta_t^{\mathrm{B}}(s,a,h)$ is the cumulative bonus defined in Equation (17).*

Let $\mathcal{X} = (\mathcal{S}, \mathcal{A}, H, K, T, 1/\delta)$. The notation $f(\mathcal{X}) \lesssim g(\mathcal{X})$ means that there exists a universal constant $C_1 > 0$ such that $f(\mathcal{X}) \leq C_1 g(\mathcal{X})$. Then we have the following lemma.

**Lemma F.3.** *For FedQ-Hoeffding algorithm (Algorithm 1 and Algorithm 2), under the event $\mathcal{G}_1$ in Lemma F.1, for any non-negative weight sequence $\{\omega_h^{k,j,m}\}_{h,k,j,m}$, it holds for any $h \in [H]$ that:*

$$\sum_{k,j,m} \omega_h^{k,j,m} \left(Q_h^k - Q_h^*\right)(s_h^{k,j,m}, a_h^{k,j,m}) \lesssim \sqrt{H^5 SA \|\omega\|_{\infty,h} \|\omega\|_{1,h} \iota} + \sum_{h'=h}^{H} \sum_{k,j,m} \omega_{h'}^{k,j,m}(h) Y_{h'}^{k,j,m},$$

*where for any $h \leq h' \leq H - 1$*

$$\omega_h^{k,j,m}(h) := \omega_h^{k,j,m},$$

$$\omega_{h'+1}^{k,j,m}(h) = \sum_{k',j',m'} \omega_{h'}^{k',j',m'}(h) \mathbb{I}\left[N_{h'}^{k'}(s_{h'}^{k',j',m'}, a_{h'}^{k',j',m'}) \geq i_0\right] \sum_{i=1}^{N_{h'}^{k'}} \tilde{\eta}_i^{N_{h'}^{k'}} \mathbb{I}\left[(k^i, j^i, m^i) = (k, j, m)\right],$$

*with*

$$Y_{h'}^{k,j,m} = \eta_0^{N_{h'}^k} H + H \mathbb{I}\left[0 < N_{h'}^k(s_{h'}^{k,j,m}, a_{h'}^{k,j,m}) < i_0\right] + \sqrt{\frac{H^3 \iota}{N_{h'}^k}} \mathbb{I}\left[0 < N_{h'}^k(s_{h'}^{k,j,m}, a_{h'}^{k,j,m}) < M\right].$$

*The same conclusion also holds for FedQ-Bernstein (Algorithm 3 and Algorithm 4) under the event $\mathcal{G}_2$ in Lemma F.2.*

*Proof.* By Lemma F.1, under the event $\mathcal{G}_1$, we have the following relationship

$$\sum_{k,j,m} \omega_h^{k,j,m} \left(Q_h^k - Q_h^*\right)(s_h^{k,j,m}, a_h^{k,j,m})$$

$$\leq \sum_{k,j,m} \omega_h^{k,j,m} \eta_0^{N_h^k} H + \sum_{k,j,m} \omega_h^{k,j,m} \sum_{i=1}^{N_h^k} \tilde{\eta}_i^{N_h^k} (V_{h+1}^{k^i} - V_{h+1}^*)(s_{h+1}^{k^i,j^i,m^i}) + \sum_{k,j,m} \omega_h^{k,j,m} \beta_{N_h^k}^{\mathrm{H}}. \tag{27}$$

**For the third term of Equation (27)**, by (e) of Lemma E.2, we have

$$\beta_{N_h^k}^{\mathrm{H}}(s, a, h) = \sum_{i=1}^{N_h^k} \eta_i^{N_h^k} c \sqrt{\frac{H^3 \iota}{i}} \leq 2c\sqrt{\frac{H^3 \iota}{N_h^k}}.$$

Then by Lemma E.3, it holds that

$$\sum_{k,j,m} \omega_h^{k,j,m} \beta_{N_h^k}^{\mathrm{H}}(s_h^{k,j,m}, a_h^{k,j,m}, h)$$

$$\lesssim \sqrt{H^3 \iota} \sum_{k,j,m} \omega_h^{k,j,m} \sqrt{\frac{1}{N_h^k(s_h^{k,j,m}, a_h^{k,j,m})}}$$

$$\lesssim \sum_{k,j,m} \omega_h^{k,j,m} \sqrt{\frac{H^3 \iota}{N_h^k}} \mathbb{I}\left[0 < N_h^k < M\right] + \sqrt{H^3 SA \|\omega\|_{\infty,h} \|\omega\|_{1,h} \iota}. \tag{28}$$

Next, we will bound the second term of Equation (27). We can decompose the term into two parts as

$$\sum_{k,j,m} \omega_h^{k,j,m} \sum_{i=1}^{N_h^k} \tilde{\eta}_i^{N_h^k} (V_{h+1}^{k^i} - V_{h+1}^*)(s_{h+1}^{k^i,j^i,m^i})$$

$$= \sum_{k,j,m} \omega_h^{k,j,m} \sum_{i=1}^{N_h^k} \tilde{\eta}_i^{N_h^k} (V_{h+1}^{k^i} - V_{h+1}^*)(s_{h+1}^{k^i,j^i,m^i}) \left(\mathbb{I}\left[0 < N_h^k(s_h^{k,j,m}, a_h^{k,j,m}) < i_0\right] + \mathbb{I}\left[N_h^k(s_h^{k,j,m}, a_h^{k,j,m}) \geq i_0\right]\right)$$

For the first part of the second term in Equation (27), we have

$$\sum_{k,j,m} \omega_h^{k,j,m} \sum_{i=1}^{N_h^k} \tilde{\eta}_i^{N_h^k} (V_{h+1}^{k^i} - V_{h+1}^*)(s_{h+1}^{k^i,j^i,m^i}) \mathbb{I}\left[0 < N_h^k(s_h^{k,j,m}, a_h^{k,j,m}) < i_0\right]$$

$$\leq H \sum_{k,j,m} \omega_h^{k,j,m} \mathbb{I}\left[0 < N_h^k(s_h^{k,j,m}, a_h^{k,j,m}) < i_0\right] \sum_{i=1}^{N_h^k} \tilde{\eta}_i^{N_h^k}$$

$$\leq H \sum_{k,j,m} \omega_h^{k,j,m} \mathbb{I}\left[0 < N_h^k(s_h^{k,j,m}, a_h^{k,j,m}) < i_0\right] \tag{29}$$

Here, the second inequality is because $\sum_{i=1}^{N_h^k} \tilde{\eta}_i^{N_h^k} \leq 1$ by (b) of Lemma E.2.

**For the second part of the second term in Equation (27),** we group the summations in a different way.

$$\sum_{k,j,m} \omega_h^{k,j,m} \sum_{i=1}^{N_h^k} \tilde{\eta}_i^{N_h^k} (V_{h+1}^{k^i} - V_{h+1}^*)(s_{h+1}^{k^i,j^i,m^i}) \mathbb{I}\left[N_h^k(s_h^{k,j,m}, a_h^{k,j,m}) \geq i_0\right]$$

$$= \sum_{k,j,m} \sum_{i=1}^{N_h^k} \omega_h^{k,j,m} \tilde{\eta}_i^{N_h^k} (V_{h+1}^{k^i} - V_{h+1}^*)(s_{h+1}^{k^i,j^i,m^i}) \mathbb{I}\left[N_h^k(s_h^{k,j,m}, a_h^{k,j,m}) \geq i_0\right] \left(\sum_{k',j',m'} \mathbb{I}\left[(k^i, j^i, m^i) = (k', j', m')\right]\right)$$

$$= \sum_{k',j',m'} \tilde{\omega}_h^{k',j',m'} \left(V_{h+1}^{k'} - V_{h+1}^*\right)(s_{h+1}^{k',j',m'}), \tag{30}$$

where

$$\tilde{\omega}_h^{k',j',m'} = \sum_{k,j,m} \mathbb{I}\left[N_h^k(s_h^{k,j,m}, a_h^{k,j,m}) \geq i_0\right] \omega_h^{k,j,m} \sum_{i=1}^{N_h^k} \tilde{\eta}_i^{N_h^k} \mathbb{I}\left[(k^i, j^i, m^i) = (k', j', m')\right].$$

Let $\|\tilde{\omega}\|_{\infty,h} = \max_{k,j,m}\{\tilde{\omega}_h^{k,j,m}\}$ and $\|\tilde{\omega}\|_{1,h} = \sum_{k,j,m} \tilde{\omega}_h^{k,j,m}$. Since $\sum_{i=1}^{N_h^k} \tilde{\eta}_i^{N_h^k} \leq 1$ by (b) of Lemma E.2, we have the following property:

$$\|\tilde{\omega}\|_{1,h} = \sum_{k',m',j'} \tilde{\omega}_h^{k',j',m'} \leq \sum_{k,j,m} \mathbb{I}\left[N_h^k(s_h^{k,j,m}, a_h^{k,j,m}) \geq i_0\right] \omega_h^{k,j,m} \leq \|\omega\|_{1,h}. \tag{31}$$

If we have proved that:
$$\|\tilde{\omega}\|_{\infty,h} \leq \exp(3/H)\|\omega\|_{\infty,h}, \tag{32}$$

**then combining the results of Equation (28), Equation (29) and Equation (30) together with Equation (27),** we reach

$$\sum_{k,j,m} \omega_h^{k,j,m} \left(Q_h^k - Q_h^*\right)(s_h^{k,j,m}, a_h^{k,j,m})$$

$$\lesssim \sum_{k',j',m'} \tilde{\omega}_h^{k',j',m'} (V_{h+1}^{k'} - V_{h+1}^*)(s_{h+1}^{k',j',m'}) + \sqrt{H^3 SA \|\omega\|_{\infty,h}\|\omega\|_{1,h}\iota} + \sum_{k,j,m} \omega_h^{k,j,m} \eta_0^{N_h^k} H$$

$$+ \sum_{k,j,m} \omega_h^{k,j,m} H \mathbb{I}\left[N_h^k(s_h^{k,j,m}, a_h^{k,j,m}) < i_0\right] + \sum_{k,j,m} \omega_h^{k,j,m} \sqrt{\frac{H^3\iota}{N_h^k}} \mathbb{I}\left[0 < N_h^k(s_h^{k,j,m}, a_h^{k,j,m}) < M\right]$$

$$\lesssim \sum_{k',j',m'} \tilde{\omega}_h^{k',j',m'} (Q_{h+1}^{k'} - Q_{h+1}^*)(s_{h+1}^{k',j',m'}, a_{h+1}^{k',j',m'}) + \sqrt{H^3 SA \|\omega\|_{\infty,h}\|\omega\|_{1,h}\iota} + \sum_{k,j,m} \omega_h^{k,j,m} Y_h^{k,j,m}. \tag{33}$$

with $\|\tilde{\omega}\|_{1,h} \leq \|\omega\|_{1,h}$ and $\|\tilde{\omega}\|_{\infty,h} \leq \exp(3/H)\|\omega\|_{\infty,h}$. Here the last inequality is because

$$V_{h+1}^{k'}(s_{h+1}^{k',j',m'}) \leq Q_{h+1}^{k'}(s_{h+1}^{k',j',m'}, a_{h+1}^{k',j',m'}) \text{ and } V_{h+1}^*(s_{h+1}^{k',j',m'}) \geq Q_{h+1}^*(s_{h+1}^{k',j',m'}, a_{h+1}^{k',j',m'}).$$

With Equation (33), we develop a recursive relationship for the weighted sum of $Q_h^k - Q_h^*$ between step $h$ and step $h + 1$. By recursions with regard to $h, h + 1, ..., H$, we finish the proof for Algorithm 1 and Algorithm 2.

For Algorithm 3 and Algorithm 4, the only difference lies in the bonus term in Equation (27) and Equation (28). According to Lemma F.2, under the event $\mathcal{G}_2$, we have the same relationship for FedQ-Bernstein algorithm as in Equation (27). Moreover, note that $\beta_{N_h^k}^B(s, a, h) \lesssim \sqrt{\frac{H^3 \iota}{N_h^k}}$ by Equation (17), it is easy to prove the same conclusion as Equation (28). Then the following part remains the same. Now we only need to prove Equation (32).

**Proof of Equation (32):** Now we have

$$\tilde{\omega}_h^{k', j', m'} = \sum_{k,j,m} \mathbb{I} \left[ N_h^k(s_h^{k,j,m}, a_h^{k,j,m}) \geq i_0 \right] \omega_h^{k,j,m} \sum_{i=1}^{N_h^k} \tilde{\eta}_i^{N_h^k} \mathbb{I} \left[ (k^i, j^i, m^i) = (k', j', m') \right]$$

$$\leq \|\omega\|_{\infty, h} \sum_{k,j,m} \mathbb{I} \left[ N_h^k(s_h^{k,j,m}, a_h^{k,j,m}) \geq i_0 \right] \sum_{i=1}^{N_h^k} \tilde{\eta}_i^{N_h^k} \mathbb{I} \left[ (k^i, j^i, m^i) = (k', j', m') \right]$$

We only need to prove for any triple $(k', j', m')$ and any $h \in [H]$,

$$\sum_{k,j,m} \mathbb{I} \left[ N_h^k(s_h^{k,j,m}, a_h^{k,j,m}) \geq i_0 \right] \sum_{i=1}^{N_h^k} \tilde{\eta}_i^{N_h^k} \mathbb{I} \left[ (k^i, j^i, m^i) = (k', j', m') \right] \leq \exp(3/H). \tag{34}$$

For $i \in [N_h^k]$, by definition of $k^i$, $j^i$ and $m^i$, for any given triple $(k', j', m')$,

$$\mathbb{I} \left[ (k^i, j^i, m^i) = (k', j', m') \right] = 1$$

if and only if

$$(s_h^{k,j,m}, a_h^{k,j,m}) = (s_h^{k', j', m'}, a_h^{k', j', m'}), \ k' < k \text{ and } i = i'(k', j', m'),$$

where $i'(k', j', m')$ is the global visiting number for $(s_h^{k', j', m'}, a_h^{k', j', m'})$ at $(k', m', j')$. When there is no ambiguity, we will use $i'$ for short. Therefore

$$\sum_{k,j,m} \mathbb{I} \left[ N_h^k(s_h^{k,j,m}, a_h^{k,j,m}) \geq i_0 \right] \sum_{i=1}^{N_h^k} \tilde{\eta}_i^{N_h^k} \mathbb{I} \left[ (k^i, j^i, m^i) = (k', j', m') \right]$$

$$= \sum_{k=k'+1}^{K} \left( \sum_{j,m} \mathbb{I} \left[ N_h^k(s_h^{k', j', m'}, a_h^{k', j', m'}) \geq i_0, (s_h^{k,j,m}, a_h^{k,j,m}) = (s_h^{k', j', m'}, a_h^{k', j', m'}) \right] \right) \tilde{\eta}_{i'}^{N_h^k}. \tag{35}$$

Let $k' < k_1 < k_2 < ... < k_t \leq K$ be all the round index such that $n_h^{k_q}(s_h^{k', j', m'}, a_h^{k', j', m'}) > 0$ and $N_h^{k_q}(s_h^{k', j', m'}, a_h^{k', j', m'}) \geq i_0$ for any $q \in [t]$, then we can simplify Equation (35):

$$\sum_{k,j,m} \mathbb{I} \left[ N_h^k(s_h^{k,j,m}, a_h^{k,j,m}) \geq i_0 \right] \sum_{i=1}^{N_h^k} \tilde{\eta}_i^{N_h^k} \mathbb{I} \left[ (k^i, j^i, m^i) = (k', j', m') \right]$$

$$= \sum_{q=1}^{t} \left( \sum_{j,m} \mathbb{I} \left[ (s_h^{k_q, j, m}, a_h^{k_q, j, m}) = (s_h^{k', j', m'}, a_h^{k', j', m'}) \right] \right) \tilde{\eta}_{i'}^{N_h^{k_q}}$$

$$\leq \sum_{q=1}^{t} n_h^{k_q}(s_h^{k', j', m'}, a_h^{k', j', m'}) \tilde{\eta}_{i'}^{N_h^{k_q}} \tag{36}$$

For any $q \in [t]$ and $n \in [n_h^{k_q}]$, by (d) of Lemma E.2, the following relationship holds

$$\frac{\eta_{i'}^{N_h^{k_q}}}{\eta_{i'}^{N_h^{k_q} + n}} \leq \exp(1/H). \tag{37}$$

Combining Equation (37) with the property (c) of Lemma E.2, for any $p \in [n_h^{k_q}]$, we have

$$\tilde{\eta}_{i'}^{N_h^{k_q}} \leq \exp(1/H)\eta_{i'}^{N_h^{k_q}} \leq \exp(2/H)\eta_{i'}^{N_h^{k_q}+n},$$

and thus

$$\sum_{q=1}^{t} n_h^{k_q}(s_h^{k',j',m'}, a_h^{k',j',m'})\tilde{\eta}_{i'}^{N_h^{k_q}} \leq \exp(2/H)\sum_{q=1}^{t}\sum_{n=1}^{n_h^{k_q}} \eta_{i'}^{N_h^{k_q}+n} \overset{(I)}{\leq} \exp(2/H)\sum_{r=i'}^{\infty} \eta_{i'}^{r} \leq \exp(3/H).$$

Here (I) is because $k' < k_1 < k_2 < ... < k_t \leq K$ and $N_h^{k_1} \geq N_h^{k'+1} \geq i'$. The last inequality is by (a) of Lemma E.2. Applying this inequality to Equation (36), we complete the proof of Equation (34), and consequently, Equation (32). $\quad\square$

### F.2. Proof of Lemma 4.1

*Proof.* The following proof holds for both FedQ-Hoeffding algorithm and FedQ-Bernstein algorithm.

Let $N = \lceil\log_2(H/\epsilon)\rceil$. For any $i < N$, $k \in [K]$ and given $h \in [H]$, let:

$$\omega_{h,i}^{k,j,m} = \mathbb{I}\left[Q_h^k(s_h^{k,j,m}, a_h^{k,j,m}) - Q_h^*(s_h^{k,j,m}, a_h^{k,j,m}) \in \left[2^{i-1}\epsilon, 2^i\epsilon\right)\right],$$

and

$$\omega_{h,N}^{k,j,m} = \mathbb{I}\left[Q_h^k(s_h^{k,j,m}, a_h^{k,j,m}) - Q_h^*(s_h^{k,j,m}, a_h^{k,j,m}) \in \left[2^{N-1}\epsilon, H\right]\right].$$

Then

$$\|\omega\|_{\infty,h}^{(i)} = \max_{k,j,m}\omega_{h,i}^{k,j,m} \leq 1, \quad \|\omega\|_{1,h}^{(i)} = \sum_{k,j,m}\omega_{h,i}^{k,j,m}.$$

Now for any $i \in [N]$, we have the following relationship:

$$\sum_{k,j,m}\omega_{h,i}^{k,j,m}\left(Q_h^k - Q_h^*\right)(s_h^{k,j,m}, a_h^{k,j,m}) \geq 2^{i-1}\epsilon\|\omega\|_{1,h}^{(i)}. \tag{38}$$

Combining the results of Lemma F.3 and Equation (38), we have:

$$2^{i-1}\epsilon\|\omega\|_{1,h}^{(i)} \lesssim \sqrt{H^5SA\|\omega\|_{1,h}^{(i)}\iota} + \sum_{h'=h}^{H}\sum_{k,j,m}\omega_{h',i}^{k,j,m}(h)Y_{h'}^{k,j,m}, \tag{39}$$

where

$$\omega_{h,i}^{k,j,m}(h) := \omega_{h,i}^{k,j,m},$$

$$\omega_{h'+1,i}^{k,j,m}(h) = \sum_{k',j',m'}\omega_{h',i}^{k',j',m'}(h)\mathbb{I}\left[N_{h'}^{k'}(s_{h'}^{k',j',m'}, a_{h'}^{k',j',m'}) \geq i_0\right]\sum_{i=1}^{N_{h'}^{k'}}\tilde{\eta}_i^{N_{h'}^{k'}}\mathbb{I}\left[(k^i, j^i, m^i) = (k,j,m)\right], h \leq h' \leq H-1,$$

Therefore, for any triple $(k, j, m)$ and $h \leq h' \leq H-1$, we have

$$\sum_{i=1}^{N}\omega_{h'+1,i}^{k,j,m}(h) = \sum_{k',j',m'}\left(\sum_{i=1}^{N}\omega_{h',i}^{k',j',m'}(h)\right)\mathbb{I}\left[N_{h'}^{k'}(s_{h'}^{k',j',m'}, a_{h'}^{k',j',m'}) \geq i_0\right]\sum_{i=1}^{N_{h'}^{k'}}\tilde{\eta}_i^{N_{h'}^{k'}}\mathbb{I}\left[(k^i, j^i, m^i) = (k,j,m)\right]$$

Then by mathematical induction on $h' \in [h, H]$, it is straightforward to prove that for any $j \in [K]$,

$$\sum_{i=1}^{N}\omega_{h',i}^{k,j,m}(h) \leq (\exp(3/H))^{h'-h} < 27, \tag{40}$$

given Equation (34) and the base case $\sum_{i=1}^{N}\omega_{h,i}^{k,j,m}(h) = \sum_{i=1}^{N}\omega_{h,i}^{k,j,m} \leq 1$.

Solving Equation (39), we can derive the following relationship:

$$\|\omega\|_{1,h}^{(i)} \lesssim \frac{H^5 SA\iota}{4^i \epsilon^2} + \frac{\sum_{h'=h}^{H} \sum_{k,j,m} \omega_{h',i}^{k,j,m}(h) Y_{h'}^{k,j,m}}{2^i \epsilon}. \tag{41}$$

We claim that

$$\sum_{h'=1}^{H} \sum_{k,j,m} Y_{h'}^{k,j,m} \lesssim MH^4 SA + M\sqrt{H^5} SA\sqrt{\iota}, \tag{42}$$

which will be proved later. Therefore, by

$$\mathbb{I}\left[\left(Q_h^k - Q_h^*\right)(s_h^{k,j,m}, a_h^{k,j,m}) \geq \epsilon\right] = \sum_{i=1}^{N} \omega_{h,i}^{k,j,m},$$

we have

$$\sum_{h=1}^{H} \sum_{k,j,m} \mathbb{I}\left[Q_h^k(s_h^{k,j,m}, a_h^{k,j,m}) - Q_h^*(s_h^{k,j,m}, a_h^{k,j,m}) \geq \epsilon\right] \leq \sum_{h=1}^{H} \sum_{k,j,m} \sum_{i=1}^{N} \omega_{h,i}^{k,j,m} = \sum_{h=1}^{H} \sum_{i=1}^{N} \|\omega\|_{1,h}^{(i)}. \tag{43}$$

By Equation (41), it holds that

$$\begin{aligned}
\sum_{i=1}^{N} \|\omega\|_{1,h}^{(i)} &\lesssim \sum_{i=1}^{N} \frac{H^5 SA\iota}{4^i \epsilon^2} + \sum_{i=1}^{N} \frac{\sum_{h'=h}^{H} \sum_{k,j,m} \omega_{h',i}^{k,j,m}(h) Y_{h'}^{k,j,m}}{2^i \epsilon} \\
&\lesssim \frac{H^5 SA\iota}{\epsilon^2} + \sum_{i=1}^{N} \frac{\sum_{h'=1}^{H} \sum_{k,j,m} Y_{h'}^{k,j,m}}{2^i \epsilon} \\
&\lesssim \frac{H^5 SA\iota}{\epsilon^2} + \frac{MH^4 SA + M\sqrt{H^5} SA\sqrt{\iota}}{\epsilon}.
\end{aligned} \tag{44}$$

Here, the second inequality is because $0 \leq \omega_{h',i}^{k,j,m}(h) < 27$ by Equation (40) and $Y_{h'}^{k,j,m} \geq 0$. The last inequality is because of Equation (42). Combing the results of Equation (43) and Equation (44), we reach

$$\sum_{h=1}^{H} \sum_{k,j,m} \mathbb{I}\left[Q_h^k(s_h^{k,j,m}, a_h^{k,j,m}) - Q_h^*(s_h^{k,j,m}, a_h^{k,j,m}) \geq \epsilon\right] \lesssim \frac{H^6 SA\iota}{\epsilon^2} + \frac{MH^5 SA + M\sqrt{H^7} SA\sqrt{\iota}}{\epsilon}.$$

Now we finish the proof of the first conclusion. Further, we can prove the second conclusion by noting that

$$\begin{aligned}
&\sum_{h=1}^{H} \sum_{k,j,m} \left(Q_h^k - Q_h^*\right)(s_h^{k,j,m}, a_h^{k,j,m}) \mathbb{I}\left[\left(Q_h^k - Q_h^*\right)(s_h^{k,j,m}, a_h^{k,j,m}) \geq \epsilon\right] \\
&\leq \sum_{h=1}^{H} \sum_{i=1}^{N} 2^i \epsilon \|\omega\|_{1,h}^{(i)} \\
&\lesssim \sum_{h=1}^{H} \sum_{i=1}^{N} \frac{H^5 SA\iota}{2^i \epsilon} + \sum_{h=1}^{H} \sum_{h'=h}^{H} \sum_{k,j,m} \left(\sum_{i=1}^{N} \omega_{h',i}^{k,j,m}(h)\right) Y_{h'}^{k,j,m} \\
&\lesssim \frac{H^6 SA\iota}{\epsilon} + \sum_{h=1}^{H} \sum_{h'=h}^{H} \sum_{k,j,m} Y_{h'}^{k,j,m} \\
&\lesssim \frac{H^6 SA\iota}{\epsilon} + MH^5 SA + M\sqrt{H^7} SA\sqrt{\iota}.
\end{aligned}$$

Here, the second inequality is by Equation (41). The second last inequality is because $\sum_{i=1}^{N} \omega_{h',i}^{k,j,m}(h) < 27$ by Equation (40) and the last inequality is because of Equation (42). Next, we only need to prove Equation (42).

**Proof of Equation (42):** By definition of $Y_{h'}^{k,m,j}$, we have the following equation

$$\sum_{k,m,j} Y_{h'}^{k,j,m} = \sum_{k,j,m} \eta_0^{N_{h'}^k} H + H \sum_{k,m,j} \mathbb{I}\left[N_{h'}^k(s_{h'}^{k,j,m}, a_{h'}^{k,j,m}) < i_0\right] + \sum_{k,m,j} \sqrt{\frac{H^3 \iota}{N_{h'}^k}} \mathbb{I}\left[0 < N_{h'}^k(s_{h'}^{k,j,m}, a_{h'}^{k,j,m}) < M\right].$$

(45)

For the first term of Equation (45), we have

$$\sum_{k,j,m} \eta_0^{N_{h'}^k} H \leq H \sum_{s,a} \sum_{k,j,m} \mathbb{I}[N_{h'}^k(s,a) = 0, (s_{h'}^{k,j,m}, a_{h'}^{k,j,m}) = (s,a)] \leq MHSA.$$

(46)

The inequality here is because if we let $k_0(s,a)$ be the round index such that $N_{h'}^{k_0}(s,a) = 0$ and $N_{h'}^{k_0+1}(s,a) > 0$, then

$$\sum_{k,j,m} \mathbb{I}[N_{h'}^k(s,a) = 0, (s_{h'}^{k,j,m}, a_{h'}^{k,j,m}) = (s,a)] = \sum_{j,m} \mathbb{I}[(s_{h'}^{k_0,j,m}, a_{h'}^{k_0,j,m}) = (s,a)] = n_{h'}^{k_0}(s,a) \leq M.$$

Let $k_1(s,a) = \max\{k \mid 1 \leq k \leq K, N_{h'}^k(s,a) < i_0\}$. Then for the second term of Equation (45), it holds that

$$\sum_{k,j,m} H\mathbb{I}\left[0 < N_{h'}^k(s_{h'}^{k,j,m}, a_{h'}^{k,j,m}) < i_0\right] = H \sum_{s,a} \sum_{k,j,m} \mathbb{I}\left[0 < N_{h'}^k(s,a) < i_0, (s_{h'}^{k,j,m}, a_{h'}^{k,j,m}) = (s,a)\right]$$

$$= H \sum_{s,a} \sum_{k=1}^{k_1} \sum_{j,m} \mathbb{I}\left[(s_{h'}^{k,j,m}, a_{h'}^{k,j,m}) = (s,a)\right]$$

$$= H \sum_{s,a} N_{h'}^{k_1+1}(s,a)$$

$$= H\left(\sum_{s,a} N_{h'}^{k_1}(s,a) + \sum_{s,a} n_{h'}^{k_1}(s,a)\right)$$

$$\leq HSAi_0 + MHSA \leq 5MH^3SA.$$

(47)

Here, the first inequality is because $N_{h'}^{k_1}(s,a) < i_0$ and then $n_{h'}^{k_1}(s,a) \leq M$ by (c) of Lemma E.1. Finally, for the last term of Equation (45), by Equation (20) with $\alpha = 1/2$ and $\omega_h^{k,j,m} = 1$, we have

$$\sum_{k,m,j} \sqrt{\frac{H^3 \iota}{N_{h'}^k}} \mathbb{I}\left[0 < N_{h'}^k(s_{h'}^{k,j,m}, a_{h'}^{k,j,m}) < M\right] \leq 2M\sqrt{H^3}SA\sqrt{\iota}.$$

(48)

Applying Equation (46), Equation (47) and Equation (48) to Equation (45), we know

$$\sum_{h'=1}^{H} \sum_{k,m,j} Y_{h'}^{k,j,m} \lesssim \sum_{h'=1}^{H} (MH^3SA + M\sqrt{H^3}SA\sqrt{\iota}) = MH^4SA + M\sqrt{H^5}SA\sqrt{\iota}.$$

$\square$

### F.3. Proof of Lemma 4.2

*Proof.* The following proof holds for both FedQ-Hoeffding algorithm and FedQ-Bernstein algorithm.

To begin, note that

$$\left(V_1^* - V_1^{\pi^k}\right)(s_1^{k,j,m}) = V_1^*(s_1^{k,j,m}) - Q_1^*(s_1^{k,j,m}, a_1^{k,j,m}) + \left(Q_1^* - Q_1^{\pi^k}\right)(s_1^{k,j,m}, a_1^{k,j,m})$$

$$= \Delta_1(s_1^{k,j,m}, a_1^{k,j,m}) + \mathbb{E}\left[\left(V_2^* - V_2^{\pi^k}\right)(s_2^{k,j,m}) \mid s_2^{k,j,m} \sim P_1(\cdot \mid s_1^{k,j,m}, a_1^{k,j,m})\right]$$

$$= \mathbb{E}\left[\Delta_1(s_1^{k,j,m}, a_1^{k,j,m}) + \Delta_2(s_2^{k,j,m}, a_2^{k,j,m}) \mid s_2^{k,j,m} \sim P_1(\cdot \mid s_1^{k,j,m}, a_1^{k,j,m})\right]$$

$$+ \mathbb{E}\left[\left(Q_2^* - Q_2^{\pi^k}\right)(s_2^{k,j,m}, a_2^{k,j,m}) \mid s_2^{k,j,m} \sim P_1(\cdot \mid s_1^{k,j,m}, a_1^{k,j,m})\right]$$

$$= \cdots = \mathbb{E}\left[\sum_{h=1}^{H} \Delta_h\left(s_h^{k,j,m}, a_h^{k,j,m}\right) \middle| s_{h+1}^{k,j,m} \sim P_h(\cdot \mid s_h^{k,j,m}, a_h^{k,j,m}), h \in [H-1]\right].$$

Here the second equation is by Bellman equation and Bellman optimality equation Equation (3). Therefore, we can get another expression of the regret

$$\mathbb{E}\left(\text{Regret}(T)\right) = \mathbb{E}\left[\sum_{k,j,m}\left(V_1^* - V_1^{\pi^k}\right)(s_1^{k,j,m})\right] = \mathbb{E}\left[\sum_{k,j,m}\sum_{h=1}^{H}\Delta_h(s_h^{k,j,m}, a_h^{k,j,m})\right].$$

By event $\mathcal{G}_1$ in Lemma F.1 (or $\mathcal{G}_2$ in Lemma F.2 for FedQ-Bernstein algorithm),

$$Q_h^k(s_h^{k,j,m}, a_h^{k,j,m}) = \max_a\{Q_h^k(s_h^{k,j,m}, a)\} \geq \max_a\{Q_h^*(s_h^{k,j,m}, a)\} = V_h^*(s_h^{k,j,m}).$$

Thus, for any episode-step pair $(k, h)$

$$\Delta_h(s_h^{k,j,m}, a_h^{k,j,m}) = \text{clip}\left[V_h^*(s_h^{k,j,m}) - Q_h^*(s_h^{k,j,m}, a_h^{k,j,m}) \mid \Delta_{\min}\right] \leq \text{clip}\left[(Q_h^k - Q_h^*)(s_h^{k,j,m}, a_h^{k,j,m}) \mid \Delta_{\min}\right].$$

which further implies

$$\mathbb{E}\left(\text{Regret}(T)\right) \leq \mathbb{E}\left[\sum_{h=1}^{H}\sum_{k,j,m}\text{clip}\left[(Q_h^k - Q_h^*)(s_h^{k,j,m}, a_h^{k,j,m}) \mid \Delta_{\min}\right]\right].$$

$\square$

### F.4. Bounding the Gap-Dependent Regret

The following proof holds for both FedQ-Hoeffding algorithm and FedQ-Bernstein algorithm (substituting $\mathcal{G}_1$ by $\mathcal{G}_2$).

Let $\delta = 1/T_1$, we have:

$$\mathbb{E}\left(\text{Regret}(T)\right) \leq \mathbb{E}\left[\sum_{h=1}^{H}\sum_{k,j,m}\text{clip}\left[(Q_h^k - Q_h^*)(s_h^{k,j,m}, a_h^{k,j,m}) \mid \Delta_{\min}\right]\bigg|\mathcal{G}_1\right]\mathbb{P}(\mathcal{G}_1)$$

$$+ \mathbb{E}\left[\sum_{h=1}^{H}\sum_{k,j,m}\text{clip}\left[(Q_h^k - Q_h^*)(s_h^{k,j,m}, a_h^{k,j,m}) \mid \Delta_{\min}\right]\bigg|\mathcal{G}_1^c\right]\mathbb{P}(\mathcal{G}_1^c)$$

$$\leq O\left(\frac{H^6 SA\iota}{\Delta_{\min}} + M\sqrt{H^7}SA\sqrt{\iota} + MH^5 SA\right) + \frac{1}{T_1}\cdot HT_1$$

$$= O\left(\frac{H^6 SA\iota}{\Delta_{\min}} + M\sqrt{H^7}SA\sqrt{\iota} + MH^5 SA\right).$$

The last inequality is because under the event $\mathcal{G}_1$, we have

$$\sum_{h=1}^{H}\sum_{k,j,m}\text{clip}\left[(Q_h^k - Q_h^*)(s_h^{k,j,m}, a_h^{k,j,m}) \mid \Delta_{\min}\right] \leq O\left(\frac{H^6 SA\iota}{\Delta_{\min}} + MH^5 SA + M\sqrt{H^7}SA\sqrt{\iota}\right).$$

by Lemma 4.1 with $\epsilon = \Delta_{\min}$ and under the event $\mathcal{G}_1^c$,

$$\sum_{h=1}^{H}\sum_{k,j,m}\text{clip}\left[(Q_h^k - Q_h^*)(s_h^{k,j,m}, a_h^{k,j,m}) \mid \Delta_{\min}\right] \leq HT_1.$$

Since $\iota = \log(\frac{2SAHT_1}{\delta}) = \log(2SAHT_1^2)$, by (e) of Lemma E.1, we have

$$\iota \leq 2\log(2SAHT_1) \leq 2\log\left(2SAH(2\hat{T} + MHSA)\right) \leq O\left(\log(SAH\hat{T}) + \log(MHSA)\right) = O\left(\log(SA\hat{T})\right). \quad (49)$$

The last inequality is because $M, H \le \hat{T}$. Therefore, applying Equation (49), we have

$$\mathbb{E}\left(\text{Regret}(T)\right) \le O\left(\frac{H^6 SA\iota}{\Delta_{\min}} + M\sqrt{H^7}SA\sqrt{\iota} + MH^5 SA\right)$$

$$\le O\left(\frac{H^6 SA\log(MSAT)}{\Delta_{\min}} + M\sqrt{H^7}SA\sqrt{\log(MSAT)} + MH^5 SA\right).$$

## G. Proofs of Theorem 3.3

### G.1. Probability Events

**Lemma G.1.** *Let $\iota' = \log(\frac{2MSAHT_1}{\delta})$ with $\delta \in (0,1)$. Then we have the following conclusion:*

*(a) With probability at least $1 - \delta$, the following event holds:*

$$\mathcal{E}_1 = \left\{\sum_{h=1}^{H}\sum_{k,j,m}\mathbb{I}\left[(Q_h^k - Q_h^\star)(s_h^{k,m,j}, a_h^{k,m,j}) \ge \Delta_{\min}\right] \lesssim C_{\min}\right\}.$$

*(b) For any given deterministic optimal policy $\pi^*$, with probability at least $1 - \delta$, the following event holds:*

$$\mathcal{E}_2 = \left\{\sum_{k=1}^{k'}\sum_{j,m}\mathbb{P}\left(a_h^{k,j,m} \ne \pi_h^*(s_h^{k,j,m}) \mid \pi^k\right) \le 3\sum_{k=1}^{k'}\sum_{j,m}\mathbb{I}\left[a_h^{k,j,m} \ne \pi_h^*(s_h^{k,j,m})\right] + 2\iota, \ \forall h \in [H], k' \in [K]\right\}.$$

*(c) For any $k' \in [K]$, let $R_{k'} = \sum_{k=1}^{k'}\sum_{j,m} 1$, which is the total number of episodes in the first $k'$ rounds. Then with probability at least $1 - \delta$, the following event holds:*

$$\mathcal{E}_3 = \left\{\left|\sum_{k=1}^{k'}\sum_{j,m}\left\{\mathbb{I}\left[s_h^{k,j,m} = s\right] - \mathbb{P}\left(s_h^{k,j,m} = s|\pi^k\right)\right\}\right|\right.$$
$$\left. \le \sqrt{24\left(\sum_{k=1}^{k'}\sum_{j,m}\mathbb{P}\left(s_h^{k,j,m} = s|\pi^k\right)\right)\iota + 9\iota}, \ \forall s' \in \mathcal{S}, h \in [H], k' \in [K]\right\}.$$

*(d) With probability at least $1 - \delta$, the following event holds:*

$$\mathcal{E}_4 = \left\{\left|\sum_{j=1}^{J_k}\left\{\mathbb{I}\left[s_h^{k,j,m} = s\right] - \mathbb{P}\left(s_h^{k,j,m} = s|\pi^k\right)\right\}\right|\right.$$
$$\left. \le \sqrt{24\left(\sum_{j=1}^{J_k}\mathbb{P}\left(s_h^{k,j,m} = s|\pi^k\right)\right)\iota' + 9\iota'}, \ \forall s \in \mathcal{S}, h \in [H], k \in [K], m \in [M]\right\}.$$

*Here, under the full synchronization assumption, we can assume in $k$-th round, each agent will generate $J_k$ episodes. Note that given the round $k$ and the policy $\pi^k$, the probability $\mathbb{P}(s_h^{k,j,m} = s|\pi^k)$ is independent of the index $m, j$. Let $\mathbb{P}_{s,h}^k = \mathbb{P}(s_h^{k,j,m} = s|\pi^k)$, then $\mathcal{E}_4$ can be written as*

$$\mathcal{E}_4 = \left\{\left|\sum_{j=1}^{J_k}\mathbb{I}\left[s_h^{k,j,m} = s\right] - J_k\mathbb{P}_{s,h}^k\right| \le \sqrt{24J_k\mathbb{P}_{s,h}^k\iota' + 9\iota'}, \ \forall s \in \mathcal{S}, h \in [H], k \in [K], m \in [M]\right\}.$$

*Proof.* (a) It is proved by Lemma 4.1.

(b) We order all the episodes in the sequence following the rule: first by round index, second by episode index, and third by agent index. Let $n(k,j,m)$ denote the position of the $j$-th episode of the $m$-th agent in the $k$-th round of the sequence. The filtration $\mathcal{F}_{n(k,j,m)}$ is the $\sigma$-field generated by all the random variables until the first $n(k,j,m)-1$ episodes. When there is no ambiguity, we will abbreviate $n(k,j,m)$ as $n$ and $\mathcal{F}_{n(k,j,m)}$ as $\mathcal{F}_n$. Then we have:

$$\mathbb{P}\left(a_{h'}^{k,j,m} \neq \pi_{h'}^*(s_{h'}^{k,j,m}) \mid \pi^k\right) = \mathbb{P}\left(a_{h'}^{k,j,m} \neq \pi_{h'}^*(s_{h'}^{k,j,m}) \mid \mathcal{F}_n\right).$$

According to Lemma D.2, with probability at least $1 - \delta/T_1^2$, the following inequality holds for any given $h = h' \in [H]$, $k' = k'_0 \in [\frac{T_1}{H}]$ and $R_{k'_0} = \sum_{k=1}^{k'_0} \sum_{j,m} 1 \in [T_1]$ :

$$\sum_{k=1}^{k'_0} \sum_{j,m} \mathbb{P}\left(a_{h'}^{k,j,m} \neq \pi_{h'}^*(s_{h'}^{k,j,m}) \mid \pi^k\right) \leq 3\mathbb{I}\left(a_{h'}^{k,j,m} \neq \pi_{h'}^*(s_{h'}^{k,j,m})\right) + 2\iota, \ \forall k' \in [K].$$

Considering all the possible values of $h = h' \in [H]$, $k' = k'_0 \in [\frac{T_1}{H}]$ and $R_{k'_0} = \sum_{k=1}^{k'_0} \sum_{j,m} 1 \in [T_1]$, with probability at least $1 - \delta$, it holds simultaneously for all $h \in [H]$, $k' \in [\frac{T_1}{H}]$ and $R_{k'} = \sum_{k=1}^{k'} \sum_{j,m} 1 \in [T_1]$ that

$$\sum_{k=1}^{k'} \sum_{j,m} \mathbb{P}\left(a_h^{k,j,m} \neq \pi_h^*(s_h^{k,j,m}) \mid \pi^k\right) \leq 3 \sum_{k=1}^{k'} \sum_{j,m} \mathbb{I}\left(a_h^{k,j,m} \neq \pi_h^*(s_h^{k,j,m})\right) + 2\iota.$$

(c) According to Lemma D.1, with probability at least $1 - \delta/ST_1^2$, the following inequality holds for any given $s' \in \mathcal{S}$, $h = h' \in [H]$, $k' = k'_0 \in [\frac{T_1}{H}]$ and $R_{k'_0} = \sum_{k=1}^{k'_0} \sum_{j,m} 1 \in [T_1]$ :

$$\left| \sum_{k=1}^{k'_0} \sum_{j,m} \left\{ \mathbb{I}\left[ s_{h'}^{k,j,m} = s'\right] - \mathbb{P}\left( s_{h'}^{k,j,m} = s' \mid \pi^k \right) \right\} \right| \leq \sqrt{24 \left( \sum_{k=1}^{k'_0} \sum_{j,m} \mathbb{P}\left( s_{h'}^{k,j,m} = s' \mid \pi^k \right) \right) \iota} + 9\iota.$$

Here, we let $\sigma^2 = T_1$, $m = \lceil \log_2(T_1) \rceil$ in Lemma D.1 and

$$W_n = \sum_{k=1}^{k'_0} \sum_{j,m} \mathbb{P}\left( s_{h'}^{k,j,m} = s' \mid \pi^k \right)\left(1 - \mathbb{P}\left( s_{h'}^{k,j,m} = s' \mid \pi^k \right)\right) \leq \sum_{k=1}^{k'_0} \sum_{j,m} \mathbb{P}\left( s_{h'}^{k,j,m} = s' \mid \pi^k \right).$$

Considering all the possible values of $s = s' \in \mathcal{S}$, $h = h' \in [H]$, $k' = k'_0 \in [\frac{T_1}{H}]$, $\hat{T} = T' \in [T_1]$, with probability at least $1 - \delta$, it holds simultaneously for all $s \in \mathcal{S}$, $h \in [H]$, $k' \in [\frac{T_1}{H}]$ and $\hat{T} \in [T_1]$ that

$$\left| \sum_{k=1}^{k'} \sum_{j,m} \left\{ \mathbb{I}\left[ s_h^{k,j,m} = s\right] - \mathbb{P}\left( s_h^{k,j,m} = s \mid \pi^k \right) \right\} \right| \leq \sqrt{24 \left( \sum_{k=1}^{k'} \sum_{j,m} \mathbb{P}\left( s_h^{k,j,m} = s \mid \pi^k \right) \right) \iota} + 9\iota.$$

(d) The proof is similar to (c) by considering all the combinations of $(s,h,m,k,R_k) \in \mathcal{S} \times [H] \times [M] \times [\frac{T_1}{H}] \times [T_1]$.

$\square$

## G.2. Proof of Lemma 5.1

*Proof.* The event $\mathcal{G}_1 \cap \mathcal{E}_1 \cap \mathcal{E}_2$ holds with probability at least $1 - 3\delta$. Next we will prove Lemma 5.1 under the event $\mathcal{G}_1 \cap \mathcal{E}_1 \cap \mathcal{E}_2$. (For FedQ-Bernstein algorithm, we will prove Lemma 5.1 under the event $\mathcal{G}_2 \cap \mathcal{E}_1 \cap \mathcal{E}_2$.)

For any $h \in [H]$, let set $D_h$ be all triples of $(s,a,h)$ such that $a \notin \mathcal{A}_h^\star(s)$, that is:

$$D_h = \{(s,a,h)|a \notin \mathcal{A}_h^\star(s)\}.$$

We also let the set $D = \bigcup_{h=1}^{H} D_h$ and the set

$$D_{\text{opt}} = \{(s, a, h) | a \in \mathcal{A}_h^\star(s)\}.$$

Then we have $|D| + |D_{\text{opt}}| = SAH$.

If for given $(h, k, j, m)$, $(s_h^{k,m,j}, a_h^{k,m,j}, h) \in D_h$, we have $\Delta_h(s_h^{k,m,j}, a_h^{k,m,j}) \geq \Delta_{\min}$. By event $\mathcal{G}_1$ in Lemma F.1 (or $\mathcal{G}_2$ in Lemma F.2 for FedQ-Bernstein algorithm),

$$Q_h^k(s_h^{k,j,m}, a_h^{k,j,m}) = \max_a\{Q_h^k(s_h^{k,j,m}, a)\} \geq \max_a\{Q_h^*(s_h^{k,j,m}, a)\} = V_h^*(s_h^{k,j,m}).$$

Therefore, it holds that

$$Q_h^k(s_h^{k,m,j}, a_h^{k,m,j}) - Q_h^\star(s_h^{k,m,j}, a_h^{k,m,j}) \geq \Delta_h(s_h^{k,m,j}, a_h^{k,m,j}) \geq \Delta_{\min}.$$

Then we have

$$\mathbb{I}\left[a_h^{k,j,m} \notin \mathcal{A}_h^*(s_h^{k,j,m})\right] = \mathbb{I}\left[(s_h^{k,m,j}, a_h^{k,m,j}, h) \in D_h\right] \leq \mathbb{I}\left[Q_h^k(s_h^{k,m,j}, a_h^{k,m,j}) - Q_h^\star(s_h^{k,m,j}, a_h^{k,m,j}) \geq \Delta_{\min}\right],$$

and thus by the event $\mathcal{E}_1$ in Lemma G.1, it holds that

$$\sum_{h=1}^{H} \sum_{k,j,m} \mathbb{I}\left[a_h^{k,j,m} \notin \mathcal{A}_h^*(s_h^{k,j,m})\right] \leq \sum_{h=1}^{H} \sum_{k,j,m} \mathbb{I}\left[Q_h^k(s_h^{k,m,j}, a_h^{k,m,j}) - Q_h^\star(s_h^{k,m,j}, a_h^{k,m,j}) \geq \Delta_{\min}\right] \leq C_{\min}. \tag{50}$$

Next we prove the second conclusion. Let $\mathcal{S}_h^0 = \{s \mid \mathbb{P}_{s,h}^* = 0\}$. For any given deterministic optimal policy $\pi^*$, we have

$$\mathbb{I}\left[a_h^{k,j,m} \neq \pi_h^*(s_h^{k,j,m})\right] = \mathbb{I}\left[a_h^{k,j,m} \neq \pi_h^*(s_h^{k,j,m}), s_h^{k,j,m} \notin \mathcal{S}_h^0\right] + \mathbb{I}\left[a_h^{k,j,m} \neq \pi_h^*(s_h^{k,j,m}), s_h^{k,j,m} \in \mathcal{S}_h^0\right]. \tag{51}$$

For $s_h^{k,j,m} \notin \mathcal{S}_h^0$, we have $\mathbb{P}_{s_h^{k,j,m},h}^* > 0$ and $|\mathcal{A}_h^*(s_h^{k,j,m})| = 1$ by condition (b) of Definition 3.2. This means $\pi_h^*(s_h^{k,j,m})$ is the only element in $\mathcal{A}_h^*(s_h^{k,j,m})$. Therefore, we have

$$\mathbb{I}\left[a_h^{k,j,m} \neq \pi_h^*(s_h^{k,j,m}), s_h^{k,j,m} \notin \mathcal{S}_h^0\right] \leq \mathbb{I}\left[a_h^{k,j,m} \notin \mathcal{A}_h^*(s_h^{k,j,m})\right]. \tag{52}$$

For the second term in Equation (51), if $h = 1$, because of the randomness of the selection of $s_1^{k,j,m}$, we have $\mathbb{P}(s_1 = s_1^{k,j,m}|\pi^*) = \mathbb{P}(s_1 = s_1^{k,j,m}) > 0$ and then

$$\mathbb{I}\left[a_1^{k,j,m} \neq \pi_1^*(s_1^{k,j,m}), s_1^{k,j,m} \in \mathcal{S}_1^0\right] = 0. \tag{53}$$

To bound the second term in Equation (51) for $h > 1$, we first prove a lemma.

**Lemma G.2.** *For any $h \in [H]$ and the trajectory $\{(s_h^{k,j,m}, a_h^{k,j,m}, r_h^{k,j,m})\}_{h=1}^H$ in $j$-th episode of agent $m$ in round $k$, if $\mathbb{P}_{s_h^{k,j,m},h}^* > 0$ and $a_h^{k,j,m}$ is the unique optimal action for state $s_h^{k,j,m}$ at step $h$, then $\mathbb{P}_{s_{h+1}^{k,j,m},h+1}^* > 0$*

*Proof.* For any given optimal policy $\pi^*$, it holds that

$$\mathbb{P}_{s_{h+1}^{k,j,m},h+1}^* = \mathbb{P}\left(s_{h+1} = s_{h+1}^{k,j,m} \mid \pi^*\right)$$

$$\geq \mathbb{P}\left(s_{h+1} = s_{h+1}^{k,j,m} \mid s_h = s_h^{k,j,m}, a_h = a_h^{k,j,m}, \pi^*\right) \times \mathbb{P}\left(s_h = s_h^{k,j,m}, a_h = a_h^{k,j,m} \mid \pi^*\right)$$

$$\overset{(I)}{=} \mathbb{P}\left(s_{h+1} = s_{h+1}^{k,j,m} \mid s_h = s_h^{k,j,m}, a_h = a_h^{k,j,m}\right) \times \mathbb{P}_{s_h^{k,j,m},h}^* > 0$$

The equation (I) is by Markov property and $\mathbb{P}(s_h = s_h^{k,j,m}, a_h = a_h^{k,j,m} \mid \pi^*) = \mathbb{P}(s_h = s_h^{k,j,m} \mid \pi^*) = \mathbb{P}_{s_h^{k,j,m},h}^*$. $\square$

For every initial state $s_1^{k,j,m}$, we know $\mathbb{P}^*_{s_1^{k,j,m},1} > 0$. Therefore, if for $h > 1$, $\mathbb{P}^*_{s_h^{k,j,m},h} = 0$ and $s_h^{k,j,m} \in \mathcal{S}_h^0$, by Lemma G.2, we know there exists $h' < h$ such that $a_{h'}^{k,m,j}$ is not an optimal action for state $s_{h'}^{k,m,j}$ at step $h'$, otherwise we have $\mathbb{P}^*_{s_h^{k,j,m},h} > 0$. Therefore, for the second term in Equation (51), we have

$$\mathbb{I}\left[a_h^{k,j,m} \neq \pi_h^*(s_h^{k,j,m}), s_h^{k,j,m} \in \mathcal{S}_h^0\right] \leq \mathbb{I}\left[s_h^{k,j,m} \in \mathcal{S}_h^0\right] \leq \sum_{h'=1}^{h-1} \mathbb{I}\left[a_{h'}^{k,j,m} \notin \mathcal{A}_{h'}^*(s_{h'}^{k,j,m})\right]. \tag{54}$$

By combining the results of Equation (52), Equation (53) and Equation (54), we can bound the Equation (51) as follows:

$$\mathbb{I}\left[a_h^{k,j,m} \neq \pi_h^*(s_h^{k,j,m})\right] \leq \sum_{h'=1}^{h} \mathbb{I}\left[a_{h'}^{k,j,m} \notin \mathcal{A}_{h'}^*(s_{h'}^{k,j,m})\right] \leq \sum_{h'=1}^{H} \mathbb{I}\left[a_{h'}^{k,j,m} \notin \mathcal{A}_{h'}^*(s_{h'}^{k,j,m})\right].$$

Therefore, using the first conclusion, Equation (50), we reach

$$\sum_{k,j,m} \mathbb{I}\left[a_h^{k,j,m} \neq \pi_h^*(s_h^{k,j,m})\right] \leq \sum_{k,j,m} \sum_{h'=1}^{H} \mathbb{I}\left[a_{h'}^{k,j,m} \notin \mathcal{A}_{h'}^*(s_{h'}^{k,j,m})\right] \leq C_{\min}$$

By combining this inequality with the event $\mathcal{E}_2$ in Lemma G.1, we can conclude that for any $h \in [H]$ and $k' \in [K]$,

$$\sum_{k=1}^{k'} \sum_{j,m} \mathbb{P}\left(a_h^{k,j,m} \neq \pi_h^*(s_h^{k,j,m}) \mid \pi^k\right) \leq 4C_{\min}.$$

$\square$

### G.3. Proof of Lemma 5.2

*Proof.* The event $\mathcal{G}_1 \cap (\bigcap_{i=1}^4 \mathcal{E}_i)$ holds with probability at least $1 - 5\delta$. Next we will prove Lemma 5.1 under the event $\mathcal{G}_1 \cap (\bigcap_{i=1}^4 \mathcal{E}_i)$. (For FedQ-Bernstein algorithm, we will prove Lemma 5.1 under the event $\mathcal{G}_2 \cap (\bigcap_{i=1}^4 \mathcal{E}_i)$.)

Under the event $\mathcal{G}_1 \cap (\bigcap_{i=1}^4 \mathcal{E}_i)$ (or $\mathcal{G}_2 \cap (\bigcap_{i=1}^4 \mathcal{E}_i)$), we have already proved the Lemma 5.1 in Appendix G.2.

Because $N_h^k(s_0, a_0) > i_1 + i_2 > C_{\min}$, by Lemma 5.1, we know $a_0 \in \mathcal{A}_h^*(s_0)$. Next we prove the second conclusion.

Using the law of total probability, for any $0 \leq h \leq H - 1$, $s \in \mathcal{S}$ and any given deterministic optimal policy $\pi^*$, we have the following relationship

$$\mathbb{P}(s_{h+1}^{k,j,m} = s \mid \pi^k) = \sum_{s'} \mathbb{P}\left(s_{h+1}^{k,j,m} = s | s_h^{k,j,m} = s', a_h^{k,j,m} = \pi_h^*(s'), \pi^k\right) \mathbb{P}\left(s_h^{k,j,m} = s', a_h^{k,j,m} = \pi_h^*(s') \mid \pi^k\right)$$

$$+ \mathbb{P}\left(s_{h+1}^{k,j,m} = s, a_h^{k,j,m} \neq \pi_h^*(s_h^{k,j,m}) \mid \pi^k\right)$$

$$= \sum_{s'} \mathbb{P}_{s,s',h}^{k,j,m} \cdot \mathbb{P}\left(s_h^{k,j,m} = s', a_h^{k,j,m} = \pi_h^*(s') \mid \pi^k\right) + \mathbb{P}\left(s_{h+1}^{k,j,m} = s, a_h^{k,j,m} \neq \pi_h^*(s_h^{k,j,m}) \mid \pi^k\right),$$

$$\tag{55}$$

where

$$\mathbb{P}_{s,s',h}^{k,j,m} = \mathbb{P}\left(s_{h+1}^{k,j,m} = s | s_h^{k,j,m} = s', a_h^{k,j,m} = \pi_h^*(s'), \pi^k\right) = \mathbb{P}\left(s_{h+1}^{k,j,m} = s | s_h^{k,j,m} = s', a_h^{k,j,m} = \pi_h^*(s')\right).$$

The last equality is because of Markov property. We also have

$$\mathbb{P}\left(s_{h+1}^{k,j,m} = s | \pi^*\right) = \sum_{s'} \mathbb{P}\left(s_{h+1}^{k,j,m} = s | s_h^{k,j,m} = s', \pi^*\right) \mathbb{P}\left(s_h^{k,j,m} = s' | \pi^*\right) = \sum_{s'} \mathbb{P}_{s,s',h}^{k,j,m} \cdot \mathbb{P}\left(s_h^{k,j,m} = s' | \pi^*\right), \tag{56}$$

where the last equation is because

$$\mathbb{P}\left(s_{h+1}^{k,j,m} = s | s_h^{k,j,m} = s', \pi^*\right) = \mathbb{P}\left(s_{h+1}^{k,j,m} = s | s_h^{k,j,m} = s', a_h^{k,j,m} = \pi_h^*(s')\right) = \mathbb{P}_{s,s',h}^{k,j,m}.$$

Combining the results of Equation (55) and Equation (56), then it holds

$$\mathbb{P}\left(s_{h+1}^{k,j,m} = s|\pi^k\right) - \mathbb{P}\left(s_{h+1}^{k,j,m} = s|\pi^*\right)$$

$$= \sum_{s'} \mathbb{P}_{s,s',h}^{k,j,m} \left[\mathbb{P}\left(s_h^{k,j,m} = s', a_h^{k,j,m} = \pi_h^*(s')|\pi^k\right) - \mathbb{P}\left(s_h^{k,j,m} = s'|\pi^*\right)\right] + \mathbb{P}\left(s_{h+1}^{k,j,m} = s, a_h^{k,j,m} \neq \pi_h^*(s_h^{k,j,m})|\pi^k\right)$$

$$= \sum_{s'} \mathbb{P}_{s,s',h}^{k,j,m} \left[\mathbb{P}\left(s_h^{k,j,m} = s' \mid \pi^k\right) - \mathbb{P}\left(s_h^{k,j,m} = s'|\pi^*\right)\right] - \sum_{s'} \mathbb{P}_{s,s',h}^{k,j,m} \cdot \mathbb{P}\left(s_h^{k,j,m} = s', a_h^{k,j,m} \neq \pi_h^*(s') \mid \pi^k\right)$$

$$+ \mathbb{P}\left(s_{h+1}^{k,j,m} = s, a_h^{k,j,m} \neq \pi_h^*(s_h^{k,j,m}) \mid \pi^k\right).$$

Therefore for any $0 \leq h \leq H - 1$ and $s \in \mathcal{S}$, by the triangle inequality, it holds that

$$\left|\mathbb{P}\left(s_{h+1}^{k,j,m} = s \mid \pi^k\right) - \mathbb{P}\left(s_{h+1}^{k,j,m} = s|\pi^*\right)\right| \leq \sum_{s'} \mathbb{P}_{s,s',h}^{k,j,m} \left|\mathbb{P}\left(s_h^{k,j,m} = s' \mid \pi^k\right) - \mathbb{P}\left(s_h^{k,j,m} = s'|\pi^*\right)\right|$$

$$+ \sum_{s'} \mathbb{P}_{s,s',h}^{k,j,m} \cdot \mathbb{P}\left(s_h^{k,j,m} = s', a_h^{k,j,m} \neq \pi_h^*(s') \mid \pi^k\right) + \mathbb{P}\left(s_{h+1}^{k,j,m} = s, a_h^{k,j,m} \neq \pi_h^*(s_h^{k,j,m}) \mid \pi^k\right). \tag{57}$$

Summing Equation (57) for all $s \in \mathcal{S}$, since $\sum_s \mathbb{P}_{s,s',h} = 1$, then we can derive the following recursive relationship:

$$\sum_s \left|\mathbb{P}\left(s_{h+1}^{k,j,m} = s \mid \pi^k\right) - \mathbb{P}\left(s_{h+1}^{k,j,m} = s|\pi^*\right)\right|$$

$$\leq \sum_{s'} \left|\mathbb{P}\left(s_h^{k,j,m} = s' \mid \pi^k\right) - \mathbb{P}\left(s_h^{k,j,m} = s'|\pi^*\right)\right| + 2\mathbb{P}\left(a_h^{k,j,m} \neq \pi_h^*(s_h^{k,j,m}) \mid \pi^k\right).$$

Since $\mathbb{P}(s_1^{k,j,m} = s \mid \pi^k) - \mathbb{P}(s_1^{k,j,m} = s|\pi^*) = 0$, by recursion, for any $h' \in [H]$ we can get the following conclusion

$$\sum_s \left|\mathbb{P}\left(s_{h'}^{k,j,m} = s \mid \pi^k\right) - \mathbb{P}\left(s_{h'}^{k,j,m} = s|\pi^*\right)\right| \leq 2 \sum_{h=1}^{h'-1} \mathbb{P}\left(a_h^{k,j,m} \neq \pi_h^*(s_h^{k,j,m}) \mid \pi^k\right)$$

$$\leq 2 \sum_{h=1}^{H} \mathbb{P}\left(a_h^{k,j,m} \neq \pi_h^*(s_h^{k,j,m}) \mid \pi^k\right). \tag{58}$$

Applying Equation (10) in Lemma 5.1 to Equation (58), then for any $h \in [H]$ and $k' \in [K]$, it holds that:

$$\sum_s \sum_{k=1}^{k'} \sum_{j,m} \left|\mathbb{P}\left(s_h^{k,j,m} = s \mid \pi^k\right) - \mathbb{P}\left(s_h^{k,j,m} = s|\pi^*\right)\right| \leq 2 \sum_{h=1}^{H} \sum_{k=1}^{k'} \sum_{j,m} \mathbb{P}\left(a_h^{k,j,m} \neq \pi_h^*(s_h^{k,j,m}) \mid \pi^k\right) \leq 8HC_{\min}.$$

Based on the property (b) of Definition 3.2, we have $\mathbb{P}(s_h^{k,j,m} = s|\pi^*) = \mathbb{P}_{s,h}^*$, then for any $s \in \mathcal{S}$, $h \in [H]$ and $k' \in [K]$, we also have

$$\left|\sum_{k=1}^{k'} \sum_{j,m} \mathbb{P}\left(s_h^{k,j,m} = s \mid \pi^k\right) - R_{k'}\mathbb{P}_{s,h}^*\right| \leq \sum_{k=1}^{k'} \sum_{j,m} \left|\mathbb{P}\left(s_h^{k,j,m} = s \mid \pi^k\right) - \mathbb{P}\left(s_h^{k,j,m} = s|\pi^*\right)\right| \leq 8HC_{\min}. \tag{59}$$

and thus by the triangle inequality

$$\sum_{k=1}^{k'} \sum_{j,m} \mathbb{P}\left(s_h^{k,j,m} = s \mid \pi^k\right) \leq R_{k'}\mathbb{P}_{s,h}^* + 8HC_{\min}. \tag{60}$$

Applying Equation (60) to $\mathcal{E}_3$ in Lemma G.1, for any $s \in \mathcal{S}$, $h \in [H]$ and $k' \in [K]$, we have

$$\left|\sum_{k=1}^{k'} \sum_{j,m} \left\{\mathbb{I}\left[s_h^{k,j,m} = s\right] - \mathbb{P}\left(s_h^{k,j,m} = s|\pi^k\right)\right\}\right| \leq \sqrt{24\left(R_{k'}\mathbb{P}_{s,h}^* + 8HC_{\min}\right)\iota} + 9\iota$$

$$\leq 5\sqrt{R_{k'}\mathbb{P}_{s,h}^*\iota} + 23HC_{\min}. \tag{61}$$

Combining the results of Equation (59) and Equation (61), by triangle inequality, we can derive the following relationship for any $s \in \mathcal{S}$, $h \in [H]$ and $k' \in [K]$

$$\left| \sum_{k=1}^{k'} \sum_{j,m} \mathbb{I}\left[ s_h^{k,j,m} = s \right] - R_{k'} \mathbb{P}_{s,h}^* \right| \leq 5\sqrt{R_{k'} \mathbb{P}_{s,h}^* \iota} + 31 H C_{\min}. \tag{62}$$

Then by triangle inequality, it holds for any $s \in \mathcal{S}$, $h \in [H]$ and $k' \in [K]$ that

$$\left| \sum_{k=1}^{k'} \sum_{j,m} \mathbb{I}\left[ s_h^{k,j,m} = s, a_h^{k,j,m} \in \mathcal{A}_h^*(s) \right] - R_{k'} \mathbb{P}_{s,h}^* \right|$$

$$\leq \left| \sum_{k=1}^{k'} \sum_{j,m} \mathbb{I}\left[ s_h^{k,j,m} = s \right] - R_{k'} \mathbb{P}_{s,h}^* \right| + \sum_{k=1}^{k'} \sum_{j,m} \mathbb{I}\left[ s_h^{k,j,m} = s, a_h^{k,j,m} \notin \mathcal{A}_h^*(s) \right]$$

$$\leq 5\sqrt{R_{k'} \mathbb{P}_{s,h}^* \iota} + 32 H C_{\min}.$$

Here, the last inequality is by Equation (62) and also the fact that

$$\sum_{k=1}^{k'} \sum_{j,m} \mathbb{I}\left[ s_h^{k,j,m} = s, a_h^{k,j,m} \notin \mathcal{A}_h^*(s) \right] \leq \sum_{k=1}^{k'} \sum_{j,m} \mathbb{I}\left[ a_h^{k,j,m} \notin \mathcal{A}_h^*(s_h^{k,j,m}) \right] \leq C_{\min}$$

due to Equation (9) in Lemma 5.1. $\qquad\square$

### G.4. Proof of Lemma 5.3

*Proof.* The event $\mathcal{E}_4$ holds with probability at least $1 - \delta$. Next we will prove Lemma 5.3 under the event $\mathcal{E}_4$.

If the trigger condition is met by the triple $(s, a, h)$ in round $k$, then we have $a = \pi_h^k(s)$. For such $(s, a, h)$, by $\mathcal{E}_4$ in Lemma G.1, it holds for any $s \in \mathcal{S}$, $h \in [H]$, $k \in [K]$ and $m \in [M]$ that

$$\sum_{j=1}^{J_k} \mathbb{I}\left[ s_h^{k,j,m} = s, a_h^{k,j,m} = a \right] = \sum_{j=1}^{J_k} \mathbb{I}\left[ s_h^{k,j,m} = s \right] \in \left[ J_k \mathbb{P}_{s,h}^k - \sqrt{24 J_k \mathbb{P}_{s,h}^k \iota'} - 9\iota', J_k \mathbb{P}_{s,h}^k + \sqrt{24 J_k \mathbb{P}_{s,h}^k \iota'} + 9\iota' \right]. \tag{63}$$

Since $(s, a, h)$ satisfies the trigger condition in round $k$, there exists an agent $m_0$ such that $n_h^{k,m_0}(s, a) = c_h^k(s, a)$. Then we reach

$$J_k \mathbb{P}_{s,h}^k + \sqrt{24 J_k \mathbb{P}_{s,h}^k \iota'} + 9\iota' \overset{\text{(I)}}{\geq} \frac{N_h^k(s, a)}{M H (H+1)} - 1 \overset{\triangle}{=} C_N > 199\iota'.$$

The last inequality is because $N_h^k(s, a) > i_1$. Solving the inequality (I), we can get the following relationship

$$\sqrt{J_k \mathbb{P}_{s,h}^k} \geq \sqrt{C_N - 3\iota'} - \sqrt{6\iota'}.$$

Then by Equation (63), for any other agent $m$,

$$\sum_{j=1}^{J_k} \mathbb{I}\left[ s_h^{k,j,m} = s, a_h^{k,j,m} = a \right] \geq J_k \mathbb{P}_{s,h}^k - \sqrt{24 J_k \mathbb{P}_{s,h}^k \iota'} - 9\iota' = \left( \sqrt{J_k \mathbb{P}_{s,h}^k} - \sqrt{6\iota'} \right)^2 - 15\iota'$$

$$\geq \left( \sqrt{C_N - 3\iota'} - 2\sqrt{6\iota'} \right)^2 - 15\iota' \geq \frac{C_N + 1}{3}.$$

The last inequality is because $C_N > 199\iota'$. Therefore, we have

$$n_h^k(s, a) = \sum_{m=1}^{M} n_h^{m,k}(s, a) = \sum_{m=1}^{M} \sum_{j=1}^{J_k} \mathbb{I}\left[ s_h^{k,j,m} = s, a_h^{k,j,m} = a \right] \geq \frac{M(C_N + 1)}{3} = \frac{N_h^k(s, a)}{3 H (H+1)},$$

and thus

$$N_h^{k+1}(s,a) = N_h^k(s,a) + n_h^k(s,a) \geq \left(1 + \frac{1}{3H(H+1)}\right) N_h^k(s,a).$$

$\square$

### G.5. Proof of Lemma 5.4

*Proof.* The event $\mathcal{G}_1 \cap (\bigcap_{i=1}^4 \mathcal{E}_i)$ holds with probability at least $1 - 5\delta$. Next we will prove Lemma 5.4 under the event $\mathcal{G}_1 \cap (\bigcap_{i=1}^4 \mathcal{E}_i)$. (For FedQ-Bernstein algorithm, we will prove Lemma 5.4 under the event $\mathcal{G}_2 \cap (\bigcap_{i=1}^4 \mathcal{E}_i)$.)

Under the event $\mathcal{G}_1 \cap (\bigcap_{i=1}^4 \mathcal{E}_i)$ (or $\mathcal{G}_2 \cap (\bigcap_{i=1}^4 \mathcal{E}_i)$), we have already proved the Lemma 5.1, Lemma 5.2 and Lemma 5.3.

For $P_{s,h}^* > 0$, the optimal action is unique. Then for any $(s,a,h)$ such that $a = \pi_h^*(s)$ and $P_{s,h}^* > 0$, we can simplify the results of Lemma 5.2 to the following equation

$$R_{k'}\mathbb{P}_{s,h}^* - 5\sqrt{R_{k'}\mathbb{P}_{s,h}^*\iota} - 32HC_{\min} \leq N_h^{k'+1}(s,a) \leq R_{k'}\mathbb{P}_{s,h}^* + 5\sqrt{R_{k'}\mathbb{P}_{s,h}^*\iota} + 32HC_{\min}. \tag{64}$$

By Equation (64), for any $s' \in \mathcal{S}$ and $h' \in [H]$ such that $\mathbb{P}_{s',h'}^* > 0$, we have

$$\frac{R_k\mathbb{P}_{s',h'}^* - 5\sqrt{R_k\mathbb{P}_{s',h'}^*\iota} - 32HC_{\min}}{R_{k-1}\mathbb{P}_{s',h'}^* + 5\sqrt{R_{k-1}\mathbb{P}_{s',h'}^*\iota} + 32HC_{\min}} \leq \frac{N_{h'}^{k+1}(s', \pi_{h'}^*(s'))}{N_{h'}^k(s', \pi_{h'}^*(s'))}.$$

To prove the second conclusion, we only need to prove that, for any $s' \in \mathcal{S}$ and $h' \in [H]$ such that $\mathbb{P}_{s',h'}^* > 0$,

$$\frac{R_k\mathbb{P}_{s',h'}^* - 5\sqrt{R_k\mathbb{P}_{s',h'}^*\iota} - 32HC_{\min}}{R_{k-1}\mathbb{P}_{s',h'}^* + 5\sqrt{R_{k-1}\mathbb{P}_{s',h'}^*\iota} + 32HC_{\min}} \geq 1 + \frac{1}{6H(H+1)}. \tag{65}$$

Next, we will prove the Equation (65). For the triple $(s_0, a_0, h_0)$, by Equation (64), we know that

$$\frac{6500H^3C_{\min}}{C_{st}} < N_{h_0}^k(s_0, a_0) \leq R_{k-1}\mathbb{P}_{s_0,h_0}^* + 5\sqrt{R_{k-1}\mathbb{P}_{s_0,h_0}^*\iota} + 32HC_{\min}.$$

Solving the inequality, we have

$$\begin{aligned}
\sqrt{R_{k-1}\mathbb{P}_{s_0,h_0}^*} &> \sqrt{\frac{6500H^3C_{\min}}{C_{st}} - 32HC_{\min} + \frac{25\iota}{4}} - \frac{5\sqrt{\iota}}{2} \\
&\overset{(I)}{>} \sqrt{\frac{6468H^3C_{\min}}{C_{st}}} - \sqrt{\frac{H^3C_{\min}}{C_{st}}} \\
&> 79\sqrt{\frac{H^3C_{\min}}{C_{st}}} > 79\sqrt{H^3C_{\min}}. 
\end{aligned} \tag{66}$$

and then

$$\sqrt{R_k\mathbb{P}_{s_0,h_0}^*} > \sqrt{R_{k-1}\mathbb{P}_{s_0,h_0}^*} > 79\sqrt{H^3C_{\min}}. \tag{67}$$

Here, the inequality (I) is because $\frac{25\iota}{4} < H^3C_{\min}$ for $H \geq 2$ and $0 < C_{st} \leq 1$. Therefore, for any $s' \in \mathcal{S}$ and $h' \in [H]$ such that $\mathbb{P}_{s',h'}^*$, we have

$$\sqrt{R_{k-1}\mathbb{P}_{s',h'}^*} = \sqrt{R_{k-1}\mathbb{P}_{s_0,h_0}^*} \cdot \sqrt{\frac{\mathbb{P}_{s',h'}^*}{\mathbb{P}_{s_0,h_0}^*}} \geq \sqrt{R_{k-1}\mathbb{P}_{s_0,h_0}^*} \cdot \sqrt{C_{st}} = 79\sqrt{H^3C_{\min}}, \tag{68}$$

and thus

$$\sqrt{R_k \overline{\mathbb{P}}^*_{s',h'}} > \sqrt{R_{k-1} \mathbb{P}^*_{s',h'}} > 79\sqrt{H^3 C_{\min}}. \tag{69}$$

For $X > 6241 H^3 C_{\min} = 79^2 H^3 C_{\min}$, note that

$$5\sqrt{X\iota} + 32 H C_{\min} \leq \sqrt{\frac{C_{\min}X}{H}} + 32 H C_{\min} \leq \frac{X}{56 H^2}. \tag{70}$$

Here, the first inequality is because $5\sqrt{\iota} < \sqrt{\frac{C_{\min}}{H}}$ for $H \geq 2$. Therefore, based on Equation (66), Equation (67), Equation (68) and Equation (69), we can apply Equation (70) for $R_{k-1}\mathbb{P}^*_{s_0,h}$ and $R_k \mathbb{P}^*_{s_0,h}$, $R_{k-1}\mathbb{P}^*_{s',h}$ and $R_k \mathbb{P}^*_{s',h}$ respectively:

$$5\sqrt{R_{k-1}\mathbb{P}^*_{s_0,h_0}\iota} + 32 H C_{\min} \leq \frac{R_{k-1}\mathbb{P}^*_{s_0,h_0}}{56 H^2}, \; 5\sqrt{R_k \mathbb{P}^*_{s_0,h_0}\iota} + 32 H C_{\min} \leq \frac{R_k \mathbb{P}^*_{s_0,h_0}}{56 H^2}. \tag{71}$$

and

$$5\sqrt{R_{k-1}\mathbb{P}^*_{s',h'}\iota} + 32 H C_{\min} \leq \frac{R_{k-1}\mathbb{P}^*_{s',h'}}{56 H^2}, \; 5\sqrt{R_k \mathbb{P}^*_{s',h'}\iota} + 32 H C_{\min} \leq \frac{R_k \mathbb{P}^*_{s',h'}}{56 H^2} \tag{72}$$

Since $N_h^k(s_0, a_0) > i_1$ and the trigger condition is satisfied by $(s, a, h)$ in round $k$, by Lemma 5.3, we have:

$$\frac{N_{h_0}^{k+1}(s_0, a_0)}{N_{h_0}^k(s_0, a_0)} \geq 1 + \frac{1}{3H(H+1)}.$$

Together with Equation (64), it holds that

$$\frac{R_k \mathbb{P}^*_{s_0,h_0} + 5\sqrt{R_k \mathbb{P}^*_{s_0,h_0}\iota} + 32 H C_{\min}}{R_{k-1}\mathbb{P}^*_{s_0,h_0} - 5\sqrt{R_{k-1}\mathbb{P}^*_{s_0,h_0}\iota} - 32 H C_{\min}} \geq \frac{N_{h_0}^{k+1}(s_0, a_0)}{N_{h_0}^k(s_0, a_0)} \geq 1 + \frac{1}{3H(H+1)}. \tag{73}$$

Applying Equation (71) to Equation (73), we have

$$1 + \frac{1}{3H(H+1)} \leq \frac{R_k \mathbb{P}^*_{s_0,h_0} + 5\sqrt{R_k \mathbb{P}^*_{s_0,h_0}\iota} + 32 H C_{\min}}{R_{k-1}\mathbb{P}^*_{s_0,h_0} - 5\sqrt{R_{k-1}\mathbb{P}^*_{s_0,h_0}\iota} - 32 H C_{\min}} \leq \frac{(1 + \frac{1}{56H^2})R_k}{(1 - \frac{1}{56H^2})R_{k-1}}.$$

Therefore, we know

$$\frac{R_k}{R_{k-1}} \geq \left(1 + \frac{1}{3H(H+1)}\right) \frac{1 - \frac{1}{56H^2}}{1 + \frac{1}{56H^2}}. \tag{74}$$

Using Equation (72), we have

$$\frac{R_k \mathbb{P}^*_{s',h'} - 5\sqrt{R_k \mathbb{P}^*_{s',h'}\iota} - 32 H C_{\min}}{R_{k-1}\mathbb{P}^*_{s',h'} + 5\sqrt{R_{k-1}\mathbb{P}^*_{s',h'}\iota} + 32 H C_{\min}} \geq \frac{(1 - \frac{1}{56H^2})R_k}{(1 + \frac{1}{56H^2})R_{k-1}} \geq \left(1 + \frac{1}{3H(H+1)}\right)\left(\frac{1 - \frac{1}{56H^2}}{1 + \frac{1}{56H^2}}\right)^2. \tag{75}$$

The last inequality is by Equation (74). Let

$$c = \frac{1 - \sqrt{\frac{6H^2+6H+1}{6H^2+6H+2}}}{1 + \sqrt{\frac{6H^2+6H+1}{6H^2+6H+2}}}.$$

Then we have

$$c = \frac{1}{6H^2 + 6H + 2} \cdot \left(\frac{1}{1 + \sqrt{\frac{6H^2+6H+1}{6H^2+6H+2}}}\right)^2 > \frac{1}{4(6H^2 + 6H + 2)} \geq \frac{1}{56H^2},$$

and thus

$$\frac{1 + \frac{1}{6H(H+1)}}{1 + \frac{1}{3H(H+1)}} = \frac{6H^2 + 6H + 1}{6H^2 + 6H + 2} = \left(\frac{1-c}{1+c}\right)^2 \leq \left(\frac{1 - \frac{1}{56H^2}}{1 + \frac{1}{56H^2}}\right)^2.$$

Applying this inequality to Equation (75) completes the proof of Equation (65), thereby proving the second conclusion. □

### G.6. Details of Final Discussion

The event $\mathcal{G}_1 \cap (\bigcap_{i=1}^4 \mathcal{E}_i)$ (or $\mathcal{G}_2 \cap (\bigcap_{i=1}^4 \mathcal{E}_i)$) holds with probability at least $1 - 5\delta$. Under the event $\mathcal{G}_1 \cap (\bigcap_{i=1}^4 \mathcal{E}_i)$ (or $\mathcal{G}_2 \cap (\bigcap_{i=1}^4 \mathcal{E}_i)$), we have proved Lemma 5.3 and Lemma 5.4. Next, we will discuss the number of communication rounds and consider four different situations:

1. In round $k$, the trigger condition is satisfied by $(s, a, h)$ when $N_h^k(s, a) \leq i_1$. We will refer to this as a Type-I trigger.

   For each time the trigger condition is met for $(s, a, h)$, the number of visits to $(s, a, h)$ increases by at least $1/(2MH(H+1))$ times. Specifically, when the trigger condition is first satisfied, the visit number increases from 0 to at least 1. Therefore, the maximum number of Type-I triggers for each triple $(s, a, h)$, denoted $t_2(s, a, h)$, satisfies

   $$\left(1 + \frac{1}{2MH(H+1)}\right)^{t_1(s,a,h)-2} \leq i_1.$$

   Therefore, we have

   $$t_1(s, a, h) \leq \frac{\log(i_1)}{\log(1 + \frac{1}{2MH(H+1)})} + 2 = O(MH^2 \log(i_1)).$$

   and thus the number of rounds with Type-I triggers is bounded by

   $$\sum_{s,a,h} t_1(s, a, h) \leq O\left(MH^3 SA \log(i_1)\right). \tag{76}$$

2. In round $k$, the triple $(s, a, h)$ satisfies the trigger condition when $i_1 < N_h^k(s, a) < i_1 + i_2$. We will refer to this as a Type-II trigger if $a \notin \mathcal{A}_h^*(s)$ or $a \in \mathcal{A}_h^*(s)$ and $\mathbb{P}_{s,h}^* = 0$, and as a Type-III trigger if $a \in \mathcal{A}_h^*(s)$ and $\mathbb{P}_{s,h}^* > 0$.

   By Lemma 5.3, for each time the trigger condition is satisfied by $(s, a, h)$, the number of visits to $(s, a, h)$ increases by at least $1/3H(H+1)$ times.

   For $(s, a, h)$ satisfying the type-II trigger, by Equation (9) in Lemma 5.1 and Lemma 5.2, we know that the maximum visit number to $(s, a, h)$ is $32HC_{\min}$. Therefore, the maximum number of Type-II triggers for each triple $(s, \pi_h^*(s), h)$, denoted $t_2(s, a, h)$, satisfies

   $$\left(1 + \frac{1}{3H(H+1)}\right)^{t_2(s,a,h)-1} \leq \frac{32HC_{\min}}{i_1} \leq O\left(\frac{MH^7 SA\iota}{MH^2\iota'\Delta_{\min}^2}\right) = O\left(\frac{H^5 SA}{\Delta_{\min}^2}\right).$$

   Therefore, we have

   $$t_2(s, a, h) \leq \frac{\log\left(\frac{H^5 SA}{\Delta_{\min}^2}\right)}{\log\left(1 + \frac{1}{3H(H+1)}\right)} + 1 = O\left(H^2 \log\left(\frac{H^5 SA}{\Delta_{\min}^2}\right)\right).$$

   and thus the number of rounds with Type-II triggers is bounded by

   $$\sum_{s,a,h} t_2(s, a, h) \leq O\left(H^3 SA \log\left(\frac{H^5 SA}{\Delta_{\min}^2}\right)\right). \tag{77}$$

3. By condition (b) of Definition 3.2, we know $a = \pi_h^*(s)$ for a Type-III trigger. Therefore, the maximum number of Type-III triggers for each triple $(s, \pi_h^*(s), h)$, denoted $t_3(s, \pi_h^*(s), h)$, satisfies

   $$\left(1 + \frac{1}{3H(H+1)}\right)^{t_3(s,\pi_h^*(s),h)-1} \leq \frac{i_1 + i_2}{i_1} \leq i_2.$$

Therefore, we have

$$t_3(s, \pi_h^*(s), h) \leq \frac{\log(i_2)}{\log\left(1 + \frac{1}{3H(H+1)}\right)} + 1 = O(H^2 \log(i_2)).$$

Because we only have $HS$ triples of $(s, \pi_h^*(s), h)$, the number of rounds with Type-III triggers is bounded by

$$\sum_{s, \mathbb{P}_{s,h}^* > 0} t_3(s, \pi_h^*(s), h) \leq O\left(H^3 S \log(i_2)\right). \tag{78}$$

4. The trigger condition is satisfied by $(s, a, h)$ in round $k$ when $N_h^k(s, a) > i_1 + i_2$.

   By Lemma 5.3, in this case, for each time the trigger condition is satisfied by $(s, a, h)$, we have $a \in \mathcal{A}_h^*(s)$. we will first prove the trigger condition cannot be satisfied by $(s, a, h)$ in round $k$ when $a \in \mathcal{A}_h^*(s)$, $\mathbb{P}_{s,h}^* = 0$ and $N_h^k(s, a) > i_1 + i_2$.

   Let $\mathcal{S}_0 = \{(s, a, h) \mid a \in \mathcal{A}_h^*(s), \mathbb{P}_{s,h}^* = 0\}$. By Lemma 5.2, we know for $(s, a, h) \in \mathcal{S}_0$, $N_h^{K+1}(s, a) \leq 32HC_{\min} < i_1 + i_2$. However, when the trigger condition is satisfied by $(s, a, h)$ in round $k$, we have $N_h^k(s, a) > i_1 + i_2$, which is contradicts the fact that $N_h^{K+1}(s, a) < i_1 + i_2$. Therefore the triple $(s, a, h)$ satisfies that $\mathbb{P}_{s,h}^* > 0$. Then by Lemma 5.4, for any $s' \in \mathcal{S}$ and $h' \in [H]$ such that $\mathbb{P}_{s',h'}^* > 0$, it holds that

   $$N_{h'}^{k+1}(s', \pi_{h'}^*(s')) \geq \left(1 + \frac{1}{6H(H+1)}\right) N_{h'}^k(s', \pi_{h'}^*(s')),$$

   indicating that the number of visits to $(s', \pi_h^*(s'), h')$ with $\mathbb{P}_{s',h'}^* > 0$ simultaneously increases by at least $1/6H(H+1)$ times. We refer to this type of trigger as Type-IV trigger. Therefore, the maximum number of Type-IV triggers, denoted $t_4$, satisfies

   $$\left(1 + \frac{1}{6H(H+1)}\right)^{t_4} \leq \frac{\hat{T}}{i_1 + i_2} \leq \frac{T}{HSA}.$$

   The last inequality is because $i_2 > MHSA$. Therefore, the number of rounds with Type-III triggers is bounded by

   $$t_4 \leq \frac{\log(\frac{T}{HSA})}{\log\left(1 + \frac{1}{6H(H+1)}\right)} = O\left(H^2 \log\left(\frac{T}{HSA}\right)\right). \tag{79}$$

By Equation (76), Equation (77), Equation (78) and Equation (79), the number of rounds is no more than

$$\sum_{s,a,h} t_1(s, a, h) + \sum_{s,a,h} t_2(s, a, h) + \sum_{s, \mathbb{P}_{s,h}^* > 0} t_3(s, \pi_h^*(s), h) + t_4$$

$$\leq O\left(MH^3 SA \log(i_1) + H^3 SA \log\left(\frac{H^5 SA}{\Delta_{\min}^2}\right) + H^3 S \log(i_2) + H^2 \log\left(\frac{T}{HSA}\right)\right)$$

$$\leq O\left(MH^3 SA \log\left(MH^2 \iota'\right) + H^3 SA \log\left(\frac{H^5 SA}{\Delta_{\min}^2}\right) + H^3 S \log\left(\frac{MH^9 SA\iota}{\Delta_{\min}^2 C_{st}}\right) + H^2 \log\left(\frac{T}{HSA}\right)\right).$$

The last inequality is because $i_2 \lesssim \frac{MH^9 SA\iota}{\Delta_{\min}^2 C_{st}}$. By (e) of Lemma E.1, we have

$$\iota' = \log\left(\frac{2MSAHT_1}{\delta}\right) \leq O\left(\log\left(\frac{2MSAH\hat{T}}{\delta}\right) + \log\left(\frac{2MSAH}{\delta}\right)\right) = O\left(\log\left(\frac{SA\hat{T}}{\delta}\right)\right). \tag{80}$$

Let $\delta = p/5$ and $\iota_0 = \log\left(\frac{MSAT}{p}\right)$. Since $\iota \leq \iota' \leq O(\iota_0)$ by Equation (80), then with probability at least $1 - p$, the number of rounds of communication is no more than

$$O\left(MH^3 SA \log\left(MH^2 \iota_0\right) + H^3 SA \log\left(\frac{H^5 SA}{\Delta_{\min}^2}\right) + H^3 S \log\left(\frac{MH^9 SA\iota}{\Delta_{\min}^2 C_{st}}\right) + H^2 \log\left(\frac{T}{HSA}\right)\right).$$

