# OpenReview forum: "Gap-Dependent Bounds for Federated $Q$-Learning"
_ICML.cc/2025/Conference — ICML 2025 poster_

### Official Review · Reviewer_Cred · 2025-03-11

**Overall Recommendation:** 4

**Summary:**

This work studies instance-dependent regret guarantees in tabular Markov Decision Processes within a federated learning framework, where multiple agents collaborate to learn the environment. Specifically, the author focuses on the minimal sub-optimality gap structure and provides the first logarithmic regret guarantee in federated reinforcement learning. Additionally, compared to the previous communication cost for $\sqrt{T}$-type regret, the proposed algorithm achieves a lower communication cost by seperating the dependency on the size of the state-action space and the log-dependency on the number of episode $T$.

**Claims And Evidence:**

The author provides a clear claim of the result in the theorems and includes a proof sketch to outline the key steps of the theoretical analysis.

**Essential References Not Discussed:**

This paper provides a comprehensive discussion of related work in federated reinforcement learning and batch reinforcement learning. However, it would be beneficial to also consider "A Simple and Provably Efficient Algorithm for Asynchronous Federated Contextual Linear Bandits," which is relevant to the setting of federated learning.

**Experimental Designs Or Analyses:**

The main contribution of this work focuses on the theoretical analysis of regret, with the simulation serving as an auxiliary tool to support its efficiency, which is reasonable and sufficient. Therefore, there is no need to evaluate the soundness or validity of the experimental designs or analyses.

**Methods And Evaluation Criteria:**

The main contribution of this work focuses on the theoretical analysis of regret, with the simulation serving as an auxiliary tool to support its efficiency, which is reasonable and sufficient.

**Other Comments Or Suggestions:**

In the main paper, it would be beneficial to provide an explicit algorithm and clearly outline how it progresses in each step. This would improve readability and help readers better understand the methodology.

**Other Strengths And Weaknesses:**

No other strengths and weaknesses.

**Questions For Authors:**

As discussed in Theoretical Claims, it is important to clarify what specific factors lead to the continuous improvement that further removes the dependency on the state space size $S$.

**Relation To Broader Scientific Literature:**

This work mainly focuses on providing an instance-dependent guarantee for federated reinforcement learning, which may be related to other works on federated learning.

**Theoretical Claims:**

The theoretical analysis primarily focuses on two aspects: the regret guarantee and the communication complexity.

For the regret guarantee, the logarithmic regret is achievable under the assumption of a minimal sub-optimality gap, and the proof sketch aligns with previous analyses of logarithmic regret but is adapted to a more complex federated learning framework.

For the communication complexity, achieving logarithmic communication complexity is reasonable with a predefined trigger criterion in federated Q-learning. However, I have concerns about the claimed improvement that separates the dependency on the size of the state-action space and the log-dependency on the number of episodes $T$. As discussed in the proof sketch, the improvement comes from the assumption of a minimal sub-optimality gap, which ensures that sub-optimal actions are taken only $O(\log T)$ times. However, a similar motivation was used in [1] to reduce the dependency on the action space size $A$, since each state has a unique optimal action. It is important to clarify what specific factors lead to the continuous improvement that further removes the dependency on the state space size $S$.

[1] Gap-Dependent Bounds for Q-Learning using Reference-Advantage Decomposition

---

> ### Author Rebuttal · Authors · 2025-04-01
>
> We thank you for your careful reading and thoughtful comments. Below are our point-by-point responses to your questions, and we hope our responses address your concerns.
>
> **Theoretical Claim**: Specific factors lead to the continuous improvement
>
> **Response**: Thanks for this insightful comment. We agree that it is important to clarify this improvement.
>
> To begin with, it is worth pointing out that [1] only removes the dependence on $A$ only for MDPs with unique optimal actions. In contrast, our work removes the dependence on $M,S,A$ from the $\log T$ term in **general MDPs (G-MDPs)**, which include MDPs with unique optimal actions as a special case. Both [1] and our paper exploit the common discrepancy between the numbers of visits to optimal triples and suboptimal triples. This allows the number of synchronizations (or policy switches) triggered by suboptimal triples to be bounded by the $\log \log T$ terms that still depend on $S$ and $A$.
>
> Our **key improvement** is in further removing the dependency on $M, S$ for the $(\log T)-$term that bounds the number of synchronization triggered by optimal triples in late rounds. This is achieved through new technical insights. In FRL, synchronization is triggered when one triple meets the termination condition (Eq. (4)). Unlike [1], our analysis carefully controls synchronization triggered by optimal triples in late rounds, which is crucial for tightening the $\log T$ dependence.
>
> Specifically, our Lemmas 5.3 and 5.4 show that, when an optimal triple triggers synchronization in late rounds, the number of visits to all optimal triples across all agents grows exponentially. This simultaneous increase allows us to eliminate $M,S,A$ from the $\log T$-term in the count of trigger events and therefore in the number of communication rounds as well. Technically, this relies on a novel concentration result (Lemma 5.2) using error recursions from visits to suboptimal triples without assuming any stationary visitation distribution for the implemented policies. To our knowledge, this approach is novel in the RL literature.
>
> We also remark that removing the $S$-dependency in [1] is impossible due to its unaligned stage design, which prevents the simultaneous updating of policies for all state-action pairs.
>
> **References Not Discussed**: a relevant work
>
> **Response**: We agree that this is a relevant reference to the federated setting and will add it in our revised version.
>
> **Suggestion**: Provide an explicit algorithm
>
> **Response**: Due to page limits, we give an overview of the algorithm in Section 3.1 of the main paper and then provide the full algorithm in Appendix C.1. For clarity and ease of understanding, we also follow your suggestion to outline each step as follows:
>
> **Step 1: Coordination.** At the beginning of round $k$, the server broadcasts $\pi^k = \\{\pi_h^k\\}\_{h=1}^H$, $\\{N_h^k(s,\pi_h^k(s))\\}\_{s,h}$ and $\\{V_h^k(s)\\}_{s,h}$ to all of the agents. When $k=1$, $N_h^1(s,a) = 0,Q_h^1(s,a) = V_h^1(s)=H,\forall (s,a,h)$ and $\pi^1$ is an arbitrary deterministic policy. Once receiving such information, the agents will execute policy $\pi^k$ and start collecting trajectories.
>
> **Step 2: Event-Triggered Termination.** During exploration, every agent $m$ will monitor $n_h^{m,k}(s,a)$, which is the total number of visits for $(s,a,h)$ in round $k$. For any agent $m$, at the end of each episode, if any $(s,a,h)$ has been visited by $\max\left\\{1,\lfloor \frac{1}{MH(H+1)} N_h^k(s,a) \rfloor \right\\}$ times by agent $m$, the agent will send a signal to the server, which will then request all agents to abort the exploration.
>
> **Step 3: Local Aggregation.** After the exploration is terminated, each agent updates the local estimate of the expected return $v_{h+1}^{m,k}(s,a)$:$$v_{h+1}^{m,k}(s,a)=\frac{1}{n_h^{m,k}(s,a)}\sum_{j=1}^{n^{m,k}}V_{h+1}^k\left(x_{h+1}^{m,k,j}\right)\mathbb{I}[\{(x_h^{m,k,j},a_h^{m,k,j})=(s,a)\}],\forall (s,a,h).$$
> Next, each agent $m$ sends $\\{r_h(s,\pi_h^k(s))\\}\_{s,h}$, $\\{n_h^{m,k}(s,\pi_h^k(s))\\}\_{s,h}$ and $\\{v_{h+1}^{m,k}(s,\pi_h^k(s))\\}\_{s,h}$ to the central server.
>
> **Step 4: Server-side Aggregation.** After receiving the information sent by the agents, the central server will update the value functions $Q_h^{k+1}(s,a)$ and $V_h^{k+1}(s)$, as well as the policy $\pi^{k+1}$, as shown in Appendix C.1.
>
> The algorithm then proceeds to round $k+1$.

---

> > ### Comment · Reviewer_Cred · 2025-04-03
> >
> > Thank you for the clarification. Regarding the further improvement of the dependence on the factors $M$ and $S$, I still have some concerns. Based on the rebuttal, the improvement appears to come from the fact that the visitation frequencies of all optimal state-action-state triples grow at similar rates. In the gap-dependent analysis, most actions incur zero regret, as long as the agent selects one of the optimal actions—thus eliminating dependence on $S$ in such cases.
> >
> > However, it’s unclear how this mechanism behaves when there are **multiple optimal actions for a given state**. Specifically, it seems possible that the agent may consistently follow a different optimal action than the one prescribed by $\pi_h^*(s)$ as defined in Lemma 5.4. If so, the result in this lemma may not hold.
> >
> > It would be helpful if the authors could provide further explanation or clarification regarding this aspect.

---

> > > ### Author Response · Authors · 2025-04-03
> > >
> > > Thank you for the thoughtful follow-up. The question of uniqueness in optimal actions is important, and we appreciate the opportunity to further clarify this aspect.
> > >
> > > In our paper, we prove the gap-dependent communication cost under the **G-MDP** setting (see Definition 3.2), which assumes:
> > >
> > > 1. For any state-step pair $(s,h)$, the stationary visiting probabilities under optimal policies are unique and they are denoted as $\mathbb{P}_{s,h}^\star$.
> > >
> > > 2. For any state-step pair $(s,h)$ with positive stationary visiting probability $\mathbb{P}_{s,h}^\star>0$ under optimal policies, the optimal action is unique.
> > >
> > > Importantly, for state-step pairs with $\mathbb{P}\_{s,h}^\star = 0$ (i.e., those that are never visited under any optimal policy), we allow **multiple optimal actions**. These include state-step pairs that are either unreachable or only visited under policies involving suboptimal actions. G-MDPs are thus **more general** than MDPs that require the uniqueness of all optimal actions. From a technical standpoint, the number of visits to such unreachable or rarely visited pairs can be upper bounded by $O(\frac{\mbox{poly}(MHSA)\log T}{\Delta_{\min}})$ (see Lemma 5.2) because any episode where such a pair appears will necessarily involve a suboptimal action (see Lemma G.2). Thus, the non-uniqueness of optimal actions at such pairs does not affect the late-round simultaneous increase for optimal triples with $\mathbb{P}_{s,h}^\star > 0$ (Lemma 5.4).
> > >
> > > However, **if the G-MDP assumption does not hold** (i.e., there exists some $(s,h)$ such that agents can visit it under some optimal policy and $\arg \max_{a'}Q_h^\star(s,a')$ is non-unique), then the simultaneous increase argument may fail. In this situation, optimism-based algorithms like FedQ-Hoeffding or UCB-Advantage [1] cannot guarantee consistent selection of a single optimal action, and neither can remove the $SA-$dependence in the $\log T-$term.
> > >
> > > To illustrate, suppose that $a_1\neq a_2$ and both $a_1,a_2$ are optimal for $(s,h)$ satisfying that $Q_{h}^\star(s,a_1) = Q_{h}^\star(s,a_2) = \max_{a'} Q_{h}^\star(s,a')$. FedQ-Hoeffding and UCB-Advantage in [1] are all optimism-based methods so that $Q_{h'}^k(s',a')> Q_{h'}^\star(s',a'),\forall (s',a',h',k)$ with high probability. If the policy at $(s,h)$ is fixed at $a_1$ so that $(s,a_1,h)$ is constantly visited, $Q_h^k(s,a_1,h)$ will converge to $\max_{a'} Q_{h}^\star(s,a')$ and fall below the unchanged $Q_h^k(s,a_2,h)$ at some time. Since the policy is determined by maximizing the estimated $Q$ function, policy switching or synchronization will inevitably happen. This implies that both optimal actions can trigger policy switching or synchronization in late rounds. Consequently, considering the general situation where multiple optimal actions exist, due to the swithcing of optimal actions, the simultaneous increase may not hold, making it difficult to remove the dependency on $A$. The dependency on $S$ might also appear due to the different transition dynamics that result from the different optimal actions at $(s,h)$.
> > >
> > > In summary, G-MDPs are **more general** than MDPs requiring the uniqueness of all optimal actions. However, **without the G-MDP structure**, it becomes difficult to eliminate the $SA-$dependence in the $\log T-$term. Addressing this more general setting may require new algorithm designs or prior knowledge of the minimum gap $\Delta_{\min}$, and we view this as an exciting direction for future work.

---

### Official Review · Reviewer_A3ZH · 2025-03-11

**Overall Recommendation:** 4

**Summary:**

The paper presents a gap-dependent analysis of regret and communication cost for on-policy Federated Q-Learning method. It improves prior worst-case regret bounds that scale as $\sqrt{T}$ by $log⁡T$. The paper introduces a refined communication cost bound that distinguishes between exploration and exploitation, leading to improved efficiency. The results offer a multi-agent speedup pattern and remove the dependence of the communication cost on MSA in the logT term.

## update after rebuttal
I appreciate the authors' detailed responses. My questions and concerns have been addressed, and I will keep my score as it is.

**Claims And Evidence:**

The paper claims to improve the theoretical bounds for FedQ-Hoeffding from prior work (Zheng, 2024), specifically providing gap-dependent bounds for regret and communication cost. Theoretical analysis supports the claims.
It also includes numerical experiments in the appendix that illustrate the theoretical dependencies of communication costs on various parameters.

**Essential References Not Discussed:**

No critical references appear to be missing. However, a more structured comparison with existing methods (e.g., via a table) would improve clarity (see below).

**Experimental Designs Or Analyses:**

As this is a theoretical study, no extensive experimental validation is expected.

**Methods And Evaluation Criteria:**

This is a theoretical study, and the methods used for evaluation rely entirely on mathematical proofs and complexity analysis. Given that the objective of this work is improving theoretical bounds, this approach is appropriate. The numerical experiments presented in the appendix serve to validate theoretical insights.

**Other Comments Or Suggestions:**

Technical novelty and contributions section could be rewritten to be more accessible to readers who are not experts in this specific area.

**Other Strengths And Weaknesses:**

Strengths:
Provides a meaningful theoretical advancement in the regret analysis of federated Q-learning.
Addresses an open problem regarding gap-dependent regret bounds in on-policy FRL.
Weaknesses (Clarity):
Providing a more intuitive explanation or a high-level overview before diving into technical details could improve readability.
The discussion of related work in the introduction is relatively unstructured; listing prior results in paragraphs makes it difficult to compare methods effectively. A table summarizing related work, regret bounds, and key assumptions would be helpful to compare and categorize the works cited.

**Questions For Authors:**

The paper assumes strictly positive suboptimality gaps. How would the results change if some actions had near-zero gaps? Would the bounds degrade?

**Relation To Broader Scientific Literature:**

The paper fits within the broader context of federated reinforcement learning. It builds on prior work (Zheng, 2024) by introducing improved theoretical guarantees.

**Theoretical Claims:**

While I did not verify the proof steps in detail, the results appear consistent with numerical experiments.

---

> ### Author Rebuttal · Authors · 2025-04-01
>
> We thank you for your careful reading and thoughtful comments. Below are our point-by-point responses, and we hope our responses address your concerns.
>
> **Weakness 1**: Improve readability of technical details
>
> **Response**: Thanks for this helpful advice. To improve readability, we now provide a high-level explanation before diving into technical details.
>
> **Regret (High-Level Intuition)**:
>
> FedQ-Hoeffding is an optimism-based algorithm that guarantees $Q_h^k(s,a)\geq Q_h^\star(s,a)$ for all $(s,a,h,k)$. Under MDPs with a nonzero minimum suboptimality gap $\Delta_{\min}$, a suboptimal action $a$ is chosen at step $h$ for state $s$ only if the estimated value function satisfies $Q_h^k(s,a) = \max_{a'} Q_h^k(s,a') \geq \max_{a'} Q_h^\star(s,a')\geq Q_h^\star(s,a) + \Delta_{\min}$. This implies that only large estimation errors $Q_h^k-Q_h^*$ (at least $\Delta_{\min}$) contribute to the regret. Our proof formalizes this intuition. Lemma 4.2 links such estimation errors to the expected regret, and Lemma 4.1 controls the estimation errors via recursions over steps.
>
> **Communication Cost (High-Level Intuition)**:
>
> In FedQ-Hoeffding, communication rounds are triggered only when a specific termination condition (Eq. (4)) is satisfied for some triple $(s, a, h)$. Thus, the total number of rounds is upper-bounded by the number of times this condition is triggered. The trigger condition ensures an exponential increase in the number of visits to the triple $(s, a, h)$ that triggers the condition.
>
> Compared to worst-case MDPs, the suboptimality gap helps identify the discrepancy between the number of visits to non-optimal and optimal triples: for non-optimal triples, the number of visits scales as $\log T$ (Lemma 5.1), which is much smaller than the average sample size $T$ and results in $\log \log T$ trigger events with linear dependence on $M,S,A$; and for optimal triples, Lemmas 5.3 and 5.4 establish that if a trigger occurs for **one** optimal triple in the later rounds, it guarantees a simultaneous exponential increase in the number of visits to **all** optimal triples across all $M$ agents. This is the key insight that allows us to eliminate the dependency of $M,S,A$ from the $\log T$-term in the count of trigger events and therefore in the number of communication rounds as well.
>
> **Weakness 2**: Summary of related work
>
> **Response**: Thanks for this helpful suggestion. To improve clarity and facilitate comparison, we now follow your advice to include a structured summary of related works in federated RL in the table below. The table highlights whether each work provides gap-dependent bounds, the corresponding regret bound, and the number of communication rounds. For simplicity, $O^*$ hides lower-order terms compared to $\log T$, and $f_M \in \\{1,M\\}$.
>
> | Algorithm            | Gap-dependent | Regret bound                                         | Number of rounds                  |
> |--------|-------|--------------|-------------|
> | Byzan-UCBVI (Chen et al., 2022)        | ×             | $\tilde{O}(\sqrt{MH^3S^2AT})$                        | $O(MHSA\log T)$                   |
> | FedQ-Hoeffding (Zheng et al., 2024)    | ×             | $\tilde{O}(\sqrt{MH^4SAT})$                          | $O(MH^3SA\log T)$                 |
> | FedQ-Bernstein (Zheng et al., 2024)    | ×             | $\tilde{O}(\sqrt{MH^3SAT})$                          | $O(MH^3SA\log T)$                 |
> | FedQ-Advantage (Zheng et al., 2025a)   | ×             | $\tilde{O}(\sqrt{MH^2SAT})$                          | $O(f_MH^2SA(\log H)\log T)$       |
> | Fed-UCBVI (Labbi et al., 2025)         | ×             | $\tilde{O}(\sqrt{MH^2SAT})$                          | $O^*(HSA\log T)$                  |
> | Our work                               | ✓             | $O^*\left(\frac{H^6SA\log(MSAT)}{\Delta_{\min}}\right)$ | $O^*(H^2\log T)$                  |
>
> **Suggestion**: Rewrite the “Technical Novelty and Contributions" section
>
> **Response**: In the revised version, we will incorporate the high-level intuition from our response to Weakness 1 into the Technical Novelty and Contributions section to make it more accessible to readers less familiar with this area.
>
> **Question**: Near-zero suboptimality gap
>
> **Response**: In finite-horizon tabular episodic MDPs, since the numbers of states, actions, and steps are finite, the minimum gap $\Delta_{\min} = \inf\\{\Delta_h(s,a) \mid \Delta_h(s,a)>0,(s,a,h)\in \mathcal{S}\times \mathcal{A}\times [H]\\}$ is positive as long as at least one suboptimal action exists, that is, $\\{\Delta_h(s,a) \mid \Delta_h(s,a)>0,(s,a,h)\in \mathcal{S}\times \mathcal{A}\times [H]\\} \neq \emptyset$. Otherwise, all actions are optimal (i.e., the set above is empty), leading to a degenerate MDP. Therefore, in all non-degenerate tabular MDPs, which are the focus of our setting, $\Delta_{\min}>0$, ensuring that our gap-dependent regret bound remains valid and does not degrade, even if some suboptimality gaps are very small.

---

### Official Review · Reviewer_vCHq · 2025-03-14

**Overall Recommendation:** 3

**Summary:**

This paper studies the gap-dependent bounds for online federated Q-learning where multiple agents collaboratively explore and learn a global Q-function under tabular MDP setting. The authors provide analysis of both performance regret and communication cost of the online federated Q-learning algorithm (FedQ-Hoeffding), proposed in the previous work, in terms of gap-dependent parameters.

**Claims And Evidence:**

Yes

**Essential References Not Discussed:**

NA

**Experimental Designs Or Analyses:**

Yes

**Methods And Evaluation Criteria:**

Yes

**Other Comments Or Suggestions:**

See questions below

**Other Strengths And Weaknesses:**

Strengths
- This work offers the first gap-dependent regret bound and communication cost analysis for online Q-learning in federated environments.

Weaknesses
- The contribution appears somewhat incremental, as the primary result stems from applying existing gap-dependent regret analysis [2,3] to a previously established algorithm [1].

**Questions For Authors:**

1. The algorithm in Section 3.1 appears nearly identical to the one in [1]. I'm curious whether any modifications were made to hyper-parameters or communication triggering mechanisms specifically for the gap-dependent analysis. If changes exist, highlighting the differences between this version of FedQ-Hoeffding and the one in [1] would be valuable. Also, I'm curious whether the algorithm can be more efficient if we have prior knowledge of the suboptimality gap.
2. Given the other gap-independent terms (C_f in (1)) in the regret bound, it seems that the algorithm can benefit from speedup in terms of number of agents when the suboptimality gap is sufficiently small. Could you provide intuition for why agents cannot enjoy the advantages of collaboration when the gap is large?

**Relation To Broader Scientific Literature:**

The key contribution of this paper is providing the gap-dependent regret bound of existing FedQ-Hoeffding, proposed in the previous work [1] with worst-case \sqrt{T}-type regret bound.  Gap-dependent regret analysis for on-policy Q-learning has been studied in [2,3] for single-agent Q-learning case, and this work extends the gap-dependent analysis for FedQ-Hoeffding in federated settings.

[1] Zheng, Z., Gao, F., Xue, L., and Yang, J. Federated qlearning: Linear regret speedup with low communication cost. In The Twelfth International Conference on Learning Representations, 2024

[2] Yang, K., Yang, L., and Du, S. Q-learning with logarithmic regret. In International Conference on Artificial Intelligence and Statistics, pp. 1576–1584. PMLR, 2021.

[3] Zheng, Z., Zhang, H., and Xue, L. Gap-dependent bounds for q-learning using reference-advantage decomposition. In The Thirteenth International Conference on Learning Representations, 2025b.

**Theoretical Claims:**

Yes

---

> ### Author Rebuttal · Authors · 2025-04-01
>
> We thank you for your careful reading and thoughtful comments. Below are our point-by-point responses, and we hope our responses address your concerns.
>
> **Weakness**: Contribution of the work
>
> **Response**: We respectfully disagree that our main result is merely an application of existing gap-dependent regret analyses [2, 3] that only focus on the single-agent setting. Our contributions go beyond adapting known techniques: we establish the first gap-dependent bounds on both regret and communication cost for federated RL (FRL) in the literature, and we develop new theoretical tools that are broadly applicable to other on-policy RL and FRL methods.
>
> Below, we clarify our contributions in two parts.
>
> First, we explain new challenges and techniques in controlling the estimation error of value functions to prove **the first gap-dependent regret bound in FRL**. In contrast to the single-agent setting, FRL introduces additional regret due to delayed policy switching, which is necessary to reduce communication frequency. To overcome this challenge, we develop new recursions (see Eq. (35)) for bounding the additional regret in the weighted sum of value estimation errors across steps.
>
> Additionally, our regret bound includes a multi-agent burn-in cost $C_f$. Applying the techniques of [2, 3] directly, this term would incur the undesired dependence on the gap $\Delta_{\min}$. We remove this dependency through a new delayed integration argument that carefully analyzes error recursion between adjacent steps (see Eq. (35) and Eq. (41)), which is not found in prior works.
>
> Second, we also establish **the first gap-dependent communication cost in FRL**. Our result (Eq. (2)) improves upon [1] by removing the dependency on $M,S,A$ from the $\log T$ term, which presents a substantial advance for the on-policy RL literature. This improvement is enabled by new technical insights. In FRL, synchronization is triggered when one triple meets the termination condition (Eq. (4)). Unlike [1], our analysis carefully controls synchronization triggered by optimal triples in late rounds, which is crucial for tightening the $\log T$ dependence.
>
> Specifically, our Lemmas 5.3 and 5.4 show that, when an optimal triple triggers synchronization in late rounds, the number of visits to all optimal triples across all agents grows exponentially. This simultaneous increase allows us to eliminate the dependency of $M,S,A$ from the $\log T$-term in the count of trigger events and therefore in the number of communication rounds as well. Technically, this relies on a novel concentration result (Lemma 5.2) using error recursions from visits to suboptimal triples without assuming any stationary visitation distribution for the implemented policies. To our knowledge, this approach is novel in the RL literature.
>
> **Question 1**: the algorithm in Section 3.1
>
> **Response**: Yes, the algorithm we analyze is identical to the one in [1]. We only give a brief overview in the main paper, and the details are provided in Appendix C.1. Our contribution lies in the new analysis that yields the first gap-dependent bounds for FRL.
>
> As for prior knowledge of the suboptimality gap, we agree this is an interesting direction. Incorporating such knowledge could potentially lead to more communication-efficient algorithms and may offer insights into reward design and its impact on learning efficiency. However, existing literature typically assumes the gap is unknown, and we follow that convention.
>
> **Question 2**: Speedup
>
> **Response**: The linear dependence on $M$ in the burn-in cost $C_f$ arises from the high estimation error in early rounds. This burn-in phase is necessary for coordinating policy updates and is common in FRL (e.g., [1, 4, 5]). Since the optimal value functions satisfy $V_h^*(s)\leq H$ for any $(s,h)$, the minimum gap $\Delta_{\min}$ is upper bounded by $H$. When the total number of steps $T$ for each agent is sufficiently large, our gap-dependent bound becomes $O\left(\frac{H^6SA\log(MSAT)}{\Delta_{\min}}\right)$, which can enjoy a near-linear multi-agent speedup in the average regret for each episode.
>
> [4] Zhong Zheng, Haochen Zhang, and Lingzhou Xue. Federated Q-learning with reference-advantage decomposition: almost optimal regret and logarithmic communication cost. ICLR, 2025.
>
> [5] Safwan Labbi, Daniil Tiapkin, Lorenzo Mancini, Paul Mangold, and Eric Moulines. Federated UCBVI: communication-efficient federated regret minimization with heterogeneous agents. AISTATS, 2025.

---

### Official Review · Reviewer_dRFT · 2025-03-16

**Overall Recommendation:** 3

**Summary:**

In the last few years, there has been an increase of interest in the topic of federated reinforcement learning (FRL), where agents interacting with similar (or exactly identical) MDPs communicate to achieve some speedup in performance. The authors consider such a formulation in the context of episodic finite-horizon problems, involving $M$ agents, each interacting with the same MDP. While this problem has been explored in the past, the authors improve upon prior results in the following ways:

- They achieve gap-dependent regret bounds in FRL, i.e., regret bounds that scale with $\log(T)$ and the inherent difficulty of the problem-instance, as opposed to the gap-independent $\sqrt{T}$ bounds.

- They manage to shave off from the $log(T)$ term some factors that depend on the number of states and actions.

- The dependence on the communication rounds is also improved from prior work.

**Claims And Evidence:**

The claims are well-supported by rigorous proofs.

**Essential References Not Discussed:**

All relevant references are discussed in adequate detail.

**Experimental Designs Or Analyses:**

There are no experimental designs since the main contributions of the paper are theoretical.

**Methods And Evaluation Criteria:**

Since this is a theoretical paper, there is no evaluation criteria as such. This is not a limitation of the paper.

**Other Comments Or Suggestions:**

In the description of the FedQ-Hoeffding algorithm from Section 3, I would urge the authors to provide more intuition behind the rationale for the update strategies of the Q-table under Cases 1 and 2. At a high-level, why are these cases treated separately? Is it this separate treatment that leads to the final improvements in the communication-complexity?

**Other Strengths And Weaknesses:**

Strengths
---------------
- This paper appears to be the first to establish gap-dependent regret and communication-complexity bounds in FRL.
- The ideas behind the design of the coordinated exploration scheme in FedQ-Hoeffding might apply more generally to similar episodic, finite-horizon problems.
- The authors provide a fairly detailed exposition of their technical analysis in the main body of the paper. This was very helpful to get a quick sense of the main proof ideas.

Weaknesses
-------------------
- My main concern with the paper is the following. Gap-dependent regret bounds are quite well known in the bandits and RL literature for the single-agent setting. This paper considers the homogeneous FRL setting where all agents interact with the *same* MDP. So essentially, this is a direct extension of the single-agent setting in a simplified tabular case with no function approximation. At least in the case where the agents communicate all the time, the gap-dependent bound analysis in the homogeneous FRL case seems a trivial extension of the single-agent counterpart. I am not sure if the extension to the case where agents communicate sporadically presents any major technical bottleneck.

- The removal of dependencies on $M$, $S$, and $A$ from $\log(T)$ is a nice contribution, but in the end, it is an improvement to a log-factor term. I do not necessarily consider this to be a significant contribution.

- Given that all agents interact with the same MDP, I was hoping that the final regret bounds would show some sort of a speedup w.r.t. the number of agents $M$, relative to when the agents don't communicate. Whether such an improvement is indeed reflected in the final regret bound was not apparent to me looking at Eq.~(1). Could the authors comment on this aspect?

- In terms of the improvement in communication claimed in Eq.~(2), I am not sure if this as significant as claimed. If I understand correctly, earlier communication bounds were on the order of $O(MSA \log(T))$, while the refined bound has dependencies of the form of $MSA \log(\log(T))$. Thus, *communication still depends linearly on the number of states and actions*; the improvement is simply in terms of a logarithmic factor multiplying the number of states and actions. Furthermore, there are additional terms governing the communication complexity that continue to depend on $|S|$ and $|A|$ through higher-order dependencies on the horizon length $H$, and logarithmic dependencies on the gap $\Delta_{min}$. As such, overall, it is unclear to me whether there is a non-trivial regime of the parameters $M, S, A, H, T, \Delta_{min}$, where the gains in communication complexity are a significant improvement over prior results.

- Since the paper is theoretical in nature, and the main contributions pertain to a refinement of prior bounds, I was hoping to see some lower bounds on the best achievable gap-dependent regret and the minimal number of communication rounds needed to achieve such a regret bound. Characterizing such lower bounds would shed light on the tightness of the guarantees presented in this paper. However, no such lower bounds are derived in this paper.

**Questions For Authors:**

I have no further comments or questions beyond those above.

**Relation To Broader Scientific Literature:**

The paper refines the understanding of the emerging FRL literature by sharpening prior regret and communication bounds.

**Theoretical Claims:**

I skimmed through the proofs of the main claims and they appear correct to me.

---

> ### Author Rebuttal · Authors · 2025-04-01
>
> We provide point-by-point responses to your thoughtful comments.
>
> **W1: A direct extension.** We respectfully disagree that extending to FRL is trivial. FRL introduces unique challenges and requires new techniques beyond those used in single-agent RL, especially in controlling the estimation error of value functions due to its algorithm design and communication constraints.
>
> Concurrent RL assumes full raw data sharing and coordination via visit count thresholds. Existing works provide only worst-case regret, and even with full communication, bounding the extra regret caused by this delayed policy update is non-trivial.
>
> FRL, in contrast, imposes privacy and communication constraints, making the setting more complex. Due to on-policy learning and policy switching, stationary visitation distributions are unavailable, and the central server cannot track visit counts directly. FedQ-Hoeffding addresses this challenge with a trigger condition (Eq. (4)) and future-dependent weights (line 1269), but breaks the martingale structure typically used in concentration inequalities.
>
> Also, FRL introduces a multi-agent burn-in cost $C_f$ in the regret. Directly applying single-agent techniques (e.g., Yang et al., 2021) would incur undesired dependency on $\Delta_{\min}$, which we avoid through delayed integration (see Lemma F.3).
>
> **W2, W4 pt.1: Comments on the log factor.** We respectfully believe that our improvement is significant, as this is the first gap-dependent communication cost bound for FRL, marking meaningful progress toward tight communication complexity with both theoretical and practical relevance.
>
> First, removing $M,S,A$ from the $\log T$ term is significant, especially in large-scale applications like recommender systems (Covington et al., RecSys, 2016) or text-based games (Bellemare et al. JAIR, 2013), where $S,A$ can be enormous.
>
> Second, the dependency of $M,S,A$ improves from $MSA\log T$ to $MSA\log\log T$. Since the policy-switching cost lower-bounds communication rounds under synchronized policies, our improvement matches key advances in Qiao et al. (2022) and Zhang et al. (2022b) on policy-switching cost, highlighting the improvement from $\log T$ to $\log\log T$ is both theoretically important and practically beneficial.
>
> Finally, the linear dependence on $S,A$ in the $\log\log T$ term is provably necessary, as Qiao et al. (2022) proves a lower bound of $O(HSA \log \log T)$ for policy switching cost. Compared to recent bounds scaling with $SA\log T$ for FRL (e.g., Zheng et al., 2024, 2025a; Labbi et al., 2024), our result marks meaningful progress toward this lower bound.
>
> **W3: Regret speedup.** Yes, our regret bound does reflect a speedup w.r.t. $M$. In our paper, the average regret per episode is $O(\frac{H^7SA\log(MSAT)}{MT\Delta_{\min}})$ for sufficiently large $T$. This indicates a nearly $1/M$ rate of improvement under gap-dependent analysis, compared to the single-agent case that has an average regret of $O(\frac{H^7SA\log(SAT)}{T\Delta_{\min}})$. Our result achieves a faster rate of improvement than the commonly observed $\sqrt{1/M}$ speedup in worst-case MDPs.
>
> **W4 pt.2: A non-trivial regime.** We respectfully argue that such regimes do exist. When $T>\tilde{\Theta}(\textnormal{Poly}(\frac{MH^2SA}{\Delta_{\min} C_{st}}))$ that is common in prior RL works (e.g., Zhang et al., 2020; Li et al., 2021), our gap-dependent result is strictly better than the prior bound $O(MH^3SA\log T)$. Moreover, for any $X\in\{M,S,A\}$, when $\log T>\Theta(X\log(\frac{X\log(MSAT)}{\Delta_{\min}C_{st}})) $ that is used in other RL contexts (e.g., Weisz et al., ALT, 2021), our bound is independent of $X$, in contrast to prior bounds that scale linearly with it.
>
> **W5: Lower bounds.** Finding lower bounds in FRL is an important open problem. However, due to the inherent trade-off between regret and communication, establishing meaningful joint lower bounds is challenging. At one extreme, no communication among agents results in zero communication cost but no speedup in regret; at the other, full communication achieves linear regret speedup at a communication cost that scales with $T$. Both deviate from FRL’s goal of balancing these two objectives.
>
> Still, existing results support the structure of our results. The regret lower bound of $O\left(\frac{HSA}{\Delta_{\min}}\log\left( \frac{T}{H}\right)\right)$ for single-agent RL in Xu et al. (2021) justifies the linear dependence on $S,A,\Delta_{\min}^{-1}$ and $\log T$ in our result. Linear dependence in $H$ remains an open question in the literature.
>
> As for communication cost, although no gap-dependent or even worst-case lower bounds exist in FRL, the linear dependency on $S$, $A$, and $\log\log T$ is provably necessary, as noted in our response to W2.
>
> **Questions on two cases.** These two cases were proposed in Zheng et al. (2024) to reduce the estimation error of value functions. This design does not contribute to our improvement in communication cost.

---

> > ### Comment · Reviewer_dRFT · 2025-04-03
> >
> > Thank you for the rebuttal. I am satisfied with the comments, and have adjusted my score accordingly.

---

### Decision · Program_Chairs · 2025-05-01

**Decision:**

Accept (poster)

**Comment:**

This paper studies the gap-dependent bounds for online federated Q-learning where multiple agents collaboratively explore and learn a global Q-function under tabular MDP setting. The authors provide analysis of both performance regret and communication cost of the online federated Q-learning algorithm (FedQ-Hoeffding), proposed in the previous work, in terms of gap-dependent parameters.

Reviewers think the results are strong and the techniques are novel. The AC agrees and thus recommends acceptance.